# Systemic short chain fatty acids limit antitumor effect of CTLA-4 blockade in hosts with cancer

Clélia Coutzac [1,2,3,4], Jean-Mehdi Jouniaux[1,2], Angelo Paci [5,6,7], Julien Schmidt[1,2], Domenico Mallardo [8], Atmane Seck[5,6], Vahe Asvatourian[9,10], Lydie Cassard [1], Patrick Saulnier[11], Ludovic Lacroix[11], Paul-Louis Woerther[12], Aurore Vozy[1], Marie Naigeon[1], Laetitia Nebot-Bral[2,13], Mélanie Desbois [1], Ester Simeone[8], Christine Mateus[14], Lisa Boselli[1], Jonathan Grivel[1], Emilie Soularue[1,2,15], Patricia Lepage [16], Franck Carbonnel[2,15], Paolo Antonio Ascierto [8,17], Caroline Robert [2,14,17] & Nathalie Chaput [1,7,13✉]

Gut microbiota composition influences the clinical benefit of immune checkpoints in patients with advanced cancer but mechanisms underlying this relationship remain unclear. Molecular mechanism whereby gut microbiota influences immune responses is mainly assigned to gut microbial metabolites. Short-chain fatty acids (SCFA) are produced in large amounts in the colon through bacterial fermentation of dietary fiber. We evaluate in mice and in patients treated with anti-CTLA-4 blocking mAbs whether SCFA levels is related to clinical outcome. High blood butyrate and propionate levels are associated with resistance to CTLA-4 blockade and higher proportion of Treg cells. In mice, butyrate restrains anti-CTLA-4-induced up-regulation of CD80/CD86 on dendritic cells and ICOS on T cells, accumulation of tumor-specific T cells and memory T cells. In patients, high blood butyrate levels moderate ipilimumab-induced accumulation of memory and ICOS + CD4 + T cells and IL-2 impregnation. Altogether, these results suggest that SCFA limits anti-CTLA-4 activity.

[1] Université Paris-Saclay, Institut Gustave Roussy, Inserm, CNRS, Analyse moléculaire, modélisation et imagerie de la maladie cancéreuse, Laboratoire d'Immunomonitoring en Oncologie, F-94805 Villejuif, France. [2] Université Paris-Saclay, Faculté de Médicine, Le Kremlin Bicêtre F-94276, France. [3] Université Paris-Descartes, Faculté de Médicine, F-75006 Paris, France. [4] Hôpital Européen Georges Pompidou, Département de Gastroentérologie et Oncologie Digestive, Assistance Publique-Hôpitaux de Paris, F-75015 Paris, France. [5] Université Paris-Saclay, Institut Gustave Roussy, CNRS, Vectorologie et thérapeutiques anticancéreuses, F-94805 Villejuif, France. [6] Institut Gustave Roussy, Pharmacology and Drug Analysis Department, Villejuif F-94805, France. [7] Université Paris-Saclay, Faculté de Pharmacie, Chatenay-Malabry F-92296, France. [8] Unit of Melanoma, Cancer Immunotherapy and Development Therapeutics, Instituto Nazionale Tumori- IRCCS –Fondazione G. Pascale, Napoli, Italia. [9] Institut Gustave Roussy, Biostatistics and Epidemiology Unit, Villejuif F-94805, France. [10] Université Paris-Saclay, UVSQ, Inserm, CESP, 94807 Villejuif, France. [11] Université Paris-Saclay, Institut Gustave Roussy, Inserm, CNRS, Analyse moléculaire, modélisation et imagerie de la maladie cancéreuse, Genomic platform Molecular Biopathology unit and Biological Resource Center, F-94805 Villejuif, France. [12] Institut Gustave Roussy, Department of Medical Biology and Pathology, Microbiology unit, Villejuif F-94805, France. [13] Université Paris-Saclay, Institut Gustave Roussy, CNRS, Stabilité génétique et oncogenèse, 94805 Villejuif, France. [14] Institut Gustave Roussy, Dermatology Unit, Department of Medicine, Villejuif F-94805, France. [15] Hôpital du Kremlin Bicêtre Department of Gastroenterology, Assistance Publique-Hôpitaux de Paris, Le Kremlin Bicêtre, France. [16] Université Paris-Saclay, INRAE, AgroParisTech, Micalis Institute, 78350 Jouy-en-Josas, France. [17]These authors contributed equally: Paolo Antonio Ascierto, Caroline Robert ✉email: nathalie.chaput@gustaveroussy.fr

In the past ten years therapeutic strategies in oncology have been revolutionized by the use of immune checkpoint blocking antibodies. In clinical practice, CTLA-4 blockade using the therapeutic monoclonal antibody (mAb) ipilimumab (anti-CTLA-4) was approved as standalone or in combo with nivolumab (anti-PD-1) in patients with metastatic melanoma (MM)[1–4]. Approximately 20 and 60% of patients with MM have an objective clinical response with ipilimumab and with ipilimumab/nivolumab combination respectively. Recently, studies suggested that gut microbiota composition was associated with antitumor efficacy in patients with MM treated with anti-CTLA-4[5,6] and anti-PD-1[7] mAb. The composition of gut microbiota also seems to be associated with the risk to develop anti-CTLA-4-induced colitis in patient with MM[6,8]. These recent studies highlight specific bacteria associated with clinical response and/or toxicities. Especially, baseline gut microbiota enriched with Faecalibacterium and other Firmicutes was associated with beneficial clinical response to ipilimumab, anti-PD-1, and ipilimumab/anti-PD-1 therapy in melanoma patients[6,7,9]. Based on these independent works, it appears that Faecalibacterium might represent a major feature associated with clinical response in MM patients treated with immune checkpoints. However, scarce findings explain how the gut microbiota composition could have an impact on a distant tumor lesion. In mice, anti-CTLA-4 blocking mAb was shown to induce a dysbiosis favoring the translocation of commensal bacteria that might allow IL-12-secretion by dendritic cells (DCs) as well as the priming of commensal-specific Th1 cells that could migrate to the tumor and recognize tumor cells due to antigen mimicry[5]. Another mechanism was described in mice treated with anti-PD-L1, where specific bacteria (i.e., Bifidobacterium) was associated with an increase DC maturation leading to enhance CD8$^+$ T cell priming and accumulation in the tumor beds under PD-L1 blockade conditions[10]. In patients with MM treated with anti-PD-1, enriched Faecalibacterium was linked to higher CD8$^+$ T cell tumor infiltrate[7]. In addition to the direct effect of commensal bacteria on immune system, it is well known that some bacterial groups produce metabolites that have also immune properties[11]. These de novo synthesized metabolites include short-chain fatty acid (SCFA), mainly acetate (C2), propionate (C3) and butyrate (C4). SCFA mediate several functions especially providing energy to intestinal epithelial cells (IEC)[12,13]. SCFA also play a pivotal role on immune modulation[11]. Butyrate is well known to exert systemic anti-inflammatory activities by affecting immune cell migration, adhesion, cytokine expression as well as affecting cellular processes such as proliferation, activation, and apoptosis[14]. Considering previous results on the association between gut microbiota composition and clinical response and the effect of SCFA on the immune system, even at distant site, we hypothesized that anti-cancer response due to anti-CTLA-4 blockade may be influenced by systemic microbial SCFA. In this study, we demonstrate that microbial systemic SCFA (butyrate and propionate) influence anti-CTLA-4 anti-tumor effect in mice models and in patients with MM and treated with ipilimumab.

## Results

**Microbiota composition and clinical outcomes in patients.** As previously analyzed and replicated with sequencing technology, baseline microbiota enriched in Faecalibacterium prausnitzii (F. prausnitzii) and other Firmicutes was associated with better outcome in a French cohort of 26 MM patients treated with ipilimumab[6]. In the present study, among the fifty MM patients included, 16S rDNA analyses were performed on 38 fecal samples at baseline ($V_1$) (Supplementary information and Supplementary Tables 1, 2 and 3). We analysed the main genera composition

(Fig. 1a). Genera linked to long-term clinical benefit (LTB; progression free survival > 6 months) were Faecalibacterium and Gemminger (Fig. 1b). High proportions of Bacteroides could be found in patients with poor clinical benefit but no statistical significance was reached compared to patients with LTB (Fig. 1b). Other genera were not associated with clinical outcome (Fig. 1b). Altogether, Faecalibacterium might represent a good surrogate marker of LTB. Considering ipilimumab-induced colitis, a tendency for higher proportions of Faecalibacterium, Lachnospiracae and Gemminger, and lower proportion of Prevotella were observed in patients that develop ipilimumab-related colitis even though not significant (Supplementary Fig. 1). High relative abundance of Faecalibacterium at baseline was linked to overall survival (OS) over than 18 months (Fig. 1c). Kaplan–Meier analyses of patients classified into two groups according to median value of the abundance of Faecalibacterium, showed that high baseline Faecalibacterium was associated with longer progression free survival (PFS) (Fig. 1d). Note that Faecalibacterium, Gemminger[6], Roseburia as well as Bifidobacterium genera described in another study as associated with clinical efficacy after anti-PD-L1 treatment[10] were positively correlated whereas Bacteroides genera was inversely correlated to Faecalibacterium (Supplementary Fig. 2).

Then, we applied quantitative PCR (Q-PCR) analyses on baseline feces in the cohort for F. prausnitzii, Bacteroides fragilis (B. fragilis) and Escherichia coli (E. coli) (Supplementary Table 1 and 3). Quantitative-PCR and 16S rDNA analysis data were highly correlated (Fig. 1e and Supplementary Fig. 3). All patients with an OS longer than 18 months showed higher proportions of F. prausnitzii contrary to B. fragilis (Fig. 2a). Interestingly, abundance of F. prausnitzii and bacterial load were correlated together (Supplementary Fig. 4) and were linked to improve PFS in patients (Fig. 2b). No such correlation and difference could be observed for B. fragilis[5] or E. coli assessed as controls (Fig. 2a, Supplementary Fig. 5). Altogether, these data indicated that microbiota composition and particularly enrichment with F. prausnitzii using Q-PCR analysis could be linked to clinical outcome in patients treated with ipilimumab as previously suggested using 16S rRNA gene sequencing[6].

**Serum SCFA concentrations and clinical outcomes in patients.** The gut microbiota is able to influence immune responses mainly through gut microbiota metabolites such as SCFA (acetate, propionate and butyrate), indoles derivatives, polyamines and others, produced in very large amounts in the colon through bacterial fermentation of dietary fiber[15,16]. Our hypothesis was driven by previous results concerning the association between the response to ipilimumab and the gut microbiota enriched in Faecalibacterium and other Firmicutes[6], known to be as butyrate producers. These data led us to assume that SCFA might play a mandatory role in ipilimumab efficacy. We determined the concentration as well as proportion, in stools and serum, of acetate, propionate and butyrate in the French cohort of patients with MM treated with ipilimumab (Supplementary Fig. 6). In stools, quantity and ratio between SCFA (3:1:1 for acetate:propionate:butyrate) were as previously described[17]. Correlations between serum and stool SCFA concentrations were observed for butyrate ($r = 0.50$; $p = 0.0009$) and to lower extent for propionate ($r = 0.35$; $p = 0.035$) (Supplementary Fig. 7). Surprisingly, Firmicutes phylum as well as F. prausnitzii were inversely linked with serum concentration of butyrate (Fig. 2c, d and e). No such association could be observed for B. fragilis or E. coli (Supplementary Fig. 8). We determined SCFA serum levels in an independent Italian cohort of patients with MM and treated with ipilimumab (Supplementary Table 1 and Supplementary Fig. 9). Serum butyrate

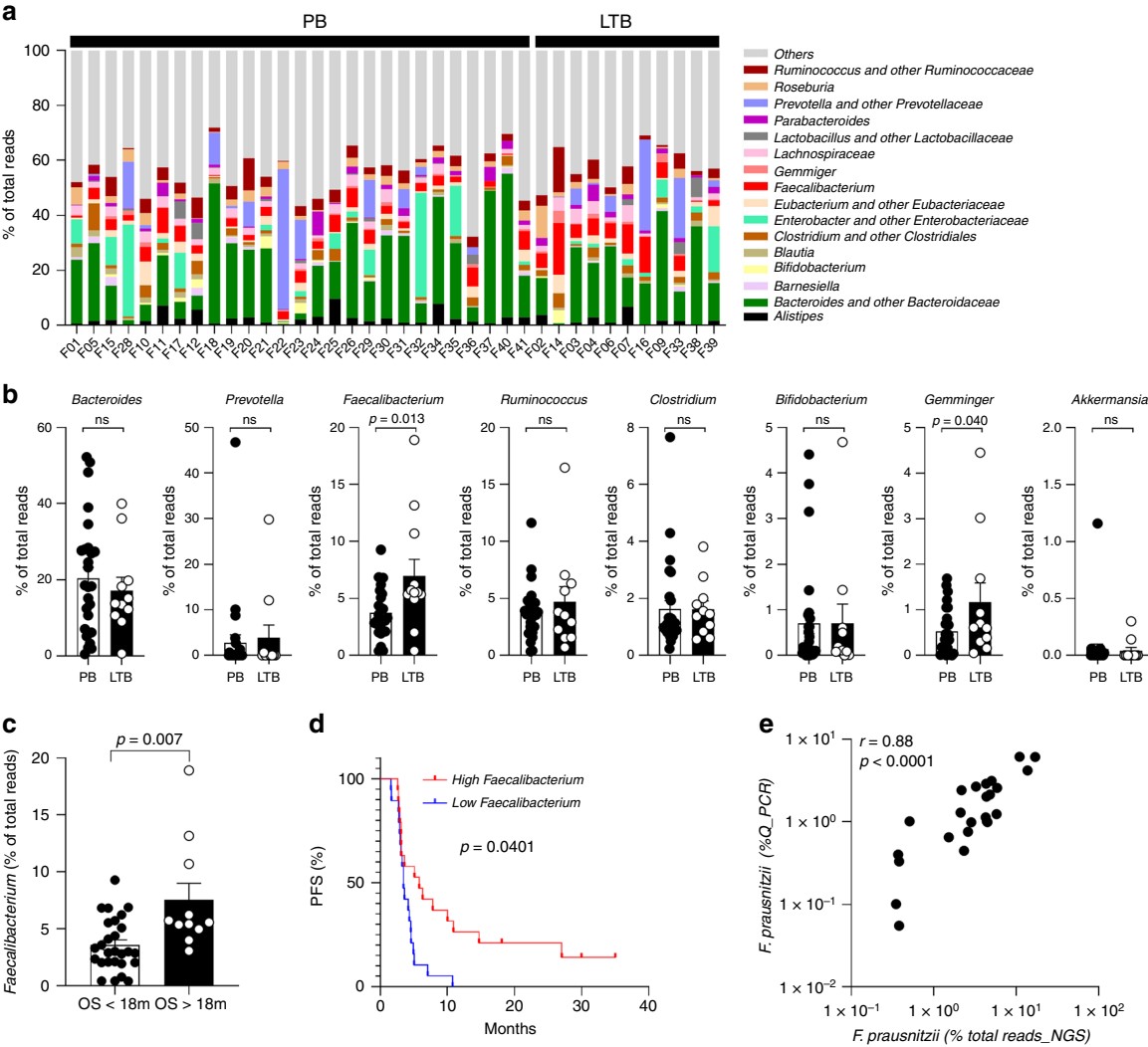

**Fig. 1 Baseline gut microbiota composition in patients with MM. a** Relative abundance of dominant (>1% of total reads) gut microbial genera are represented for each patient at V1 (baseline, $n = 38$). **b** Histograms of relative abundance of eight genera at baseline between poor (PB) and long-term clinical benefit (LTB) patients. Mann–Whitney two-tailed tests were used. $p$-values are indicated on each graph. **c** Histograms of relative abundance of *Faecalibacterium* according to overall survival (OS over 18 months (OS > 18 m) and OS under 18 months (OS < 18 m)) in patients. Mann–Whitney two-tailed tests were used. $p$-values are indicated on the graph. **d**, Kaplan–Meier curve of progression-free survival (PFS) of patients classified into two groups according to the median relative abundance of *Faecalibacterium* ($n = 38$). Log-rank (mantel-cox) test was used. **e** Pearson correlation between the relative abundance of *Faecalibacterium* assessed by 16S rDNA analysis (% of total reads NGS) and Q-PCR (% Q-PCR) analyses on feces at baseline ($n = 24$). $r$ and $p$ values are indicated on each graph. ns, not significant.

and propionate were correlated together but not with acetate in both cohorts (Supplementary Fig. 10). Kaplan Meier analyses of patients classified into two groups according to median value of serum SCFA concentrations, showed that low baseline butyrate was associated with longer PFS in the French cohort (Fig. 2f, $p = 0.0054$) and low baseline propionate was associated with longer PFS in the Italian cohort (Fig. 2g, $p = 0.0010$). Serum butyrate concentration tended to be associated with PFS in the Italian cohort (Fig. 2g, $p = 0.0688$). No such association was observed for serum acetate in both cohorts in Kaplan–Meier analyses (Supplementary Fig. 10). Moreover, serum butyrate and propionate were inversely correlated to PFS and OS (Fig. 2h and i) while no association could be found for acetate. All data from both cohorts were pooled ($n = 85$), both butyrate and propionate were negatively correlated to clinical outcomes (Supplementary Fig. 10d). Moreover, Kaplan Meier analyses of patients classified into two groups according to median value of serum SCFA concentrations, showed that both low baseline butyrate and low baseline

propionate was associated with longer PFS in the pooled cohort (Supplementary Fig. 10e, $p = 0.0015$ and $p = 0.0029$ respectively). These results suggest that serum butyrate and propionate concentrations could represent an indirect systemic marker of the microbiota composition associated with clinical outcomes in patients treated with ipilimumab.

**Butyrate reduces efficacy of CTLA-4 blockade in mice.** To better understand if SCFA may have a direct role on anti-CTLA-4 clinical efficacy, sodium butyrate was given to mice in drinking water at the concentrations of 100 mM and was compared to mice drinking pH-matched water, two weeks before tumor inoculation and all along experiments. Mice also received i.p. injections of anti-CTLA-4 or its isotype control (IgG2b) (Fig. 3a and Supplementary information). We measured SCFA concentrations in mice serum to confirm higher levels of butyrate in supplemented mice. Despite some heterogeneity due to the variability of water intake between mice, concentrations of

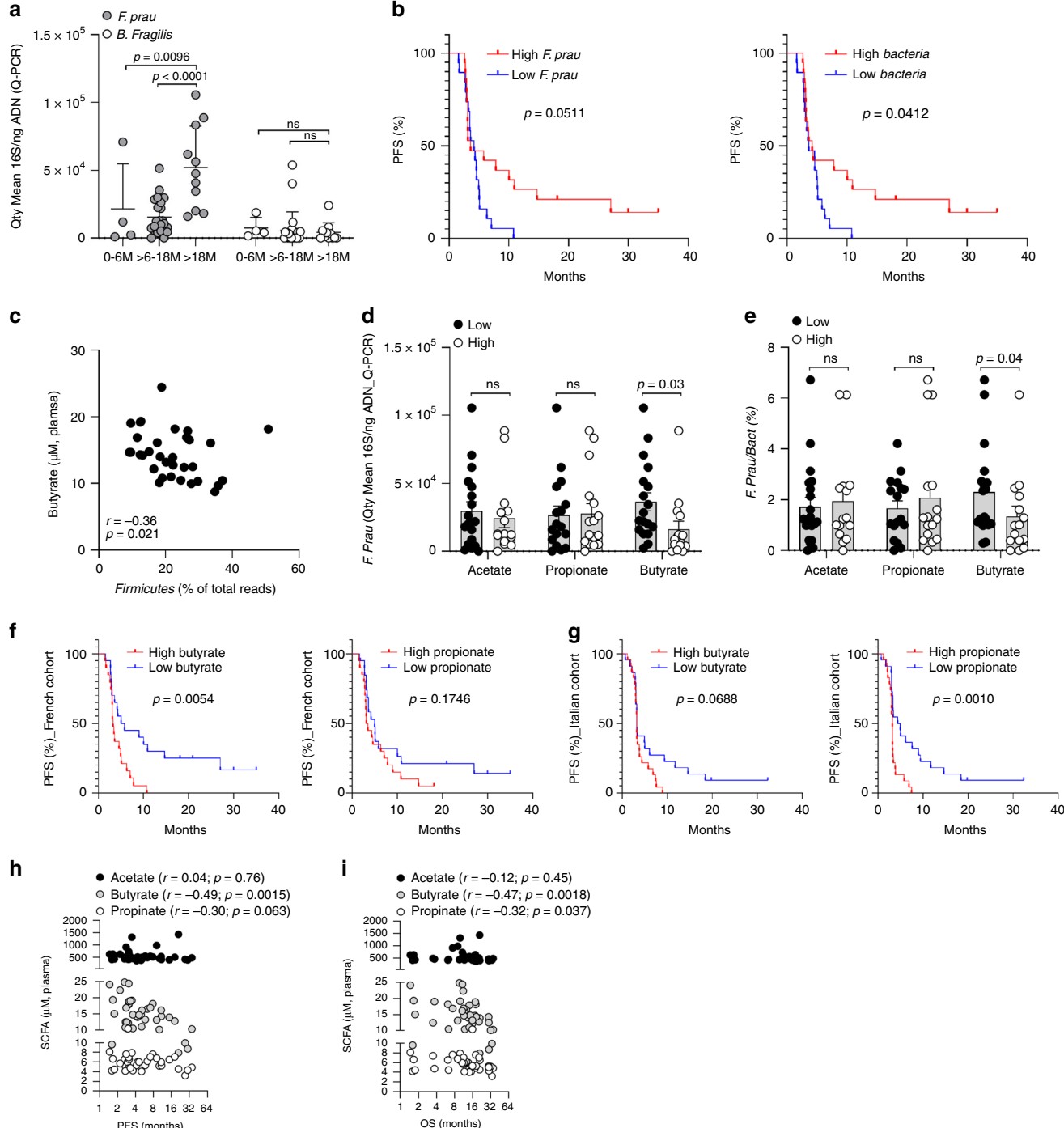

**Fig. 2 Association between *F. prausnitzii*, systemic concentrations of SCFA and clinical outcomes. a** Quantity Mean 16S/ng DNA of *F. prausnitzii* and *B. fragilis* at baseline (Q-PCR). Each patient's microbiota is presented in the graph ($n = 38$). 0–6 M: overall survival ranging from 0 to 6 months ($n = 4$), >6–18 M: overall survival ranging from 6 to 18 months ($n = 23$), >18 M: overall survival >18 months ($n = 11$). Two-Way ANOVA followed by Tukey's tests for multiple comparisons was performed. *p*-values are indicated on each graph; ns, not significant. **b** Kaplan–Meier survival curves of PFS of patients classified into two groups according to median of *F. prausnitzii*/ng DNA (Low vs high *F. prau*; left panel) and 16S BACTERIA/ ng DNA (low vs high Bacteria; right panel). Date were performed using Q-PCR analyses on feces at baseline ($n = 38$), Log-rank (mantel-cox) test was used **c** Spearman correlation between the relative abundance of *Firmicutes* assessed by 16S DNA analysis (% of total reads NGS) and serum butyrate at baseline ($n = 33$). **d** Q-PCR analyses on feces at baseline of *F. prausnitzii* (16S *F.prau*/ng DNA) according to the median concentrations of serum acetate, propionate and butyrate ($n = 33$). **e** As in (**d**) but proportion ($n = 33$). Each dot represents a mean quantity (assessed in duplicate for each sample) for one patient. Two-tailed Mann–Whitney tests were used (**d**, **e**). *p*-values are indicated on each graph; ns, not significant. Kaplan–Meier survival curves of PFS according to the median value of serum propionate (left panels) and butyrate (right panels) concentrations in French (**f**, $n = 40$) and Italian (**g**, $n = 45$) cohort, Log-rank (mantel-cox) test was used. **h** Pearson correlations between serum concentrations of acetate, butyrate and propionate and PFS ($n = 40$) in French cohort; **i** As in (**h**) but for OS ($n = 40$) in French cohort. Each dot represents one patient. *p* and *r* are indicated on each graph.

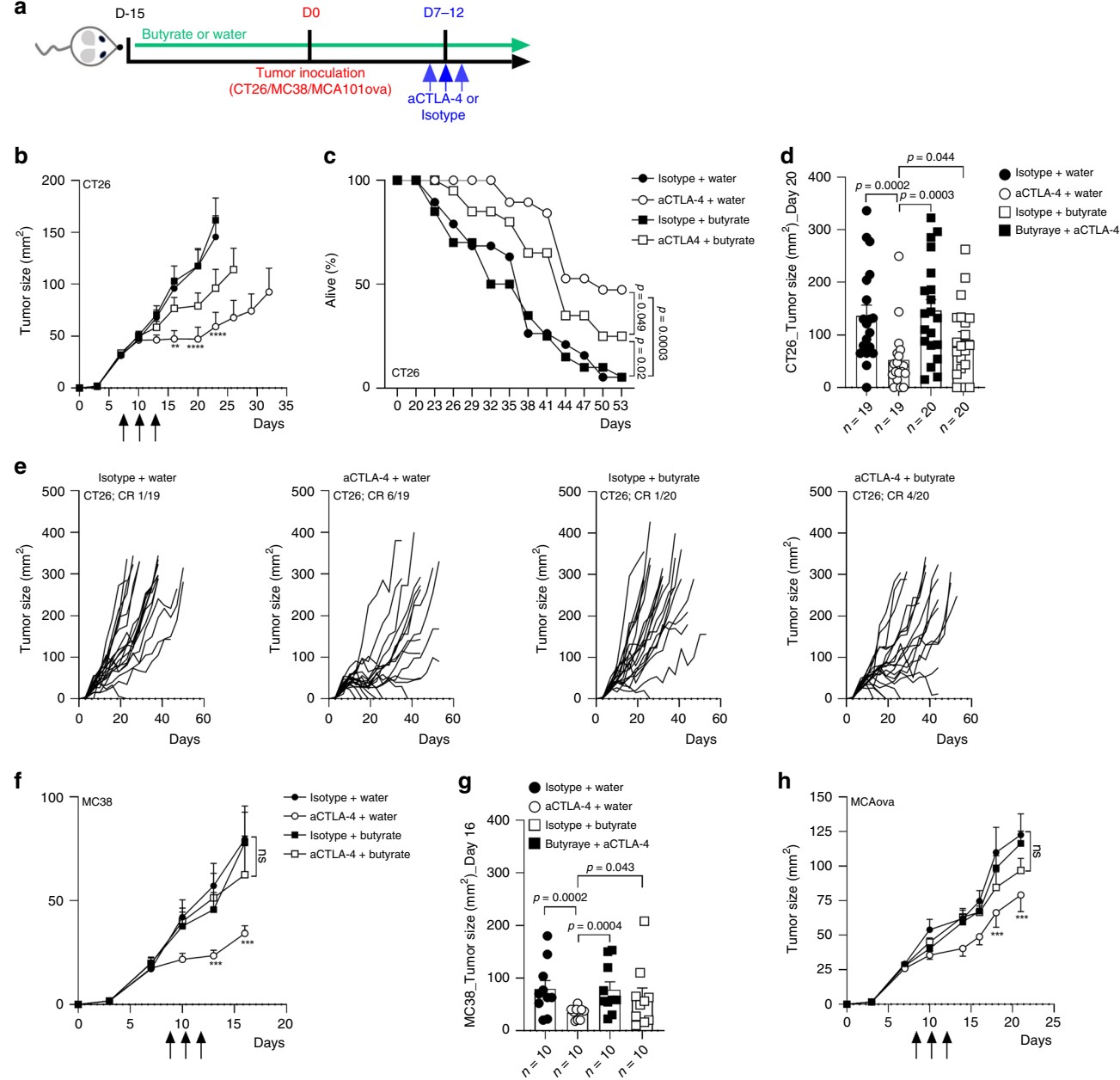

**Fig. 3 Sodium butyrate diminished antitumor efficacy of CTLA-4 blockade in mice. a** Experimental setting. **b** Tumor growth of CT26 in Balb/c mice. White circles anti-CTLA-4 (aCTLA-4; clone 9D9) + water ($n = 19$); Black circles isotype control (IgG2b; clone MPC-11) + water ($n = 19$); white squares anti-CTLA-4 (aCTLA-4) + sodium butyrate in drinking water ($n = 20$); black squares isotype control + sodium butyrate in drinking water ($n = 20$); **p = 0.0029 and ****p < 0.0001. **c** % of survival in each group at each day of the experiment. **d** As in (**b**), but median tumor size (mm$^2$ +/− SEM) at Day 20 in each group. **e** As in (**b**) but, CT26 tumor size over time after tumor inoculation in each group. Each line represents one mouse. Complete responses (CR) are indicated on each graph. **f** Tumor growth of MC38 in C57/Bl6 mice ($n = 10$ per group); **p = 0.0098 and ***p = .0002. **g** As in (**f**) but median tumor size (mm$^2$ +/− SEM) at Day 16 in each group. **h** Tumor growth of MCA101$_{OVA}$ in C57/Bl6 mice ($n = 9$ (Isotype + water and aCTLA-4+water) or 10 (Butyrate + water and aCTLA-4+butyrate) per group); ***p = 0.0001. Each dot represents one mouse; a pool of two independent experiments is shown (**b**, **c**, **d** and **e**); one representative experiment out of two experiments is shown (**f** and **h**). Two-way ANOVA was used (**b**, **f**, **g** and **h**); Log-rank (Mantel-Cox) test was used for comparison of two-to-two survival curves (**c**). Only aCTLA-4 +water and Isotype + water p-values are shown on the graphs (**b**, **f** and **h**). Mann Whitney (two-tailed) test was used (**d**, **g**). Exact p-values are indicated on each graph; ns, not significant.

butyrate, as well as the proportion of butyrate among total SCFA, were higher in supplemented mice (Supplementary Fig. 11). Conversely, the concentrations of acetate and propionate were lower in butyrate-supplemented mice (Supplementary Fig. 11). Three tumors models were used (CT26, MC38 and MCA101$_{OVA}$). In the CT26 tumor model, stabilization as complete regression of the tumor growth were observed in mice

treated with anti-CTLA-4 without sodium butyrate (Fig. 3b) as well as longer survival (Fig. 3c). In butyrate groups, although CTLA-4 blockade could exert some anti-tumor effect (Fig. 3b, c), no significant reduction of the tumor growth was observed at day 20 (D20) compared to control group (Fig. 3d). Moreover a significant difference was observed between anti-CTLA-4+ water and anti-CTLA-4+ butyrate (Fig. 3d). In line with these results,

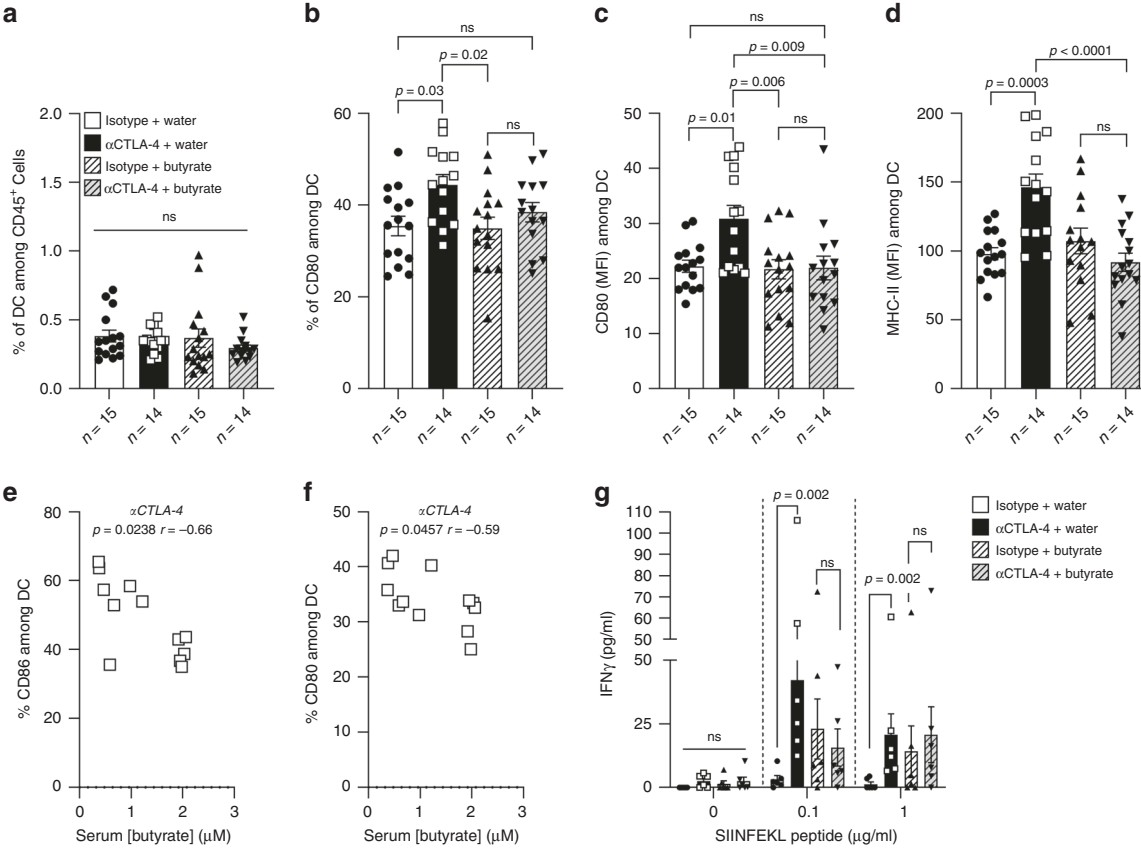

**Fig. 4 Sodium butyrate inhibited anti-CTLA-4-induced DC maturation and T cell priming. a** Percentages of dendritic cells (DC defined as CD3⁻CD19⁻NKp46⁻CD11c⁺MHC-II⁺ among CD45⁺ cells) in tDLN from three pooled independent experiments. **b** Percentages (%) of CD80 among DC in the tDLN from pooled independent experiments. **c** Mean fluorescence intensity of CD86 among DC in the tDLN. **d** Mean fluorescence intensity (MFI) of MHC class-II among DC in the tDLN. **e, f** Correlation between the percentage of CD86 (**e**) and CD80 (**f**) and serum butyrate concentration in mice treated with anti-CTLA-4 ($n = 12$ mice) **g** IFNγ concentration after ex vivo restimulation of tDLN with MHC-I OVA (SIINFEKL) peptides from one representative experiment out of two independent experiments ($n = 6$ mice per group). *p*-values are indicated on each graph; Each dot represent one mouse (**a, b, c, d, e, f** and **g**). *n* (number of mice per group) is indicated on the graph (**a, b, c** and **d**). Bars represent mean +/− sem (**a, b, c, d** and **g**). One-way ANOVA followed by Tukey's tests for multiple comparisons was used (**a, b, c** and **d**). Spearman test were used (**e** and **f**). Two-tailed Mann–Whitney tests were used (**g**). *p*-values are indicated on each graph; ns, not-significant.

stabilization (SD) and complete response (CR) rate were 37% and 32% respectively in anti-CTLA-4+ water group compared to 25 and 20% in anti-CTLA-4+ butyrate group (Fig. 3e). MC38 and MCA101_OVA tumor models were less responsive to anti-CTLA-4 with no CR observed, in these two models adding butyrate abolished the anti-tumor effect of anti-CTLA-4 (Fig. 3f–h). Altogether these data provided evidence that butyrate could diminish anti-CTLA-4 anti-tumor efficacy in mice.

**Butyrate limits anti-CTLA-4-induced DC maturation.** Butyrate has been described to modulate immunity and inflammation[18]. Note that within SCFA, butyrate and propionate are often associated with induction of regulatory T cells (Treg) contrary to acetate which has been mainly incriminated in the modulation of B cells immunity[19–22]. We hypothesized that butyrate could interfere with anti-CTLA-4-induced immune modulation. In the MCA101_OVA model where antigen-specific T cell responses could be monitored, no modification of CTLA-4 expression levels within T cells was observed in tumor-draining lymph nodes (tDLN) in any groups (Supplementary Figs. 12 and 19). Butyrate was shown to prevent DC maturation in vitro[23,24]. In our mouse model, CTLA-4 blockade did not modify the proportion of DC in tDLN (Fig. 4a). However, it increased the proportion of DC expressing CD80 and the density of CD86 on the DC surface.

Modulations of CD80 and CD86 expression were not observed in mice treated with butyrate + anti-CTLA-4 (Fig. 4b and c and Supplementary Fig. 20). For MHC Class II molecules (MHC-II) expression among DC, we observed a significant difference between mice treated with water + anti-CTLA-4 (Mean MFI MHC-II +/− SD: 150.7 +/− 40.26) and mice treated with butyrate + anti-CTLA-4 (Mean MFI MHC-II +/− SD: 85.62 +/− 33.58, $p = 0.026$; Fig. 4d). Finally, a negative correlation between the proportion of CD86 and CD80 among DC and serum concentration of butyrate in anti-CTLA-4 treated mice (Fig. 4e and f) was detected. Overall, these data suggested that butyrate limited anti-CTLA-4-induced maturation of DC in the tDLN in mice.

**Butyrate restrains anti-CTLA-4-induced OVA-specific T cells.** We next evaluated OVA-specific CD8⁺ T cell induction in the tDLN of MCA101_OVA bearing mice. In vitro restimulation of tDLN with MHC class I (MHC-I; SIINFEKL) evidenced in mice that drank butyrate-free water and treated with anti-CTLA-4, a significant induction of IFNγ compared to mice treated with isotype control (Fig. 4g). No such difference could be observed in mice that drank sodium butyrate-supplemented water following anti-CTLA-4 therapy (Fig. 4g). In butyrate mice and not treated with anti-CTLA-4 (isotype group), a slight augmentation of IFNγ was observed following restimulation with 0.1 and 1 μg/mL of

SIINFEKL compared to control without SIINFEKL even though not significant (Fig. 4g). This might be explained by the ability of butyrate to up-regulate MHC/antigen complexes through its histone deacetylase (HDAC) inhibitory activity[25,26]. In our hands, a clear up-regulation of MHC molecules was observed in vitro after butyrate treatment at the tumor cell surface in a dose-dependent manner (Supplementary Fig. 13). This capacity could favor an increase of the tumor immunogenicity in butyrate-supplemented mice leading to a slight augmentation of IFNγ after ex vivo restimulation. Despite this, the anti-tumor efficacy of anti-CTLA-4 was reduced in mice supplemented with sodium butyrate. These results suggest that butyrate was able to restrain OVA-specific T cell responses following CTLA-4 blockade. Furthermore, previous studies suggested that increased peripheral memory CD4[+] T cells[27] as well as ICOS expression on T cells[28] after anti-CTLA-4 treatment was associated with clinical benefit in MM patients. We thus analyzed the proportion of memory CD4[+] T cells (CD44[hi]CD62L[−]among CD4[+]CD3[+]CD45[+]) and ICOS expression in a longitudinal immune monitoring in the whole blood of mice. As previously described in patients, anti-CTLA-4 could induced accumulation of memory T cells as well as ICOS[+] T cells in the blood of treated mice (Supplementary Figs. 14 and 21). Interestingly, accumulation of memory T cells as well as ICOS induction after anti-CTLA-4 treatment was reduced in butyrate supplemented mice (Supplementary Fig. 14). We also analyzed the proportion of memory CD4+ T cells in the tDLN and we observed an augmentation of memory CD4+ T cells following anti-CTLA-4 treatment compared to isotype group (Supplementary Figs. 15 and 21) that was not observed in butyrate + anti-CTLA-4 treated mice compared to butyrate + isotype group.

**Patient's baseline immune markers and serum butyrate.** Butyrate and propionate have been previously described as immunoregulatory metabolites capable of favouring regulatory T cells[19,20]. However no or scarce data exist in humans in vivo, on peripheral Treg cells and the link with systemic SCFA concentrations. We observed that serum butyrate and propionate before introduction of ipilimumab, were correlated with Treg and sCD25 (Fig. 5a, b, c and d and Supplementary Fig. 22). Note that in butyrate supplemented mice accumulation of Treg cells in the blood and in the tDLN was also observed (Supplementary Figs. 16, 17 and 19). To test if micromolar (μM) concentrations of butyrate could increase proportion of Treg cells, we stimulated PBMC with concentrations close to those found in the serum of patients plus or minus anti-CD3ε stimulation. As shown in Fig. 5e a significant increase of Treg was observed in culture after CD3 stimulation at 50 and 100 μM but not at lower concentration i.e 10 μM. This data seemed consistent with observations made in patients where an expansion of Treg cells was observed only at higher concentrations. SCFA were proposed as anti-inflammatory metabolites through an inhibition of NF-kappaB[29,30]. In our cohort of patients we found that higher concentrations of butyrate and propionate were associated to higher concentrations of inflammatory proteins such as TNFα and MCP-1 (Fig. 5f and g). These results seemed in discordance with previous studies. However, in most of published studies demonstrating the anti-inflammatory properties of SCFA, millimolar concentrations are used in in vitro assays[29,30]. In these different studies, it clearly appeared that NF-KappaB inhibition was dose and time dependent without any effect observed at low concentrations such as concentrations observed in the serum of our patients[30]. Using concentrations close to those found in the serum of patients no inhibition of the production of inflammatory proteins was observed (Fig. 5h) confirming that at these μM concentrations no

inhibition of NF-kappaB might be induced. However it doesn't mean that these μM concentrations could induce inflammation. Indeed, we believe that the concentration of butyrate (or propionate) found in the stool and therefore at the periphery is greater when these metabolites are less locally used in the gut thus their action on the intestinal barrier might be limited. To test this hypothesis we measured zonulin a marker of intestinal permeability[31,32], in the serum of patients. Zonulin was correlated with stools' concentrations of propionate and butyrate (Fig. 5i) suggesting that higher concentrations of butyrate and propionate remaining in the stool of patients were related to an increased intestinal permeability. Since increased intestinal permeability is also related to inflammation, we assessed correlation between zonulin and serum inflammatory proteins; we found that inflammatory proteins such as TNFα and MCP-1 were correlated with zonulin (Fig. 5j). In conclusion, we believe that the link between measurable SCFA and peripheral inflammation was not a direct cause-and-effect relationship, but was rather a refection of intestinal permeability.

**Immune blood markers during the course of ipilimumab.** We assessed longitudinal modification of immune parameters in patients treated with ipilimumab according to serum butyrate concentration and clinical response. Butyrate levels were not modified during ipilimumab treatment as well as other SCFA (Supplementary Fig. 6). We and others have previously documented that sCD25 and accumulation of CD4[+] memory T cells were rapidly increased by anti-CTLA-4 treatment in patients that will finally have a long-term clinical benefit[6,27,33]. T-distributed stochastic neighbor embedding (t-SNE) analysis of longitudinal fresh whole blood immune monitoring of CD3[+] T cells (V$_1$ versus V$_{2-3}$, Fig. 6a, b) was realized in patients according to clinical response. Modification of t-SNE map between V$_1$ and V$_{2-3}$ was different in patients with long-term clinical benefit compared to patient with poor clinical benefit (Fig. 6c). Accumulation of memory CD4[+] and CD8[+] T cell populations was observed (Fig. 6c and Supplementary Fig. 23) only in patients that benefit from treatment. Accordingly in patients with low butyrate concentrations at baseline, higher increase of memory CD4[+] T cells was observed following ipilimumab compared to patients with high serum butyrate as demonstrated by post-ipilimumab to baseline ratio (Fig. 6d). In line with this observation, no or weak augmentation of sCD25, a marker of IL-2 impregnation, was observed during the course of ipilimumab in patients with high butyrate compared to those with low butyrate levels (Fig. 6e) as well as ICOS induction on CD4[+] T cells (Fig. 6f). Since butyrate inhibited anti-CTLA-4-induced immunity, we assume that butyrate might counter CD28 signaling. To address this specific issue, we performed in vitro experiments with escalating concentration of butyrate (0, 10, 50, and 100 μM). PBMC were stimulated in complete medium alone or with anti-CD3ε to mimic TCR engagement or anti-CD3ε + anti-CD28. This latest condition recapitulates what should be observed after ipilimumab where the blockade of CTLA-4 favors the engagement of CD28. As shown in Fig. 6, butyrate decreased sCD25 secretion (Fig. 6g) and ICOS induction (Fig. 6h) in a dose dependent manner only in the condition where both CD3 and CD28 were engaged. These data demonstrates that butyrate counters CD28 engagement/ signaling and are in lines with observations made in patients where high levels of butyrate was associated with low induction of ICOS and sCD25 after ipilimumab treatment (Fig. 6e and f). In conclusion, a weak restoration of the CD28/B7 axis after CTLA-4 blockade is accompanied by a reduced IL-2 impregnation during the course of ipilimumab leading to low accumulation of memory and ICOS[+] T cell in MM patients with high serum butyrate levels.

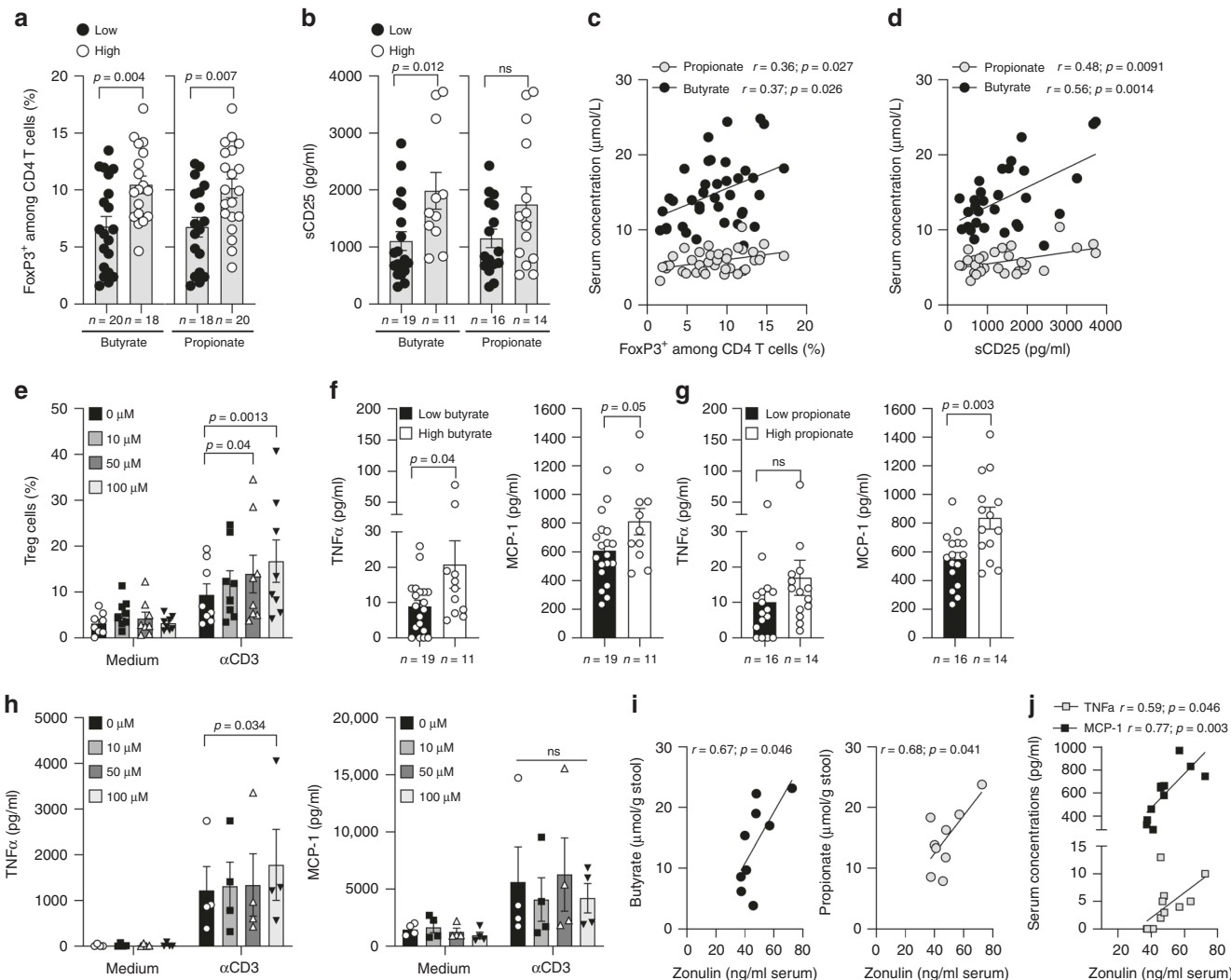

**Fig. 5 Baseline immunological markers according to serum butyrate or propionate levels. a, b** Mean (SEM) percentage of blood Treg (FoxP3+ among CD4+ T cells; **a**) and mean (SEM) sCD25 (**b**) according to median concentration of serum butyrate (left panel) and propionate (right panel) concentrations before ipilimumab treatment (V1). **c, d** Pearson correlations between butyrate and propionate quantification in serum and percentages of peripheral Treg (**c**; $n = 38$) and sCD25 (**d**; $n = 30$) before ipilimumab treatment (V1). **e** Mean (SEM) percentage of Treg in human PBMC ($n = 8$) after 4 days of culture with or without anti-CD3 and escalating concentrations of butyrate. **f** Mean (SEM) concentrations of systemic TNFa (left panel) and MCP-1 (right panel) according to median concentration of serum butyrate. **g** Mean (SEM) of systemic TNFa (left panel) and MCP-1 (right panel) according to median concentration of serum propionate. **h** Mean (SEM) TNFa (left panel) and MCP-1 (right panel) secretions by human PBMC ($n = 4$) stimulated during 4 days with or without anti-CD3 and escalating concentrations of butyrate. **i** Spearman correlation between butyrate (left panel) and propionate (right panel) quantifications in the stool and zonulin in the serum from patients ($n = 9$, where both data were available) before ipilimumab treatment (V1). **j** Spearman correlation between zonulin and serum concentrations of TNFa (light gray squares) and MCP-1 (black squares) ($n = 12$, where both data were available). Each dot represents one patient; $n$ (number of patients) are indicated on each graph otherwise stated in the legend. P-values are indicated on each graph; Two-tailed unpaired $t$ test (**a**, **b**, **f**, **g**) and two-way RM ANOVA followed by Tukey's tests for multiple comparisons was used (**e** and **h**). ns, not significant.

Altogether, this study provides insights on how the composition of the gut-microbiota might influence clinical responses to ipilimumab at distant sites through gut-microbial metabolites recovered in the periphery.

## Discussion

This work showed that systemic microbial SCFA modulate anti-CTLA-4-induced immune responses and its antitumor efficacy. SCFA, as butyrate and propionate have important metabolic functions and are crucial for intestinal health. In our mice models, serum butyrate was artificially increased without any modification of gut microbiota as demonstrated previously[19]. Modulating only this parameter in mice limited anti-CTLA-4 blockade efficacy. Butyrate has been described to modulate

immunity and inflammation[18]. Note that within SCFA, butyrate is always associated with immune modulatory capacities (as for propionate) mainly through induction of Treg cells[21]. In mice, CTLA-4 blockade induces an up-regulation of CD80 and CD86 co-stimulatory molecules, which was abolished by oral administration of sodium butyrate. CTLA-4 acts through cell-intrinsic[34] and cell-extrinsic pathways[35]. Cell-intrinsic pathway involves the cytoplasmic domain of CTLA-4 resulting in negative signal transduction machinery in T cells. The cell extrinsic pathway evidenced in in vivo models[36] and in vitro with Human cells[35], consists of capture and removal of CD86 from monocyte-derived DC by the CTLA-4 molecule expressed by T cells by a process of trans-endocytosis[35]. These phenomena resulted in decreased expression of CD86 on DC and impairment of co-stimulation via

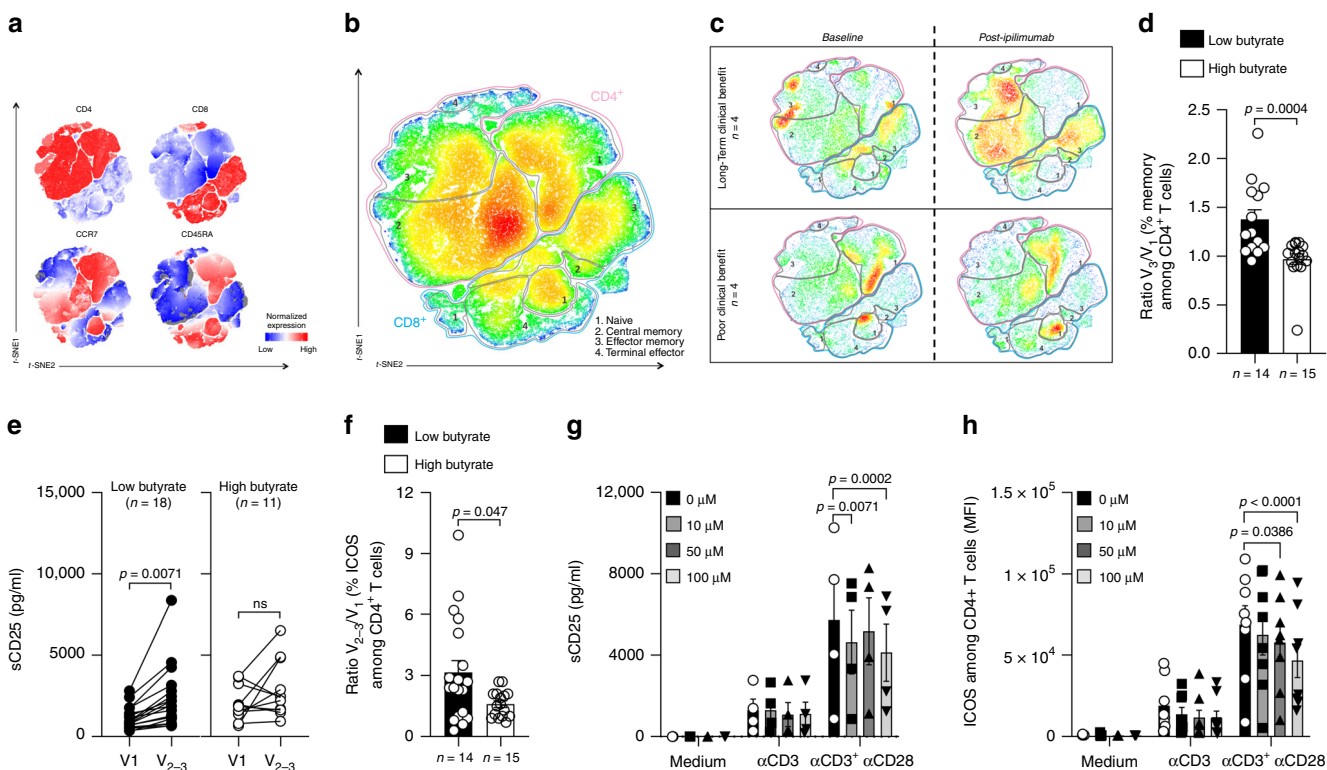

**Fig. 6 Immune markers during the course of ipilimumab. a** $t$-SNE map of markers included in the T cell 1 panel in all patients ($n = 8$). **b** $t$-SNE map with the gating of T cell (CD3$^+$) populations: CD4$^+$ T cells (red outline), CD8$^+$ T cells (blue outline), black outlines (regions **1**: Naive CD4$^+$ or CD8$^+$ T cells; regions **2**: Central memory CD4$^+$ or CD8$^+$ T cells; region **3**: Effector memory CD4$^+$ or CD8$^+$ T cells; regions **4**: Terminally effector CD4+ or CD8+ T cells). **c** $t$-SNE map at baseline ($V_1$) and post-ipilimumab ($V_{2-3}$) in patients with long-term clinical benefit (upper panel) and poor clinical benefit (lower panel). **d** Post-ipilimumab ($V_3$) to baseline ($V_1$) ratio (Ratio $V_3/V_1$) of memory CD4$^+$ T cells (% among CD4$^+$ T cells) according median concentration of serum butyrate at baseline. **e** Serum concentration of sCD25 was monitored before ipilimumab treatment ($V_1$) and after one or two injections of ipilimumab ($V_{2-3}$) according to butyrate concentration. **f** As in (**d**) but for ICOS$^+$CD4$^+$ T cells (% ICOS$^+$ among CD4$^+$ T cells;). **g** sCD25 quantification in the supernatants of PBMC after 4 days of culture in medium $+/-$ plastic-bound anti-CD3$\varepsilon$ $+/-$ anti-CD28 agonist with escalating dose of butyrate (n = 4). **h** As in (**g**) but ICOS mean fluorescence intensity (MFI) among CD4$^+$T cells ($n = 8$). Histograms represent mean $+/-$ sem (**d**, **f**, **g** and **h**). Each dot represents one patient. $n$ are indicated on graphs; $p$-values are indicated on each graph; Wilcoxon (**e**) and Mann–Whitney (**d**, **f**) and two-way RM ANOVA followed by Tukey's tests for multiple comparisons was used (**g**, **h**) tests were used. Exact $p$-values are indicated on each graph otherwise *$p < 0.05$; **$p < 0.01$; ***$p < 0.001$; ns, not significant.

CD28. This mechanism is specific of CTLA-4/CD28 ligands and operates in an antigen-dependent manner in vivo[35]. Butyrate was shown to prevent DC maturation in vitro[23,24]. In these studies, butyrate was shown to act directly on DC maturation through the modulation of expression of several genes indicating a potent anti-inflammatory activity. In our settings, adding butyrate in mice treated with IgG2b did not induce modification of CD86/CD80 expression (in tumor-bearing mice). These data suggest that blocking the interaction between CTLA-4 and CD86/CD80 is mandatory to observe butyrate-induced regulation. CTLA-4 being blocked we can rule-out the possibility that butyrate could augment trans-endocytosis by T cells; butyrate might rather prevent DC maturation through a direct mechanism as suggested by in vitro studies[23,24]. In the MCA101$_{OVA}$ tumor model, blocking CTLA-4 could increase antigen-specific T cell responses in the tDLN; addition of butyrate could limit these responses. Several studies observed that sodium butyrate may induce T cell anergy[37] or reduce T cell priming capability[38]. In these studies butyrate did not act directly on T cells but limited the capacity of DC to stimulate T cells. In our work, since butyrate abolished anti-CTLA-4-induced CD80/CD86 up-regulation in vivo, restoration of the B7/CD28 axis might remain insufficient to promote the expansion of memory T cells and antigen-specific T lymphocytes.

In line with mice data where butyrate was shown to regulate the size and function of the colonic Treg pool by up-regulating IL-10 [19,39] as well as accelerating extrathymic peripheral differentiation of Treg cells[20,22], an immune regulatory profile was associated with high level of serum butyrate before introduction of ipilimumab in patients. As in mice, longitudinal immune monitoring in patients with MM, provided evidence that high butyrate was associated with less accumulation of memory and less ICOS induction on T cells in hosts with cancer and treated with anti-CTLA-4. No or weak increase of sCD25 in patients with high butyrate levels during ipilimumab treatment could be evidenced. Finally in vitro stimulation of Human PBMC demonstrated that butyrate could counter CD28 signaling pathway. Altogether, these data suggested a poor restoration of CD28 signaling after CTLA-4 blockade in patients with high levels of blood butyrate. Since butyrate was able to inhibit ipilimumab-induced DC maturation and CD28 signaling pathway, restoration of CD80/CD86 through the use of TLR agonists or other DC maturating agents might not be sufficient to counter the deleterious effect of these metabolites. In line with this, CpG agonists could not overcome the influence of butyrate in mice treated with anti-CTLA-4 (Supplementary Fig. 18). In consequence the specific modulation of butyrate/propionate signaling pathway might be mandatory to restore anti-CTLA-4 efficacy in

patients with high systemic levels of SCFA. SCFA are known to act also through specific receptors (GPR41, GPR43 and GPR109A) expressed on immune cells, the use of GPR specific inhibitors might be a better option. However this has to be demonstrated and will be part of future research.

In this study patients with high butyrate concentration had higher concentration of inflammatory proteins in the serum. This data seemed contradictory with previous published works. However and as demonstrated, at μM concentrations no inhibition of inflammatory proteins production could be observed. This is in lines with previous studies where dose escalation were used; a minimal concentration of 3 mM (3000 μM) butyrate could induce a reproducible and significant diminution of inflammatory proteins such as TNFα and IL-6[30].

The level of faecal SCFA is influenced by several factors such as the gut microbiota composition[40], the transit time and the available substrata[41] and the gut integrity. Firmicutes and F. prausnitzii are well known for its butyrate producing capabilities. Faecal concentrations of SCFA measured in the stools (and serum) depend on both the production by microbiota and their consumption by bacteria (cross-feeding) and colonocytes[13], playing a crucial role in proliferation, differentiation of epithelial cells and gut permeability[17]. Systemic concentrations of SCFA depend on both the production and absorption rates in the gut, acetate being absorbed in the highest amount in the liver and reaches the highest peripheral concentrations (19–160 μmol/l) compared to propionate (1–13 μmol/l) and butyrate (1–12 μmol/l)[17]. Peripheral concentrations measured in some patients with MM were clearly above these previously published values. Unfortunately, very few studies assessed SCFA in serum. Most data accumulated on SCFA production rely on metabolites production by bacteria in culture[42], or metagenomics analysis and detection of pathways used by bacteria for SCFA formation[43,44]. Considering that these data do not measure the real concentrations, it appears difficult to conclude on the balance between production and consumption. In the current study, zonulin was correlated with stools' concentrations of propionate and butyrate suggesting that higher concentrations of butyrate and propionate remaining in the stool of patients were related to an increased intestinal permeability. This result suggests that high concentrations of SCFA in stools of patients might reflect a poor consumption by colonocytes and/or bacteria increasing gut permeability. Since increased intestinal permeability is also related to inflammation, it was not surprising to observe correlations between zonulin and systemic inflammatory proteins.

It has been shown that gut microbiota impacts on the level of expression of enzymes involved in SCFA metabolism. Particularly butyrate-producing microbes increase the level of acyl CoA dehydrogenase involved in β-oxidation[13] and colonic mitochondrial 3-hydroxy 3-methyl glutaryl CoA (mHMGCoA) synthase responsible for the biosynthesis of ketone bodies from butyrate[45]. On the contrary, absence of butyrate-producing microbes led to low levels of expression of these enzymes and low metabolism of butyrate[13,45]. Recently, a negative correlation between the abundance of fecal Firmicutes and fecal SCFA levels was observed in healthy volunteers[46]. Moreover in mice, a high-amylose maize starch butyrylated (HAMSB) diet seems to be associated with expansion of Bacteroides species as well as higher concentrations of systemic butyrate leading to resolution of type I diabetes in mice models through induction of immunosuppression[22]. In line with these studies, we brought evidence in patients that microbiota enriched with Faecalibacterium is negatively correlated with Bacteroides genera and associated with lower systemic butyrate levels.

Our study has limitations; in patients due to low number of patients we could not realize multivariate analysis including major clinical and biological parameters already known to be associated with PFS and OS. It pushed us to validate our first observation in an independent cohort of patients with MM and treated with ipilimumab. Due to the discovery nature of our work, one's cannot exclude that it won't be validated in larger cohorts of patients. Nowadays, ipilimumab is not used as a standalone treatment in many patients, thus first we will have to validate if SCFA are also associated to clinical outcomes in patients treated with anti-PD-1/PD-L1 since Faecalibacterium was also associated to clinical outcomes in MM patients[7]. In mice, we used butyrate supplementation in water; however the use of HAMSB diet as described elsewhere[22] could have been a more direct manner to incriminate metabolites produced by microbiota and response to anti-CTLA-4 blocking mAbs. Finally, we described immune parameters in mice and patients associated with high systemic levels of butyrate but we did not identified a specific pathway affected by butyrate that could be blocked in order to restore anti-tumor activity of anti-CTLA-4 blockade in hosts with high systemic butyrate.

In conclusion, systemic butyrate (and propionate) appeared to limit antitumor activity of anti-CTLA-4. This study provides a vision on the relationship between the composition of the gut-microbiota and the clinical response to ipilimumab via microbiota-derived metabolites. Finally, the identification of a specific pathway responsible for butyrate/propionate-associated immune-modulatory effects should represent an exciting research area that could allow the discovery of targets to improve anti-CTLA-4 response rate in patients with MM.

## Methods

**Mice and cell lines**. C57Bl/6 and BALB/c mice were purchased from Harlan Laboratories (Gannat, France) between 8 and 14 weeks of age. Mouse colon carcinoma MC38 and CT26 and fibrosarcoma MCA101$_{OVA}$ cell lines were cultured in complete medium (RPMI 1640 medium supplemented with 1 mM sodium/pyruvate, 1 mM of non-essential amino acids (NEAA), 100 IU/mL penicillin/streptomycin, 2 mM L-Glutamine and 10% fetal bovin serum (FBS, Life Technologies)). For the MCA101$_{OVA}$ cell line RPMI complete medium was supplemented with 1 mg/ml hygromycin B (Thermofisher).

**Choice of syngeneic tumor models in mice**. CT26, MC38 were demonstrated as partial responder models (CT26 > MC38) to immune checkpoints thus providing good models for combination treatment effect (here Butyrate + anti-CTLA-4) (https://www.crownbio.com/wp/murine-model-guide-immuno-oncology;[47]). Note that melanoma preclinical models in mice, such as B16 models, are not responding to immunotherapy due to low MHC molecules expression, B16 appeared to be immunologically "silent" with low expression of immune cell type–specific genes thus did not appeared as a good preclinical model to test immune-checkpoints in mice[47]. The fibrosarcoma MCA$_{OVA}$ was used for its expression of OVA antigen thus allowing the "easy" monitoring of H-2Kb:OVAp (peptide sequence SIIN-FEKL)-specific T cell responses. Moreover in our hands MCA$_{OVA}$ was also a partial responder model after anti-CTLA-4 treatment.

**Antibodies and sodium butyrate**. Anti-CTLA-4 (clone 9D9) and its isotype control IgG2b (clone MPC-11) were both from BioXcell. All of these reagents were diluted in sterile Phosphate Buffer Saline (PBS). Sodium butyrate (CH$_3$CH$_2$CH$_2$COONa; ≥ 98.5%) was purchased from Sigma-Aldrich.

**Tumor inoculation in mice and antitumor treatments**. For all mouse experiments in the butyrate group, mice were treated 2 weeks before tumor inoculation and all along the experiment with sterile sodium butyrate in the drinking water at the concentrations of 100 mM (pH-matched) and changed every week.

Five hundred thousand of CT26 tumor cells were inoculated subcutaneously (s.c) in the right flank of Balb/c mice. Five hundred thousand MCA101$_{OVA}$ cells were inoculated s.c. in the right flank of C57Bl/6 mice. Five hundred thousand of MC38 cells were inoculated s.c. in the right flank of C57Bl/6 mice. Mice received intraperitoneally (i.p.) injections of 100 μg of anti-CTLA-4 (clone 9D9) or its isotype control control (IgG2b) at D7, D10 and D13. In the experiment with CpG agonists, s.c injections near to the tumor site were done with 30 μg of CpG (Invivogen) or ODN control (Invivogen) at D7, post tumor inoculation. Tumor growth was followed-up three times a week by measuring the length and width with a caliper and was calculated as follows: size (mm$^2$) = length x width. Mice were euthanized when the tumor size was ≥300 mm$^2$ or boundary points were

reached according to the French and European laws and regulations for the use of mice for scientific purposes. To analyze draining lymph nodes immune population, mice were sacrificed 14 days post tumor inoculation and draining lymph nodes were removed. Single cells were obtained after mechanical disruption. Serum samples were collected at the same time and were stored at −80 °C until further analysis of SCFA dosages. For longitudinal blood immune monitoring in whole blood of mice, tumor bearing mice (CT26 model) were bleed before the first anti-CTLA-4 injection (D6 post tumor injection) and 2 days after the third injection of anti-CTLA-4 (D15) in both PH-matched water and sodium butyrate drinking groups.

**Ex vivo experiments in mice.** Ex vivo restimulation of T cells from tDLN: mice were sacrificed 21 days after MCA101$_{OVA}$ inoculation, tDLN were removed and after mechanic dissociation 2.10$^5$ mononuclear cells were stimulated with escalating doses of Ovalbumin CD8 epitope 257-264 (SIINFEKL; Genecust) at 0, 0.1, and 1 μg/ml. The supernatants were assayed at 48 h by ELISA for mouse IFN-γ (Becton Dickinson) according to manufacturer's instructions.

**Patients.** Eighty-five patients were prospectively enrolled in the French study (GOLD cohort) at Gustave Roussy Campus (Villejuif) between March 2013 and June 2015. Among them, 28 patients did never receive ipilimumab, four patients received only a single infusion of ipilimumab and died soon thereafter due to their disease and three patients were lost for the follow-up. They were therefore excluded from further analyses. The remaining fifty patients with MM (48/50) or metastatic prostate carcinoma (2/50) were followed-up for at least 6 months. Fifty-Italian patients with MM treated with ipilimumab were prospectively enrolled at Instituto Nazionale Tumori Fondazione G. Pascale (Napoli) between July 2014 and March 2016 respectively. Among them, five patients received only a single infusion of ipilimumab and were therefore excluded from further analyses. The remaining forty-five patients with MM were followed-up for at least 6 months. Patients were informed of the study and consented to participate. French patients had a pre-specified clinical workup; feces and blood were collected at baseline (V$_1$), prior to each ipilimumab infusion (V$_2$, V$_3$, V$_4$). Patient's characteristics are described in Supplementary Table 1. In the GOLD cohort, all blood, serum and fecal samples used in this study are listed in Supplementary Table 2. In the Italian cohort only serum were collected at baseline for all patients. Ipilimumab was administered intravenously every 3 weeks, at a dose of 3 or 10 mg/kg and could be continued after V$_4$, at a maintenance dose of one infusion every 12 weeks, in patients whose disease was controlled (response or stable disease). Peripheral blood mononuclear cells (PBMC) were isolated by Ficoll density gradient, frozen in Hyclone media (Thermo Scientific) and stored at −196 °C for further in vitro experiments.

**In vitro experiments in human PBMC.** PBMC were isolated by Ficoll density gradient from four healthy donors and four MM patients before ipilimumab treatment and cryopreserved in liquid nitrogen. $1.5 × 10^5$ thawed PBMC were stimulated, into 96-well flat bottom Maxisorp (Thermofisher, USA) with escalating concentrations of sodium butyrate (0, 10, 50 and 100 μM) in complete medium alone, with pre-coated anti-CD3ε alone (0.5 μg/ml; BD Pharmingen) or anti-CD3ε + soluble anti-CD28 (0.1 μg/ml; BD Pharmingen) during 4 days at 37 °C, 5% CO$_2$. At day 4, PBMC were collected to evaluate the percentage of Treg and ICOS expression among CD4$^+$ T cells using flow cytometry analysis. At day 4, supernatants were collected and cryopreserved at −80 °C until the day of quantification. sCD25, TNFα, MCP-1 and IL-10 were quantified using Bio-Plex® MAGPIXTM Multiplex assay (Millipore) according to manufacturer's instructions and plates were analyzed on a MAGPIX (MAGPX13337701, Luminex). Determination of cytokines concentrations was assessed with XPONENT software solutions.

**Soluble immune markers in patients.** All serum samples were stored at −80 °C until analysis of soluble immune markers. Soluble CD25 (sCD25) was quantified using Bio-Plex® MAGPIXTM Multiplex assay HSCRMAG-32K-01 (Millipore, Darmstadt, Germany); MCP-1 and TNFα Bio-Plex® MAGPIXTM Multiplex assay HCYTOMAG-60K-05 (Millipore) and soluble CTLA-4 (CTLA-4 (Soluble) Human Pre-coated 96 well plate ELISA Kit; Invitrogen) according to manufacturer's instructions. Zonulin concentrations in serum were determined using an ELISA kit (Immundiagnostik AG, Bensheim, Germany) according to manufacturer's instructions and concentrations were determined with an ELISA plate reader at 450 nm against 620 nm as reference.

**Flow cytometry analysis.** In mouse experiments, before staining, Fc receptors were blocked for 15 min at 4 °C using anti-CD16/32 functional grade purified antibodies (eBioscience) at 0.5 μg/10$^6$ cells. For staining on PBMC, cells were incubated for 30 min at 4 °C with antibodies for cell surface staining. For surface staining on whole blood, after Fc receptor blocking and surface staining, red cells were lysed with a mix of Versalyse lysing solution + Fixative solution (1000:25, Beckman Coulter) during 15 min at room temperature. Cells were then washed two times with PBS 1×. For FoxP3 staining, cells were fixed and permeabilized after cell surface staining according to the FoxP3 kit protocol (eBioscience). Antibody characteristics are described in Supplementary Table 4. Samples were acquired on a Gallios Cytometer (Beckman Coulter) and fluorescence was analyzed using Kaluza

software (Beckman Coulter). DCs were defined among CD45 positive cells by lineage negative (CD3/CD19/NKp46) as classe II MHC (IAb$^+$) and CD11c-positive cells. In humans, immune monitoring on whole blood and on PBMC was performed as described previously[6].

**Short-chain fatty acid dosages.** Quantitation of SCFA including acetate, butyrate and propionate from serum and fecal samples was performed using LC-MS/MS method adapted from Lu et al.[48]. Hydrochloric acid was added to serum samples (25 μL) and to faeces samples (20 mg) prior hydrazinquinolin (HQ) derivatization and before protein precipitation using acetonitrile. Organic phase (25 μL) was diluted in the mobile phase (475 μL) prior to injection for mass spectrometry analysis. SCFA free materials were used. The analyses were performed on a Acquity® UPLC system (Waters, Milford, MA, USA) that included a binary solvent manager, a cooled auto-sampler and a column oven coupled with a Xevo® TQ-MS (Waters, Milford, MA, USA). A Waters BEH-C18 (2.1 × 100 mm, 1.7 μm) from Waters (Milford, MA, USA) was used. The column was maintained at 40 °C. The Waters Xevo® TQ-MS mass spectrometer was interfaced to the Acquity® UPLC system via an electrospray ionization (ESI) source used in positive mode. The system control and data acquisition were performed using MassLynx® software (v4.1, Waters, Milford, MA, USA). This method was adequately validated with good precision and recovery values according to ICH and EMA guidelines (International Conference on Harmonization. Validation of Analytical Procedures: Text and Methodology (Q2R1) revised in November 2005 and Guideline on Bioanalytical Method Validation 21 Jul 2011). Acetate was measured in serum from 2.5 to 1,000 μM. Butyrate and propionate were measured in serum from 0.125 to 50 μM. Acetate was measured in faeces from 2.25 to 150 μmol/g. Butyrate and propionate were measured in faeces from 0.0225 to 7.5 μmol/g.

**Quantification of bacteria by quantitative PCR.** In the French cohort, for each patient, 200 mg of feces were used. DNA extraction included a first step of mechanical lysis: samples were diluted in 400 μL of ASL buffer in presence of 0.5 mm glass beads. Lysis was performed at 30 Hz during 30 min with the help of a TissueLyser (Qiagen). Next, samples were centrifuged at 10000 g 12000 rpm during 4 min and 200 μL of the supernatant was further extracted using the QIAamp DNA Stool Mini Kit (Qiagen), according to the manufacturer's instructions. Extractions series were realized during five consecutive days (5 series), for each series, stool sample from the same patient (patient #30) was extracted, this sample confirmed the reproducibility of analyzes. All DNA samples were quantified using Qubit™ 3 Fluorometer and Qubit™ dsDNA BR Assay Kit (ThermoFisher) according to manufacturer instructions. For each sample dilutions were realized to test 10 ng of DNA per qPCR. All qPCR were performed using the TaqMan™ Gene Expression Master Mix kit (LifeTechnologies) in 25 μl final volume (4 μl of genomic DNA added per tube, 300 nM each primer and 100 to 200 nM of each probe, 10$^3$ copies of an internal amplification control (IAC). All samples were analyzed in duplicate. All negative, very weak or discordant results were verified by another series of qPCR (in duplicate). All primers and probes used in this study (all Bacteria, *Faecalibacterium prausnitzii*, *Escherichia coli*, *Bacteroides fragilis* and Internal Amplification Control (IAC)) are presented in Supplementary Table 3 and were described in supplementary bibliography[49–52]. Standard curves were performed using genomic DNA preparations: *F. prausnitzii* A2-165 (from 0.4 ng/Q-PCR (~1.2 × 10$^5$ 16S targets) to 0.04 pg/Q-PCR (~120 16S targets); *E. coli* CIP 7621 (~1.2 × 10$^6$ 16S targets) to 0.4 pg/Q-PCR (~120 16S targets)) and *B. fragilis* CIP 7716 (from 1.2 ng/Q-PCR (~1.2 × 10$^6$ 16S targets) to 0.12 pg/Q-PCR (~120 16S targets). All primers, probes and IAC'sDNA were purchased from Sigma Life Science (Woolands, TX, USA). Amplifications were carried out using the following ramping profile: 1 cycle at 50 °C for 1 min, 1 cycle at 95 °C for 10 min, followed by 40 cycles of 95 °C for 30 s, 60 °C for 1 min. Excepted for the *B. fragilis* Q-PCR, the annealing was at 58 °C for 1 min. Q-PCR data were normalized to the expression levels of total bacteria measured by the all bacterial Q-PCR with *E. coli* standard curve. Final analyses according to serum butyrate levels were realized on data from 33 patients for whom both serum butyrate concentrations and stool were available at V$_1$.

**Ion S5 torrent sequencing analysis.** In the same French cohort, 5 ng of stool DNA genomic were amplified with Ion 16S metagenomics Kit (ThermoFisher Scientific) (part Number A26216) using 20 amplification cycles according to the manufacturer's instructions (Manual MAN0010799). As recommended, *E. coli* DNA control (2 ng/PCR) was used in parallel. All library preparations were quantified with Universal Quantitation Kit (Part number A26217). Templating and enrichment for Ion Torrent sequencing were performed by using Ion Torrent CHEF (ThermoFisher Scientific) according to the manufacturer's instructions (Manual MAN0010846). To optimize the 16S sequences read number, 11differents barcoded samples were used for sequencing in 530 Ship. Sequencing was performed on an Ion S5 sequencer (ThermoFisher Scientific), using 400-bp sequencing kits with 850 run flows. Base calling and run demultiplexing were performed by using Torrent_Suite Software, version 4.0.2, with default parameters for the General sequencing application. All sequencing data were exported in Ion Reporter Software, version 5.6 with Metagenomic workflow which detects the bacterial diversity from a metagenomics sample from Ion semiconductor reads from the Ion 16S Metagenomics Kit. The experimentally generated sequence data were compared to

two libraries of representative reference sequences download from (1) Curated MicroSEQ (R) 16S reference Library (V2013.1) and (2) Curated Grengenes data base (V13.5).

**Statistical analyses**. Statistical analyses were performed using the Prism 7.0 (GraphPad) Software. All tests used are systematically indicated on graphs. The paired *T* test or Wilcoxon matched-pairs signed rank tests could be used to compare paired groups, while the unpaired T test with Welch's correction or Mann-Withney tests could be performed to compare two unmatched groups. One-way ANOVA was used when more than two groups were compared. Two-way ANOVA multiple comparisons was used when two or more grouping variables were analyzed. Two-way RM ANOVA multiple comparisons was used when two or more grouping matched variables were analyzed. Correlations were assessed using the Spearman (don't assume that data are sampled from Gaussian distribution) or Pearson (assume that data are sampled from Gaussian distribution) correlation coefficient. Contingency could be determined using the Fisher's exact test. Unsupervised cell population analysis was performed using t-distributed stochastic neighbor embedding (t-SNE) algorithm using the online R software (version 3.5.0, cytofkit package). The "T cell 1" flow cytometry panel (CD95/CCR7/HLA-DR/ CD25/ICOS/ CD45RA/CD3/CD127/CD4/CD8); as previously described[53] was used, after setting the compensation matrix cells were manually gated using Kaluza Analysis (version 2.0) on size/granularity discrimination and then on the expression of CD3 and logicle transformation was applied to extracted data for all samples. t-SNE analyses was achieved on 8500 cells CD3$^+$ for each sample, using T cell 1 panel markers for T cell differentiation status (CCR7/CD45RA/CD4/CD8). Supervised analysis was performed using Kaluza Flow Cytometry Software (Beckman Coulter). OS and progression-free survival (PFS) according to SCFA dosages were estimated using the Kaplan-Meier method and compared using the log-rank test. OS and PFS analyses were carried out using PROC lifetest of SAS software, Version 9.4 (SAS Institute, Inc., Cary, NC). All tests are two-tailed and were considered significant at a *p* value < 0.05. Exact *p* values were annotated on graphs otherwise ns or p > 0.05 was annotated for non-significant data.

**Human and mouse studies approval**. Immune monitoring in the blood and the serum of patients were approved by the Kremlin Bicêtre Hospital Ethics Committee (SC12-018; ID-RCB-2012-A01496-37) for the French cohort and by the internal ethics board of the Instituto Nazionale Tumori IRCCS Fondazione "G. Pascale" of Napoli (number of registry 33/17) for the Italian cohort. The Declaration of Helsinki protocols were followed for both cohorts. Patients provided their written informed consent to participate in these studies prior to inclusion in the study. Blood samples of healthy volunteers (HV) were obtained after inform consent signature and were collected according to a procedure validated by the CCPSL UNT-N°12/EFS/079. All animal experiments were carried out in accordance with French and European laws and regulations and approved by the French Animal Experimentation Ethics Committee n°26 (02004.02).

**Reporting summary**. Further information on research design is available in the Nature Research Reporting Summary linked to this article.

## Data availability
Data such as human demographic data are not publicly available in accordance with the General Data Protection Regulation since it could compromise research participant privacy/consent. All other data that support the findings of this study are available from the corresponding author [N.C.] upon reasonable request. We have submitted all source data of this manuscript to Springer Nature Research Data Support, the submission number is: RDS-SPRN-00421[54].

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

## Acknowledgements

The authors thank Dr. Patrick Gonin from the PlatForm for Preclinical Evaluation (PFEP). Fundings: This study was funded by Gustave Roussy Cancer Campus, Fondation Gustave Roussy, the Institut national de la santé et de la recherche médicale (INSERM), the Centre national de la recherche scientifique (CNRS), SIRIC SOCRATE (INCa DGOS INSERM 6043), SIRIC SOCRATE 2.0 (INCa-DGOS-INSERM_12551), MMO program: ANR-10IBHU-0001); Direction General de l'Offre de Soins (DGOS; TRANSLA 12-174); Institut National du Cancer (INCa; 2012-062 N_ Canceropole: 2012-1-RT-14-IGR-01). Dr. Clélia Coutzac was supported by fellowships from Fondation pour la Recherche Medicale (FRM) from 2015 to 2016.

## Author contributions

C.C., J.M.J., L.C., J.S., L.N.B., P.L.W., A.V., M.N., M.D., L.B., J.G., E.S. (E. Soularue), were involved in acquisition, analysis and/or interpretation of immunological data. A.P. and A.S. were involved in validation, acquisition and interpretation of SCFA assay. C.C., P.S., L.L., and P.L.W. were involved in validation, acquisition and interpretation of PCR and metagenomic analysis in stools from patients. V.A., D.M., E.S. (E. Simeone), C.M., P.A. A., F.C., and C.R. were involved in collection, analysis and interpretation of clinical data. P.L., F.C., C.R., P.A.A., L.C., and D.M. were involved in critical review of the manuscript. N.C. and C.C. were involved in the study concept and design, analysis and interpretation of data, drafting of the manuscript. N.C. obtained funding and supervised the entire study.

## Competing interests

N.C. reports research grants from Cytune Pharma, GSK, Sanofi, has participated in advisory boards for AstraZeneca and has been a speaker for Sanofi and AstraZeneca. C.R. has participated in advisory boards for Roche, GSK, Merck, Novartis, Amgen, BMS and Novartis. Franck Carbonnel received boards or lecture fees from enterome, Amgen, Astra, MSD, BMS, Janssen, Pfizer, Abbvie, Mayoly Spindler, Takeda, Pileje, Roche. E. Soularue has participated in advisory board for Novartis. L.L. has participated in advisory boards for Roche, Astrazeneca, BMS, Genomic Health, Illumina, Qiagen, Novartis Thermofisher and has been a speaker for Amgen, AstraZeneca, Dyn, Vela diagnostics, Luye Pharma, and Roche. P.A.A. has/had a consultant/advisory role for Bristol Myers-Squibb, Roche-Genentech, Merck Sharp & Dohme, Array, Novartis, Amgen, Merck Serono, Pierre Fabre, Incyte, NewLink Genetics, Genmab, Medimmune, AstraZeneca, Syndax, SunPharma, Sanofi, Idera. He also received Research funds from Bristol Myers-Squibb, Roche-Genentech, Array. The other authors have no conflicts of interest to declare.
