## [Peer Review File · Nature Communications]

Reviewers' comments:

Reviewer #1 (Remarks to the Author):

This is a very interesting and novel paper that explores the functional role of the microbiome in the response to cancer immunotherapy. While previous studies have linked microbiome composition to the degree of response to immunotherapy and have implicated direct effects of bacteria on immune cells, this paper introduces the novel concept that metabolites (SCFAs) produced by bacteria determine the likelihood of response to immunotherapy. The major claim of the paper is that microbial produced SCFAs might favor an immune tolerance profile that limits anti-CTLA-4 activity. This conclusion is well supported by the data. Strengths include the vast scope of the study and the use of both animal and human studies. Limitations are appropriately mentioned in the discussion.

Major points

1. Many critical data are (e.g. fig 5a) presented by dividing the study population in two arbitrary groups: Low and High. Considering that the depicted variables are continuous variables (e.g. Treg number), the analysis should be completed by using correlations or other statistical methods suitable for continuous variables.

2. Lines 299-301 and figure 5b. It is stated that positive association between serum concentrations of inflammatory proteins (MCP-1, TNF, sCD25) and serum SCFAs concentrations were noted. Considering that SCFAs increase the number of Tregs and exert anti-inflammatory activities, this is opposite than expected. Please explain.

Lines 404-406. It is stated that :the data suggest a poor restoration of CD28 signaling after CTLA-4 blockade in patients with high levels of blood butyrate". This is an important issue. The effects of butyrate on CD28 signaling could be studied more directly.

Reviewer #2 (Remarks to the Author):

The paper entitled "Systemic gut microbial metabolites limit the anti-tumor effect of CTLA-4 blockade in hosts with cancer" discuss an exciting topic, using human and murine data which could be of interest for the journal. However, additional experimental work is necessary. The current data set may not adequately supports the main claim of the manuscript. Some of the data lack of strength to fit the requirement for a publication in Nature Communication.

- The authors validate their previous findings with a positive association of *Faecalibacterium prausnitzii* and clinical benefit under ipilimumab therapy. However, their previous finding of negative association of *Bacteroides* and efficacy of ipilimumab are less convincing and partially invalidated.
- Due to the authors previous work describing a protective role of *F. prausnitzii* against colitis mediated by ipilimumab and the role of *F. prausnitzii* to protect against IBD, we expected the author will clarify the status of colitis of their cohort.
- The use of antibiotics and the toxicity related to ipilimumab in the patient cohort should be added.
- A lot of inconsistencies have been found all along the manuscript in the patient numbers and description and need to be addressed (some prostate cancer patients, the number of patients depicted in figure 1 do not match the text and the table of patients' description and the data from the dosage of SCFA in the same compartment at the same time point are different across figures).
- Several data highlighted in the paper are not statistically significant (figure 1d, 2b) and the second human cohort do not validate the French one for butyrate. The role of propionate is suggested but not demonstrated as well as butyrate, other than the increase of Tregs in PBMC at

baseline.

- The focus on *B.fragilis* and *E.Coli* do not seem to be relevant in the context of SCFA. Other genera were found significant between LTB and PB patients and 1- should be represented 2-could be used in relevant comparison with *F. Prausnitzii* (other butyrate-producing bacteria belonging to the genera of *Roseburia*).
- The data associating high *F. prausnitzii* abundance and Low Butyrate (in the blood but correlated to the fecal concentration) should be investigated deeper. Due to the well-known butyrate producing capabilities of *F. Prausnitzii*, this is actually suggesting another key player missing in the current demonstration.
- How the authors explain the increase of pro-inflammatory proteins in the High butyrate group?
- Two out of the three murine model used are colon cancer lines. The phenotype of the third model, the MCA-OVA sarcoma, used for mechanistic demonstration, is less convincing, and the mechanistic data are sometimes opposite to the human data. Due to the well described role of butyrate on normal and malignant colonocytes, the use of melanoma cell lines, other than B16, such as BRAF/PTEN, RET could represent a better option.
- Several data do not fit the flow of the demonstration and do not match human data (role of butyrate in the antigen-specific production of IFN γ and in the accumulation of memory T cells in tDLN, figure 4f and supp 10)
- Too many indirect evidences are made between clinical benefit, *F. prausnitzii*, Butyrate and anti-cancer immunity.

Specific comments are as follows:

Major comments:

Title: The author should specify the type of "gut microbial metabolites" they refer to in their study (buturate and propionate or SCFAs)

Abstract: Butyrate does not limit the accumulation of memory T cells in mice with cancer (cf comments on Figure Supp 10)

Results text:

Line 108: Are the 39 French patients included in this study also part of previous study from the same group published in 2017, in *Annals of Oncology* or is this an independent cohort? The cohort (GOLD) and period of collection (March 2013-December 2014) are the same in the two studies. The authors mentioned they studied 39 melanoma patients, but in the clinical table there are 38 melanoma patients and 2 prostate cancer patients.

The toxicity status of these 39 patients should be clarified, especially due to the properties of SCFAs and *Feacalibacterium prausnitzii* to protect against IBD.

Significant changes in human fecal *F. prausnitzii* populations have been described with host age, Could the author launch a multivariate analysis to validate that *F. prausnitzii* does correlate only with clinical response. The age range of the cohort varies from 36 to 85 y/o and there is a significant drop of *F. prausnitzii* in elderly patient.

Roseburia genera, if significant between PB and LTB should be represented in the figure 1, especially due to the butyrate-producing capability of *Roseburia hominis* specie.

Line 120-125: The effort provided to invalidate the published data of the association of *Akkermansia* and clinical benefit in anti-PD1 treated patient, in the context of ipilimumab, should be also provided toward *Bifidobacterium* as there is not significant higher reads % in LTB than PB. Moreover, the published data demonstrating association of *Akkermansia* with clinical benefit was done in epithelial cancers (lung and renal) and not in MM patient. Due to the difference of mechanisms of action of CTLA4 and PD1 blockades, and the different diseases, this difference might not be surprising.

Line 150, SCFAs are not the only metabolites produced by bacteria from dietary components which have an effect on immune system (indoles derivatives, polyamines).

Line 246-249: The authors described MHCII staining differences that are not shown in the paper.

Line 288-290: Please complete the sentence to indicate the group in which the memory T cells are actually increased.

Line 293: In vitro data on PBMC from healthy donor show that butyrate promotes the induction of

Tregs. (Asarat et al, 2016 Immunol Invest.)

Figure 1 a,

- The figure legend and text indicate that the analysis was performed on n=39 Metastatic Melanoma patients, however there are 42 bars. Please clarify what these 42 bars correspond to? This seems to correspond to the 41 stools samples available at V1 according to the supp table 2, but do not match the 40 cancer patients in supp table 1 (among the 40, 2 are prostate cancer)
- The patients should be clustered by clinical response to help the reader to see the increase of the genera mentioned in LTB versus PB.

Figure 1b,

The authors should also mention that with a larger cohort, they do not validate their previous data published in Annals of oncology, 2017 where they have shown an association with higher % of Bacteroides read at baseline and poor benefit. Same comment for Clostridium.

Figure 1d,

By using the median of relative abundance of *F. Prausnitzii*, the p value is not significant, this should be added to the main text. And the figure should be maybe replaced by a Kaplan-Meier graph with OS.

Figure 1e

The authors should validate the high correlation between their 16S Metagenome data and qPCR data for the 2 other bacteria tested in figure 2 (*B.fragilis* and *E.coli*)

Figure 2a

The spearman correlation between 16S *F. Prausnitzii*/ng of DNA and PFS and between 16S *B.fragilis*/ng of DNA and PFS appear to be very similar.

The correlation seems to be even better for *B.fragilis*/BACT and PFS than it is for *Prausnitzii*/ BACT. This is contradictory with the data shown in 2b and Supp 2b, and also opposite to the data already published by the same group in 2017, in Annals of Oncology.

Figure 2b

Again, the data in left panel 2a is approaching significant but not significant as the pvalue is not <0.05

As the correlation seems to be stronger in OS (figure 2a) than PFS it's maybe what should be depicted.

Figure 2c

The author should mention that the study is only considering 33 patients out of the 39 and should explain why 6 were excluded in the method section.

Due to the serum and blood SCFA correlation, the authors should maybe also represent and discussed the correlation between fecal SCFA (measured in figure supp3) and *F. Prausnitzii* abundance in the same compartment.

The data would be be easier to read and more convincing if shown as correlation and not histograms with the median of concentration.

The literature largely agreed on the butyrate producing capabilities of *F. prausnitzii*. The inverse correlation described in this study suggests that the lack of production of butyrate (or its primary substrate) by other bacteria might actually responsible to the lack of response associated to the patients who harbor a high butyrate concentration.

Figure Supp 3 and 6: Why the values (of butyrate in particular) in the serum at baseline are different between the figure supp 3 and 6? In Supp 3, one patient has a butyrate concentration close to 0umol/L of serum and no patient have value above 30 umol/L while in the figure supp 6, no patient has value close to 0umol/L and 3 are above 30umol/L.

Figure 3. The choice of the murine cell lines where two out of the three murine model used are colon cancer lines generate concern. Maybe some of the new melanoma cell lines (BRAF/PTEN, RET) could represent a better option. Due to the well described role of butyrate on normal and malignant colonocytes, colon cancer cell line might induce more questions, such as the actual concentration of butyrate reaching the tumor site, the growth rate in presence of butyrate, etc... Butyrate has profound effects on differentiation, proliferation and apoptosis of colonic epithelial cells by regulating expression of various genes associated with these processes. In colorectal cancer cells, butyrate inhibits histone deacetylases to increase the expression of genes that slow the cell cycle and induce apoptosis. Old data from the 1990's showed a decrease of invitro growth

of several human colon carcinoma cell lines in presence of butyrate. Butyrate has also been suggested for its protective role in colon cancer.

Figure Supp 8:

The increase of butyrate in the serum of mice receiving butyrate in drinking water is not striking. The decrease of acetate and propionate following oral administration of butyrate are more convincing.

In order to better appreciate the similarity between murine and human data, the author should clarify when the serum has been collected for the dosage of SCFA in mice.

Figure 4f:

Although butyrate restrains the OVA-specific IFN γ production following CTLA4 blockade during in vitro restimulation, the data also suggest that butyrate as standalone is able to induce OVA-specific IFN γ production (comparison between the two IgG2b treated groups). This could also explain the slightly slower tumor growth in isotype + butyrate group compared to isotype + water.

Figure supp 10:

The authors analyzed the memory T cell in the tumor draining lymph node in presence or absence of butyrate. Human data have shown that increase of peripheral memory CD4 cells following CTLA4 blockade is associated to a better clinical response. The authors claim that they are validating this data by observing an increase of memory CD4 cells in the tumor draining lymph node (please add following anti-CTLA4 therapy in the text, line 289)

But more importantly, the data actually suggest that butyrate increases again by itself the memory CD4 pool (Isotype treated group + butyrate versus isotype + water).

The two groups receiving anti-CTLA4 have a similar % of memory CD4, even slightly higher in the group receiving butyrate.

This requires more justification.

Figure supp 11

Did the authors also find an increase of T regs in the blood of mice following Butyrate oral administration?

Figure 5

The authors discussed the association between F. Prausnitzii and lower inflammatory proteins previously described in their 2017 study (IL-8, IL-6 and sCD25) in line with their actual data (MCP-1, TNF α , sCD25).

A (F-prausnitzii) and B (inflammatory proteins) have been associated in the previous study; B and C (Butyrate) and A and C have been associated in the current study. This doesn't necessarily mean that F.prausnitzii is responsible for the lower inflammatory proteins in the current study.

How do the authors explain the increase of TNF α in the High butyrate group? Butyrate and propionate repress LPS-induced TNF α expression by upregulation of PGE2 and COX-2 via inhibition of HDAC in PBMCs and RAW 264.7 (Cox et al, World J Gastroenterol 2009; Usami et al, Nutr Res 2008, etc)

Figure 6g and discussion text (line 404).

The data are overinterpreted and only show that IL-2 is increased significantly in patients with LTBI compared to PB after ipilimumab therapy. No link has been demonstrated with butyrate in the current graph.

Discussion:

Line 400 "As in mice, longitudinal immune monitoring in patients with MM, provided evidence that butyrate was associated with less accumulation of memory T cells and resistance to ipilimumab treatment." This is actually not true in mice. In figure supp 10, the accumulation of memory T cells is similar in the two anti-CTLA4 treated groups. However, butyrate induces by itself an accumulation in the IgG2b treated group, rendering not significant the comparison between treated and not treated with anti-CTLA4 in the butyrate-treated groups.

Minor comments:

Introduction,

Reference 10, should be added along with reference 7, lines 71 and 77.

The authors should be more precise when discussing microbiome and avoid the term "groups" of

bacteria line 89

Line 127: 16S rRNA sequencing should be referenced along to metagenomic to avoid the confusion with shotgun metagenomic.

The graphs 2a-b should be added to figure 1 to keep consistent.

The graph 4a should be added to figure 3 to validate the negative impact of butyrate in anti-tumor efficacy of CTLA4 blockade

The design of the graphs could be ameliorated and sometimes the legends is confusing. Ex: Figure 6 e,f, g. The legend is considering the color of the bars or the color of the dots.

In general, the manuscript needs careful editing to address grammatical and spelling inaccuracies.

Reviewer #3 (Remarks to the Author):

It is my honor to review the paper " Systemic gut microbial metabolites limit the anti-tumor effect of CTLA-4 blockade in hosts with cancer" by Coutzac et al.

The paper addresses a very current, important and interesting topic of interaction between microbiome and immune system, and the influence of the microbiome on the responses to immune therapies, here to a CTLA-4 blocking antibody. The paper is interesting to the community and thought provoking. The paper is well written and the references are appropriate.

I have a few comments and questions.

1. This paper is loaded with experiments and data. A lot of data is included in the supplementary submission. Because of publisher limitations on the length of submitted papers it appears that most of results are described in a very abridged way and the discussion could have more in-depth analysis. Some journals would allow authors to submit longer papers.

2. The therapy with ipilimumab results in response rate of around 15%, and about 22% of patients have a long term benefit. It is important to clarify:

- how were the discussed 39 subjects selected for the baseline stool analysis? If they had been selected sequentially

we would have expected to see many more patients in the poor-benefit group. If they had not been selected prospectively the selection could have influenced the results of the study

- were all V1 samples collected before subjects started ipilimumab?

- was ipilimumab the first line treatment?

- what were the results of stool analysis from V2-V4 samples? Did the bacteria profile change over the course of therapy?

3. It appears that the statistical difference between 2 groups was mainly achieved by the presences of outliers while the majority of results were comparable, for example Fig 1b, 2d.

4. The main concern is about differences between French sample and Italian sample analysis. How would you explain that butyrate level is important for French patients the propionate level for Italian ones?

5. Figure 4c - it is not clear from this figure if there is a difference between aCTLA4+ water and anti-CTLA4+butyrate. This is what is suggested in the text of the paper, but I am not sure if this figure actually supports it.

6. Did you attempt to stimulate dendritic cells (for example with TLR9 agonists) to overcome the influence of butyrate?

7. Is it possible that the clinical correlation is incidental and related to other prognostic factors of

these patients rather than microbiome?

8. Fig 4d shows that patients with a high butyrate level appear to have a high level of memory cells before the treatment has been started. How can it be explained?

First we would like to thank the 3 reviewers that took time to read, analyze and comment our work. We really appreciated the comments that help us to increase the clarity and the quality of our work.

We have highlighted in blue reviewers' questions.

Reviewer #1 (Remarks to the Author):

This is a very interesting and novel paper that explores the functional role of the microbiome in the response to cancer immunotherapy. While previous studies have linked microbiome composition to the degree of response to immunotherapy and have implicated direct effects of bacteria on immune cells, this paper introduces the novel concept that metabolites (SCFAs) produced by bacteria determine the likelihood of response to immunotherapy. The major claim of the paper is that microbial produced SCFAs might favor an immune tolerance profile that limits anti-CTLA-4 activity. This conclusion is well supported by the data. Strengths include the vast scope of the study and the use of both animal and human studies. Limitations are appropriately mentioned in the discussion.

We thank the reviewer for this general comment on our paper highlighting strengths and major take home messages.

Major points

1. Many critical data are (e.g. fig 5a) presented by dividing the study population in two arbitrary groups: Low and High. Considering that the depicted variables are continuous variables (e.g. Treg number), the analysis should be completed by using correlations or other statistical methods suitable for continuous variables.

We agree with reviewer comment and have added data in the revised figure 5. Butyrate and propionate (C4 and C3) are correlated with Treg while acetate (C2) failed to do so. Moreover C4 and C3 are also correlated with sCD25 at baseline. Altogether C4 and C3 but not C2 are positively associated with Tregs and sCD25.

2. Lines 299-301 and figure 5b. It is stated that positive association between serum concentrations of inflammatory proteins (MCP-1, TNF, sCD25) and serum SCFAs concentrations were noted. Considering that SCFAs increase the number of Tregs and exert anti-inflammatory activities, this is opposite than expected. Please explain.

We totally agree with reviewers comment on inflammation and high levels of SCFA, we must admit that we had the same conclusion. Indeed, this result seems clearly contradictory at first glance.

Reading the numerous papers exploring the anti-inflammatory role of SCFA particularly inhibition of NF-KappaB and inflammatory proteins, all studies utilized mainly *in vitro* SCFA at concentrations ranging from 1 mM to 100 mM (1000 μ M to 100000 μ M) (Segain JP, et al. *Gut*. 2000 Sep;47(3):397-403.; Lührs H, et al. *Int J Colorectal Dis*. 2001 Aug;16(4):195-201.; Lührs H, et al. *Dig Dis Sci*. 2001 Sep;46(9):1968-73.; Park JS, et al. *Int Immunopharmacol*. 2007 Jan;7(1):70-7; Schwab M, et al. *Mol Immunol*. 2007 Jul;44(15):3625-32; Cox et al, *World J Gastroenterol* 2009; Usami et al, *Nutr Res* 2008). In these different studies, it clearly appeared that NF-KappaB inhibition was dose and time dependent without any effect observed at low concentrations such as concentrations observed in the sera of our patients (Butyrate concentration ranging from: 6,7 μ M to 67,6 μ M in our patients both cohorts). In one study authors did a dose escalation of SCFA to determine EC50 of the % inhibition of NF-KappaB activity using cell lines, they also determined the secretion of inflammatory proteins such as TNFa or IL-6. A minimal concentration of 3 mM for butyrate was mandatory to observe a significant diminution of these inflammatory proteins (Tedelind S, et al. *World J Gastroenterol*. 2007 May 28;13(20):2826-32).

Interestingly, lower *in vitro* concentrations of butyrate such as 0.1 to 0.25 mM (100 to 250 μ M) were able to induce Treg across studies (Arpaia N et al. *Nature*. 2013; Furusawa Y et al. *Nature*. 2013; Schmidt A. et al. *PLoS*

One. 2016). Thus we believe that SCFA induce Treg at low dose but induce potent inhibition of inflammation at higher dose. In our study we clearly observed a positive correlation between Treg cells and butyrate/propionate concentrations but we also observed positive correlations with some peripheral inflammatory proteins probably due to the fact that at these concentrations no inhibition of NF-KappaB might be induced.

At these μM concentrations butyrate/propionate might not induce NF-KappaB inhibition; however it doesn't mean that they can induce inflammation. Indeed, we believe that the concentration of butyrate (or propionate) found in stools and therefore at the periphery is greater when these metabolites are less locally used in the gut thus their action on the intestinal barrier might be limited. To test this hypothesis we measured zonulin in the serum described previously as a marker of intestinal permeability (Wegh CAM et al. *J Nutr Metab.* 2019 Apr.; Sturgeon C, et al. *Ann N Y Acad Sci.* 2017). We realized this assay in a small number of patients at baseline since low numbers of samples were remaining from our cohort since several analyses have been done. Zonulin was clearly correlated with stools' concentrations of propionate/butyrate suggesting that higher concentrations of butyrate/propionate remaining in stools of patients were related to an increased intestinal permeability. This result suggests that high concentrations of SCFA in stools of patients may reflect a poor consumption by colonocytes and/or bacteria increasing gut permeability. Since increased intestinal permeability is also related to inflammation, we assessed correlation between zonulin and serum inflammatory proteins; we found that inflammatory proteins such as $\text{TNF}\alpha/\text{MCP-1}$ and zonulin were correlated (see next figure).

In conclusion, we believe that the link between measurable SCFA and peripheral inflammation is not a direct cause-and-effect relationship, but rather reflects intestinal permeability. These new data have been added revised figure 5. Moreover we added explanations in the discussion section of the revised manuscript to address this point.

Zonulin is correlated to stools' butyrate and propionate concentrations and inflammatory proteins before ipilimumab introduction. a. Data from SCFA quantification in the stool and zonulin in the serum from patients were tested for correlation ($n=9$, where both data could be assessed) before ipilimumab treatment (V1). b. Data from $\text{TNF}\alpha$ and MCP-1 quantifications in the serum were tested for correlation against serum's zonulin from patients ($n=12$, where both data were available) before ipilimumab treatment (V1). p and r are both indicated on each graph. Zonulin concentrations in serum were determined with an ELISA kit (Immundiagnostik AG, Bensheim, Germany) and measured with an ELISA plate reader at 450 nm against 620 nm as reference (Wegh CAM et al. *J Nutr Metab.* 2019 Apr 1; 2019 : 2472754)

To better characterize if butyrate could induce or not inflammation at μM concentrations, we have realized *in vitro* experiments with human PBMC. We used escalating concentration of butyrate that are consistent with the concentrations quantified in our cohorts of patients (0, $10\mu\text{M}$, $50\mu\text{M}$ and $100\mu\text{M}$). PBMCs were harvested in complete medium alone or anti-CD3 alone to mimic TCR engagement. In these experiments we monitored after 4 days of culture Treg cells and we harvest the supernatants from all conditions at the end of the culture to be able to quantify the inflammatory proteins.

As shown in the revised Figure 5

- Using physiological range of butyrate concentrations, no inhibition of the production of inflammatory proteins (revised figure 5) was observed contrary to previous published data using higher concentrations confirming that at these physiological concentrations no inhibition of NF-KappaB might be induced as already discussed earlier in this point by point reply.
- Significant increase of regulatory T cells was observed in culture after engagement of CD3 at 50 and 100 μ M but not at lower concentration i.e 10 μ M. This data is consistent with the observation made in patients where an augmentation of Treg cells is observed only in patients with the highest butyrate levels.

Revised Fig. 5. Baseline immunological markers according to serum butyrate or propionate levels. **a** and **b**, Percentage of Treg (FoxP3+ among CD4+ T cells (**a**) and sCD25 (**b**) according to median concentration of serum butyrate (left panel) and propionate (right panel) concentrations. **c** and **d**, Pearson correlations between butyrate and propionate quantification in serum and percentages of peripheral Treg (**c**; n=38) and sCD25 (**d**; n=30) before ipilimumab treatment (V1). **e**, Percentage of Treg in human PBMC (n=8) after 4 days of culture with or without anti-CD3 and escalating concentrations of butyrate. **f**, Mean (SEM) systemic TNFα according to median concentration of serum butyrate. **g**, TNFα secretion by human PBMC stimulated during 4 days with or without anti-CD3 and escalating concentrations of butyrate. **h**, Spearman correlation between SCFA quantification in the stool and zonulin in the serum from patients (n=9, where both data were available) before ipilimumab treatment (V1). **i**, Spearman correlation between zonulin and serum concentrations of TNFα (n=12, where both data were available). Each dot represents one patient. P-values are indicated on each graph; Mann-Whitney (**a**, **b**, **f**) and Two-way ANOVA followed by Tukey's tests for multiple comparisons was used (**e** and **g**). * $p < 0.05$; ** $p < 0.01$; *** $p < 0.001$; ns, not significant.

Thus it does not seem incoherent to observe inflammatory proteins along with Treg increase in patients with high butyrate concentrations in the periphery. These new data and discussion are now added in the revised Figure 5 the revised manuscript.

Lines 404-406. It is stated that :the data suggest a poor restoration of CD28 signaling after CTLA-4 blockade in patients with high levels of blood butyrate”. This is an important issue. The effects of butyrate on CD28 signaling could be studied more directly.

We thank the reviewer for this question which helps us to increase the strengths of our work. To address this specific point we have realized *in vitro* experiments with human PBMC. We used escalating concentration of butyrate that are consistent with the concentrations quantified in our cohorts of patients (0, 10 μ M, 50 μ M and 100 μ M). PBMCs were harvested in medium alone or anti-CD3 alone to mimic TCR engagement or anti-CD3 + anti-CD28 to be able to monitored the effects of butyrate when CD28 is stimulated. These experiments also recapitulate what should be observed before ipilimumab, where T cells might mostly be stimulated through TCR engagement with poor costimulatory engagement and after ipilimumab where the blockade of CTLA-4 favors the stimulation of CD28. Moreover, these experiments brought evidence a direct effect of butyrate on human PBMC. In these experiments we monitored after 4 days of culture Treg cells accumulation, ICOS induction on T cells and we harvest the supernatant from all conditions at the end of the culture to be able to quantified key soluble factors that where associated to butyrate in sera of patients (i.e. sCD25; TNFa).

As shown in the revised Figure 5 and 6 of the revised manuscript:

- Using physiological range of butyrate concentrations, no inhibition of the production of inflammatory proteins (revised figure 5) was observed contrary to previous published data using higher concentrations of butyrate confirming that at these μ M concentrations no inhibition of NF-KappaB might be induced as already discussed earlier in this point by point reply.
- Significant increase of regulatory T cells was observed in culture after engagement of CD3 at 50 and 100 μ M but not at lower concentration i.e 10 μ M. This data is consistent with observation made in patients where an augmentation of Treg cells is observed only in patients with the highest butyrate levels.
- ICOS induction on T cells after ipilimumab treatment has been suggested as a surrogate marker of ipilimumab efficacy in patients (*Fu T et al. Cancer Res 2011; 71(16): 5445–5454*). In these new experiments ICOS expression was clearly induced after anti-CD3 and anti-CD3 + anti-CD28 engagement compared to medium alone. Very interestingly butyrate limited ICOS induction in a dose dependent manner only in the condition where both CD3 and CD28 were stimulated. This data demonstrates that butyrate counters CD28 engagement/signaling. This is consistent with patients (revised figure 6) and mice data (new supplemental figure 12) where butyrate limited anti-CTLA-4-induced immunity. In patients we confirmed that the increase of ICOS on T cells after ipilimumab treatment was higher in patients with low levels of butyrate compared to patients with high levels (revised figure 6).
- Quantification of sCD25 (an IL-2 surrogate marker (*Lotze MT. et al. Cancer Res 1987; 47(8): 2188–2195*) confirmed the previous observation; butyrate limited sCD25 in a dose dependent manner only in the condition where both CD3 and CD28 were stimulated. Again consistent with patients data where the increased of sCD25 after ipilimumab treatment was higher in patients with low levels of butyrate compared to patients with high levels.

The data clearly suggest that butyrate counters CD28 signaling since inhibition is observed even in the presence of a strong agonistic antibody (anti-CD28 agonist). This inhibition led to less IL-2 production (as demonstrated by diminution of sCD25) and less ICOS induction on T cells, probably as the reflection of the diminution of IL-2 production since IL-2 is well known as a key ICOS inducer (*Riley JL et al. J Immunol 2001; 166(8):4943–4948*).

These new *in vitro* data and data observed in mice suggest that these metabolites are able to inhibit DC maturation (less induction of CD80/86) but also inhibit CD28 signaling. These new data and discussion are now added in the revised Figure 5 and 6 of the revised manuscript.

Revised Fig. 6. Immune markers during the course of ipilimumab. **a**, *t*-SNE map of markers included in the T cell 1 panel in all patients ($n=8$). **b**, *t*-SNE map with the gating of T cell ($CD3^+$) populations: $CD4^+$ T cells (red outline), $CD8^+$ T cells (blue outline), black outlines (regions 1: Naïve $CD4^+$ or $CD8^+$ T cells; regions 2: Central memory $CD4^+$ or $CD8^+$ T cells; region 3: Effector memory $CD4^+$ or $CD8^+$ T cells; regions 4: Terminally effector $CD4^+$ or $CD8^+$ T cells). **c**, *t*-SNE map at baseline (V_1) and post-ipilimumab (V_{2-3}) in patients with long term clinical benefit (upper panel) and poor clinical benefit (lower panel). **d**, Post-ipilimumab (V_{2-3}) to baseline (V_1) ratio (Ratio V_{2-3}/V_1) of memory $CD4^+$ T cells (% among $CD4^+$ T cells) according to median concentration of serum butyrate at baseline. **e**, Serum concentration of sCD25 was monitored before ipilimumab treatment (V_1) and after one or two injections of ipilimumab (V_{2-3}) according to butyrate concentration. **f**, As in **d** but for ICOS $^+$ $CD4^+$ T cells (% ICOS $^+$ among $CD4^+$ T cells). **g**, sCD25 quantification in the supernatants of PBMC after 4 days of culture in medium +/- plastic-bound anti-CD3ε +/- anti-CD28 agonist with escalating dose of butyrate. **h**, as in **g** but ICOS mean fluorescence intensity (MFI) among $CD4^+$ T cells. Each dot represents one patient. n are indicated on graphs; p -values are indicated on each graph; Wilcoxon (**e**) and Mann-Whitney (**d,f**) and two-way ANOVA (**g,h**) tests were used. * $p < 0.05$; ** $p < 0.01$; *** $p < 0.001$; ns, not significant.

Reviewer #2 (Remarks to the Author):

The paper entitled “Systemic gut microbial metabolites limit the anti-tumor effect of CTLA-4 blockade in hosts with cancer” discuss an exciting topic, using human and murine data which could be of interest for the journal.

We thank the reviewer for this enthusiastic comment.

However, additional experimental work is necessary. The current data set may not adequately supports the main claim of the manuscript. Some of the data lack of strength to fit the requirement for a publication in Nature Communication.

For clarity of our reply, we have pooled some questions from the reviewer 2.

- The authors validate their previous findings with a positive association of *Faecalibacterium prausnitzii* and clinical benefit under ipilimumab therapy. However, their previous finding of negative association of *Bacteroides* and efficacy of ipilimumab are less convincing and partially invalidated.

The goal of this study was not to demonstrate that *Bacteroides* or other bacteria were associated with resistance to ipilimumab but more to understand how microbiota composition could influence response to immunotherapy at a distant site such as melanoma lesions. In our previous work we described that while high proportions of *Bacteroides* were present at baseline in patients with poor clinical benefit ($p=0.034$), *Faecalibacterium* percentages were significantly higher in patients with long-term benefit ($p=0.0092$) (*Chaput N and Lepage P et al. Annals Oncology 2017*). Indeed in our cohort with new patients we do not obtain the same p values: for *Bacteroides* percentages were lower in patients with poor clinical benefit ($p=0.11$), for *Faecalibacterium* percentages were higher in patients with long-term benefit ($p=0.013$). We utilized different methodology and a larger cohort compared to our previous work and we still observed effects that are going in same ways. That said we must admit that we believe that *Faecalibacterium* at least in melanoma patients might represent a better surrogate marker of clinical efficacy. Moreover we brought evidence that *F. prau* enrichment is positively correlated to bacteria quantity (supplementary Figure 4) suggesting that *F. Prau* might be a surrogate marker of the gut-microbiota diversity and/or “richness” the latter being linked to clinical responses in patients as now suggested by many researchers and clinicians working on microbiota composition and response to immunotherapy in patients with cancer. This is the main reason why, today, we don’t really feel that focusing on single bacteria from the microbiota is really pertinent but we believe that metabolites produced by the microbiota composition reflecting its complex composition might be much more indicative for future studies. We have written the results accordingly in the revised manuscript.

- Due to the authors previous work describing a protective role of *F. prausnitzii* against colitis mediated by ipilimumab and the role of *F. prausnitzii* to protect against IBD, we expected the author will clarify the status of colitis of their cohort. Major comments: The toxicity status of these 39 patients should be clarified, especially due to the properties of SCFAs and *Faecalibacterium prausnitzii* to protect against IBD.

We added information on colitis development in the Supplementary Table 1.

The principal objective of this study was to understand clinical responses and not colitis. Since it is asked we have realized subgroups analyses in patients that developed colitis and those resistant to colitis. However and due to our main objective of our study these analyses should be taken with caution. Total number of patients developing ipilimumab-induced-colitis ($n=10/50$) represented 20% of the cohort, whereas in the Annals of Oncology 2017, they were 8/26 patients (31%) developing gastrointestinal toxicities. Indeed we have shown an association between microbiota composition at baseline and colitis development in patients (*Chaput N and*

Lepage P et al. *Annals Oncology* 2017). In competing risk analysis, patients belonging to cluster A (driven by *Faecalibacterium* and other Firmicutes) tended to be associated with a shorter colitis-free cumulative incidence compared with cluster B (driven by *Bacteroides*) (Gray's test; $p=0.054$ Figure 4 in *Chaput N and Lepage P et al. Annals Oncology* 2017). Moreover in the same study we showed that 15 bacterial OTUs were detected as potential biomarkers either of colitis-free progression or colitis onset during ipilimumab treatment, the top OTU associated with absence of colitis was *Prevotella* sp. (Bacteroidetes phylum, Prevotellaceae family, prevotella genus), whereas OTUs associated with subsequent colitis were from *Faecalibacterium*, *Gemmiger* and *Lachnospiraceae* (*F. prausnitzii*, butyrate producing bacterium and *G. formicilis*).

In the present cohort we could not reach significance for *Bacteroides* or *Faecalibacterium* genus suggesting that these (at the genus level) might not be the best to predict colitis even though we observed the same tendency. We have performed additional analysis for *Prevotella*, *Lachnospiraceae*, *Gemmiger*, *Faecalibacterium prausnitzii* assessed with NGS and also *Faecalibacterium prausnitzii* assessed with Q-PCR (see following Figure of this reply). Even though an enrichment of the same family, genus or species was observed in patients prone to ipilimumab-induced colitis no significance could be reached in this cohort. We observed a tendency for *Prevotella* enrichment in patients resistant to ipilimumab-induced colitis ($p=0.15$).

Revised Supplementary Fig. 1. Gut microbiota composition at baseline and ipilimumab-induced colitis. Histograms (mean \pm SEM) of relative abundance of dominant family (*Bacteroidaceae*, *Lachnospiraceae*, *Eubacteriaceae*, *Prevotellaceae* and *Ruminococcaceae*), genus (*Lachnospiraceae*, *Gemmiger*, *Prevotella*) and *Faecalibacterium prausnitzii* at baseline between patients prone ($n=9$) to or resistant ($n=29$) to ipilimumab-induced colitis. Mann-Whitney test has been applied to assess significance and p -values or ns (not significant) are indicated on the graph.

In our previous study we had also demonstrated that baseline inflammatory proteins were lower in patients that were prone to develop colitis during ipilimumab treatment (*Chaput N and Lepage P et al. Annals Oncology* 2017, supplemental Table S7). For protein concentrations measurement we use the same kits and technology. As in this our previous work, we could confirm that patients who developed ipilimumab-induced colitis have significantly lower levels of IL-6, IL-8, sCD25 and TNF α at baseline (see following Figure of this reply).

Concentration of IL-6, IL-8, sCD25 and TNF α are lower in patients that will develop colitis during ipilimumab treatment. Histograms (mean \pm SEM) concentrations IL-6, IL-8, sCD25, TNF α and MCP-1 at baseline between patients prone to or resistant to ipilimumab-induced colitis. Wilcoxon test has been applied to assess significance and P-values or ns (not significant) are indicated on the graph.

Since our current study is mainly focusing on anti-cancer activity we added data concerning colitis only in one supplementary figure 1 in order to limit confusions for future readers.

• **The use of antibiotics and the toxicity related to ipilimumab in the patient cohort should be added.**

We have already described that no clear association between ATB and microbiota composition and clinical response or toxicity after ipilimumab treatment could be observed in our previous study (*Chaput N and Lepage P et al. Annals Oncology 2017*).

However and as asked by the reviewer, we have checked in our cohort, ATB prior to ipilimumab treatment was realized only in 16% of patients (8/50). Among the 10 patients developing ipilimumab induced-colitis, 3 (30%) received antibiotics before ipilimumab introduction and among the 40 patients without colitis, 5 (12.5%) received antibiotics (Fisher's exact test one-sided p value was calculated since we have predicted the direction of the association before collecting data; p value = 0.19). Thus no clear association between colitis and ATB treatment was observed. We have added in the revised Table S1 information regarding ipilimumab-induced colitis in both cohorts.

Revised supplemental table 1:

Characteristics	French (n=50)	Italian (n=45)
Median age – year [range]	64 [36-85]	66 [22-86]
Male sex (%)	27 (54)	31 (62)
Immune checkpoint-blockade indication, n (%)		
Melanoma	48 (96)	45 (100)
Prostate carcinoma	2 (4)	0 (0)
Melanoma stage *, n (%)		
M1a	12 (24)	7 (16)
M1b	2 (4)	6 (13)
M1c	36 (72)	32 (71)
Prior therapies, n (%)		
None	31 (62)	30 (67)
Chemotherapy	5 (10)	3 (7)
BRAF-inhibitor	2 (4)	9 (20)
Hormonotherapy	2 (4)	0
Immune check point inhibitor	7 (14)	1 (2)
Others	3 (8)	2 (4)
Treatment after progression, n (%)		
None	20 (40)	9 (20)
Anti-PD-1	21 (42)	28 (62)
Others	9 (18)	8 (18)
Ipilimumab induced colitis (%)	10 (20)	9 (20)
Median survival - month [range]	18 [1-35]	12 [2-36]
Median progression - month [range]	4 [1-35]	3 [1-32]

- A lot of inconsistencies have been found all along the manuscript in the patient numbers and description and need to be addressed (some prostate cancer patients, the number of patients depicted in figure 1 do not match the text and the table of patients' description and the data from the dosage of SCFA in the same compartment at the same time point are different across figures).

We apologize for the lack of clarity. We have now indicated the number of patients for each figure and added a comprehensive table explaining the variation in the number of patient. Note that we never excluded any data from our study. The variation in numbers is due to the fact that we did not have all the samples (whole blood, dry tubes for sera, stools...) at each time point (V1, V2, V3...) for every patients.

Revised Supplementary Table 2. French cohort samples collection. The shaded cells correspond to the samples collected for each patient while ND corresponds to samples not collected for serum, fresh whole blood and stools at V₁ (baseline i.e. before ipilimumab introduction), V₂ (before the second injection of ipilimumab) and V₃ (before the third injection of ipilimumab)

Inclusion number	Stools (Q-PCR/NGS) (V1)	SCFA (stools) (V1)	SCFA (serum) (V1)	Serum (markers) (V1)	Serum (marqueurs) (V2)	PBMC (V1)	PBMC (V2)	PBMC (V3)
1		ND						ND
3								
4								
5	ND	ND	ND					ND
6								
8								
9								
11	ND	ND	ND					
12								
14								
15					ND		ND	ND
16					ND			ND
17								
19	ND	ND	ND			ND		ND
20								
21			ND	ND	ND	ND		
22								
23,5	ND			ND	ND		ND	
24								
27	ND	ND	ND					
28								
30	ND							
34								
35			ND	ND		ND		
36			ND	ND	ND	ND	ND	
37								
38								
39								
41	ND	ND		ND	ND	ND		
42				ND	ND	ND		
46								
47	ND	ND	ND	ND	ND	ND	ND	ND
48								
52	ND	ND						
55								
56	ND	ND						
57								
57,5	ND						ND	ND
64								
66								
67								
71			ND	ND	ND			
73				ND	ND			ND
74				ND	ND			
75				ND	ND		ND	
76				ND	ND			
78				ND	ND			
80				ND	ND			
82			ND	ND	ND			ND
83	ND			ND	ND			
	38	41	40	34	33	44	44	38

- Several data highlighted in the paper are not statistically significant (figure 1d, 2b) and the second human cohort do not validate the French one for butyrate.

The reviewer is right for Figure 1d which is totally misleading since it not the right data and in discordance with figure 2b in reality. We really apologize for this mistake this graph do not correspond to PFS of patients. Looking at the curve it is not PFS but OS data. We have corrected the data from OS to PFS to be in accordance with figure 2b showing PFS (revised Figure 1). For OS data we agree with reviewer, but we must keep in mind that OS is not the best echo of ipilimumab efficacy contrary to PFS since progressive patients will rapidly be switched to another treatment. In our cohort, 60% of patients did receive subsequent treatment after progression (supplemental table 1). That said we have added OS data in the revised manuscript to fit with reviewer 2 commentary.

We have corrected this figure and checked all other figures to be sure no errors remain in the revised manuscript (the new revised figure 1 is shown in this reply).

Revised Fig. 1. Baseline gut microbiota composition in patients with MM. **a**, Relative abundance of dominant (>1% of total reads) gut microbial genera are represented for each patient at V1 (baseline, n=38). **b**, Histograms of relative abundance of eight genera at baseline between poor (PB) and long term clinical benefit (LTB) patients. Mann–Whitney tests were used. *p*-values are indicated on each graph. **c**, Histograms of relative abundance of *Faecalibacterium* according to overall survival (OS over 18 months (OS>18m) and OS under 18 months (OS<18m)) in patients. Mann–Whitney tests were used. *P*-values are indicated on the graph. **d**, Kaplan–Meier of progression-free survival (PFS) of patients classified into two groups according to median relative abundance of *Faecalibacterium* (4% total reads). *P*-values are indicated on the graph. **e**, Pearson correlation between the relative abundance of *Faecalibacterium* assessed by 16S rDNA analysis (% of total reads NGS) and Q-PCR (% Q-PCR) analyses on feces at baseline (n=25). *r* and *p* values are indicated on each graph.

For the figure 2b Log-rank (Mantel Cox) test of progression-free survival (PFS) of patients classified into two groups according to median of 16S *Fprau/ng* DNA showed a *p* value of 0.0511. We can conclude that the two data have the same conclusion when looking at *Faecalibacterium* determine with NGS or *F. prau* determined with Q-PCR. That said we have analyzed in different ways to be more convincing. As shown in the next figure of this reply, we conclude and believe that patients with higher *F. prau* have a better outcome (PFS, OS, clinical benefit). These data were added in the revised figure 2 in the revised manuscript.

Association of F. Prau with clinical outcomes. a. PFS and b. OS according to the median of *F. prausnitzii* (16S *F.prau*/ng DNA) assessed by Q-PCR analyses on feces at baseline. Thirty-nine patients were included in these analyses; Unpaired T tests with Welch's correction were used. p-values are indicated on each graph.

Revised Fig. 2. Association between *F. prausnitzii*, systemic concentrations of SCFA and clinical outcomes. **a**, Quantity Mean 16S / ng DNA of *F. prausnitzii* and *B. fragilis* at baseline (Q-PCR). Each patient's microbiota is presented in the graph (n=38). 0-6M: overall survival ranging from 0 to 6 months (n=4), >6-18M: overall survival ranging from 6 to 18 months (n=23), >18M: overall survival >18 months (n=11). Two-Way ANOVA analysis was performed. *P*-values are indicated on each graph ** $p < 0.01$; *** $p < 0.001$; ns, not significant. **b**, Kaplan–Meier survival curves of PFS of patients classified into two groups according to median of *F. prausnitzii*/ng DNA (Low vs high *F. prausnitzii*/ng DNA; left panel) and 16S BACTERIA/ng DNA (low vs high BACT; right panel). Data were performed using Q-PCR analyses on feces at baseline (n=38) **c**, Spearman correlation between the relative abundance of *Firmicutes* assessed by 16S DNA analysis (% of total reads NGS) and serum butyrate at baseline (n=33). **d**, Q-PCR analyses on feces at baseline of *F. prausnitzii* (16S *F. prausnitzii*/ng DNA) according to the median concentrations of serum acetate, propionate and butyrate. **e**, As in **d** but proportion. Each dot represents a mean quantity (assessed in duplicate for each sample) for one patient. Mann–Whitney tests were used. *P*-values are indicated on each graph * $p < 0.05$; ** $p < 0.01$; ns, not significant. Kaplan–Meier survival curves of PFS according to the median value of serum propionate (left panels) and butyrate (right panels) concentrations in French (**f**) and Italian (**g**) cohort. **h**, Pearson correlations between serum concentrations of acetate, butyrate and propionate and PFS where both data were available (n=40); **i**, As in (**h**) but for OS (n=40). Each dot represents one patient. *P* and *r* are indicated on each graph.

We have also realized a Log-rank maximization method to identify a cut-off value and dichotomized patients accordingly to this value in order to see whether or not the median values chosen at first were relevant. For *Faecalibacterium* the cut of value determined (maxstat.test(Surv(time, cens) using R software) was 4.37% of total reads for PFS. Note that the median value for our population was 4% of total reads. Using the cut of value we obtained a p value of 0.0186 as shown in the following figure of this reply.

Kaplan–Meier of progression-free survival (PFS) of patients classified into two groups according to the cut-off value of abundance of *Faecalibacterium*. Log-rank maximization method was used to identify the cut-off value (4.37% of total reads). P-value is indicated on the graph.

Altogether results seemed similar with a systematic association between better outcomes and higher proportion of *Faecalibacterium*; we decide to keep data generated with medians to dichotomized patients in our study which might be less biased in small cohorts.

• **The role of propionate is suggested but not demonstrated as well as butyrate, other than the increase of Tregs in PBMC at baseline.**

We would like to moderate reviewer's comments. Indeed 1/ butyrate and propionate are two SCFA that do have similar biological effects (contrary to acetate). Butyrate and propionate are often associated with the induction of Treg contrary to acetate which has been mainly incriminated in the modulation of B cells immunity. 2/ serum butyrate and propionate before introduction of ipilimumab, were both associated with Treg while acetate failed to do so (revised fig 5). 3/ we have now added new analysis showing that both butyrate and propionate are linked to PFS and/or OS (revised figure 2 and revised Supp Fig. 10) and 4/ one's must keep in mind that serum butyrate and propionate are closely correlated contrary to acetate in both cohorts (revised Supp Fig. 10).

We have analyzed butyrate and propionate in our cohorts not only using Kaplan–Meier but also correlations between serum concentrations of butyrate, propionate and PFS or OS (revised Fig. 2). We found that both butyrate and propionate were indeed related to patients' outcomes while acetate failed to do so. Since we have the data for two cohorts we pooled the data from the Italian and French cohort (n=85) and still both butyrate and propionate from pooled cohorts are associated to PFS while acetate failed to do so (revised Supp Fig. 10).

Revised Fig. 2. Association between *F. prausnitzii*, systemic concentrations of SCFA and clinical outcomes. **a**, Quantity Mean 16S / ng DNA of *F. Prausnitzii* and *B. fragilis* at baseline (Q-PCR). Each patient's microbiota is presented in the graph (n=38). 0-6M: overall survival ranging from 0 to 6 months (n=4), >6-18M: overall survival ranging from 6 to 18 months (n=23), >18M: overall survival >18 months (n=11). Two-Way ANOVA analysis was performed. *P*-values are indicated on each graph ** *p* < 0.01; *** *p* < 0.001; ns, not significant. **b**, Kaplan–Meier survival curves of PFS of patients classified into two groups according to median of *F. Prausnitzii*/ng DNA (Low vs high *F. prau*; left panel) and 16S BACTERIA/ ng DNA (low vs high BACT; right panel). Data were performed using Q-PCR analyses on feces at baseline (n=38) **c**, Spearman correlation between the relative abundance of *Firmicutes* assessed by 16S DNA analysis (% of total reads NGS) and serum butyrate at baseline (n=33). **d**, Q-PCR analyses on feces at baseline of *F. prausnitzii* (16S *F.prau*/ng DNA) according to the median concentrations of serum acetate, propionate and butyrate. **e**, As in **d** but proportion. Each dot represents a mean quantity (assessed in duplicate for each sample) for one patient. Mann–Whitney tests were used. *P*-values are indicated on each graph * *p* < 0.05; ** *p* < 0.01; ns, not significant. Kaplan–Meier survival curves of PFS according to the median value of serum propionate (left panels) and butyrate (right panels) concentrations in French (**f**) and Italian (**g**) cohort. **h**, Pearson correlations between serum concentrations of acetate, butyrate and propionate and PFS where both data were available (n=40); **i**, As in (**h**) but for OS (n=40). Each dot represents one patient. *P* and *r* are indicated on each graph.

Revised Supplementary Fig. 10. Correlations between serum SCFA in two independent cohorts of MM patients. **a**, Pearson correlations between serum of propionate (C3) and butyrate (C4, left panel), acetate (C2) and butyrate (middle panel) and acetate and propionate (right panel) in the French ($n=40$). **b**, As in (a) but for the Italian cohort ($n=45$). **c**, Pearson correlation between serum concentrations of acetate, butyrate and propionate and PFS pooling French and Italian cohorts ($n=85$); Each dot represents one patient. **d**, Kaplan–Meier survival curves of PFS according to the median value of serum acetate concentrations in French (left panel) and Italian (right panel) cohort. p and r are indicated on each graph; ns means non-significant.

Finally, associations between propionate/butyrate and immune biomarkers other than Treg cells such as sCD25 at baseline is in line numerous studies showing that butyrate and propionate have similar biological effects contrary to acetate (see next figure). Altogether these data support the fact that both butyrate and/or propionate are associated with poor outcomes as well as immune regulatory profile at baseline in patients.

Propionate and butyrate are correlated to Treg and sCD25 at baseline. Pearson correlations between butyrate, propionate and acetate quantification in serum and percentages of peripheral Treg (right panel; n=38) and sCD25 (left panel; n=30) before ipilimumab treatment (V1).

• The focus on *B.fragilis* and *E.Coli* do not seem to be relevant in the context of SCFA. Other genera were found significant between LTB and PB patients and 1- should be represented 2-could be used in relevant comparison with *F. Prausnitzii* (other butyrate-producing bacteria belonging to the genera of Roseburia).

We used *B.fragilis* and *E.Coli* in our Q-PCR to have controls compared to *F. Prau*. But we agree that butyrate-producing bacteria would be interesting. As shown in figure 1b, we observed that high *Gemmiger* genera was also associated with LTB as we published previously (Chaput et al. Annals Oncol. 2017).

We have checked for association between clinical responses and *roseburia* as asked by the reviewer. *Roseburia* represented a little fraction of the microbiota but we could observe that patients with longer OS have an enrichment compared to patients with shorter OS (see the next figure). However, relative abundance in Roseburia was not linked to PFS as shown in the revised Fig. 1b. For *Bacteroides* we observe a diminution of the proportion in patients with longer OS even though not significant (p=0.14). We could not find a clear variation for clostridium. Thus *roseburia* and *gemminger* genera seemed associated with OS but for *roseburia* association with PFS was not convincing. We have added *roseburia* in the revised Fig. 1 and added a commentary in the revised manuscript.

Percentages of four genus at baseline and overall survival. Each patient is presented in the graph. 0-6M : overall survival ranging from 0 to 6 months (n=4), 6-12M : overall survival ranging from 6 to 12 months (n=12), 12-18M overall survival ranging from 12 to 18 months (n=11), >18M : overall survival >18 months (n=11). Kruskal-Wallis test followed by multiple comparisons were realized. * : p<0.05.

• The data associating high *F. prausnitzii* abundance and Low Butyrate (in the blood but correlated to the fecal concentration) should be investigated deeper. Due to the well-known butyrate producing capabilities of *F. Prausnitzii*, this is actually suggesting another key player missing in the current

demonstration. The literature largely agreed on the butyrate producing capabilities of *F. prausnitzii*. The inverse correlation described in this study suggests that the lack of production of butyrate (or its primary substrate) by other bacteria might actually be responsible to the lack of response associated to the patients who harbor a high butyrate concentration.

We feel uncomfortable with the reviewer statement: “*data suggest that the lack of production of butyrate (or its primary substrate) by other bacteria might actually be responsible to the lack of response associated to the patients who harbor a high butyrate concentration*”. In other words, and if we have correctly understood the reviewer commentary, the review suggests that lack of production would be associated with an overall decrease in other butyrate-producing bacteria in responders and vice versa in non-responders. However when we look at other bacteria (genus, family...) largely associated with high butyrate production we still have the same conclusions i.e. enrichment with genus, families... producing butyrate is associated with the lack of clinical responses. However, we agree with the reviewer that this result is counter-intuitive.

Indeed and as reminded by the reviewer, *F. prausnitzii* is well-known for its butyrate producing capabilities. The concentration of SCFAs in stools is influenced by several factors especially dietary components, the gut microbiota composition and gut integrity. Faecal concentrations of SCFAs measured in the stools depend on both the production by microbiota and their consumption by bacteria (cross-feeding) and colonocytes. We believe that the concentration of SCFA measured in the stool (or serum) is not a correct reflection of the production of SCFA in the gut but rather a reflection of SCFA that were not consumed. Unfortunately, very few studies assessed SCFA in serum. Most data accumulated on SCFA production rely on 1/ metabolites production by bacteria in culture (Scott KP et al. (2006) *J Bacteriol* 188:4340–4349) or 2/ metagenomics analysis and detection of pathways used by bacteria for SCFA formation (Louis P et al. (2010) *Environ Microbiol* 12:304–314 ; Reichardt N, et al. (2014). Considering these data that do not measure the concentrations, it appears difficult to conclude on the balance between production and consumption. Our data may suggest that high *F. prausnitzii* may be associated with “high use” of SCFA. In line with this explanation, it has been shown that gut microbiota impacts on the level of expression and the activity of enzymes involved in SCFAs metabolism. Particularly butyrate-producing microbes increase the level of acyl CoA dehydrogenase involved in β -oxidation, and colonic mitochondrial 3-hydroxy 3-methyl glutaryl CoA (mHMGCoA) synthase responsible for the biosynthesis of ketone bodies from butyrate. On the contrary, absence of butyrate-producing microbes led to low levels of expression of these enzymes thus low metabolism of such substrate (Cherbuy, C. et al. *Eur. J. Biochem.* 2004). Recently, it has been shown a negative correlation between the abundance of faecal *Firmicutes* and faecal SCFAs levels in healthy volunteers (Rodrigues Carrio, *Front Immunol* 2017). We observed the same negative correlation between *Firmicutes* enrichment and serum butyrate (Fig. 2c of the revised paper). Moreover, in mice, a high-amylose maize starch butyrylated (HAMSB) diet seems to be associated with expansion of *Bacteroides* species as well as higher concentrations of systemic butyrate, propionate and acetate leading to resolution of type I diabetes in mice models through induction of immunosuppression (Mariño, E. et al. *Nat. Immunol.* 2017) which could appear as counter intuitive.

Overall, these data suggest that butyrate-producing bacteria may be associated with low levels of SCFA in serum due to “higher use” via metabolic pathways using this type of substrate in the gut. We have added comments in the revised manuscripts this point in the discussion section.

• How the authors explain the increase of pro-inflammatory proteins in the High butyrate group. How the authors explain the increase of TNF α in the High butyrate group? Butyrate and propionate repress LPS-induced TNF α expression by upregulation of PGE2 and COX-2 via inhibition of HDAC in PBMCs and RAW 264.7 (Cox et al, World J Gastroenterol 2009; Usami et al, Nutr Res 2008, etc).

We totally agree with reviewers comment on inflammation and high levels of SCFA, we must admit that we had the same conclusion. Indeed, this result seems clearly contradictory at first glance.

Reading the numerous papers exploring the anti-inflammatory role of SCFA particularly inhibition of NF-KappaB and inflammatory proteins, all studies utilized mainly *in vitro* SCFA at concentrations ranging from 1 mM to 100 mM (1000 μ M to 100000 μ M) (Segain JP, et al. Gut. 2000 Sep;47(3):397-403.; Lührs H, et al. Int J Colorectal Dis. 2001 Aug;16(4):195-201.; Lührs H, et al. Dig Dis Sci. 2001 Sep;46(9):1968-73.; Park JS, et al. Int Immunopharmacol. 2007 Jan;7(1):70-7; Schwab M, et al. Mol Immunol. 2007 Jul;44(15):3625-32; Cox et al, World J Gastroenterol 2009; Usami et al, Nutr Res 2008). In these different studies, it clearly appeared that NF-KappaB inhibition was dose and time dependent without any effect observed at low concentrations such as concentrations observed in the sera of our patients (Butyrate concentration ranging from: 6,7 μ M to 67,6 μ M in our patients both cohorts). In one study authors did a dose escalation of SCFA to determine EC50 of the % inhibition of NF-KappaB activity using cell lines, they also determined the secretion of inflammatory proteins such as TNF α or IL-6. A minimal concentration of 3 mM for butyrate was mandatory to observe a significant diminution of these inflammatory proteins (Tedelind S, et al. World J Gastroenterol. 2007 May 28;13(20):2826-32).

Interestingly, lower *in vitro* concentrations of butyrate such as 0.1 to 0.25 mM (100 to 250 μ M) were able to induce Treg across studies (Arpaia N et al. Nature. 2013; Furusawa Y et al. Nature. 2013; Schmidt A. et al. PLoS One. 2016). Thus we believe that SCFA induce Treg at low dose but induce potent inhibition of inflammation at higher dose. In our study we clearly observed a positive correlation between Treg cells and butyrate/propionate concentrations but we also observed positive correlations with some peripheral inflammatory proteins probably due to the fact that at these concentrations no inhibition of NF-KappaB might be induced.

At these μ M concentrations butyrate/propionate might not induce NF-KappaB inhibition; however it doesn't mean that they can induce inflammation. Indeed, we believe that the concentration of butyrate (or propionate) found in stools and therefore at the periphery is greater when these metabolites are less locally used in the gut thus their action on the intestinal barrier might be limited. To test this hypothesis we measured zonulin in the serum described previously as a marker of intestinal permeability (Wegh CAM et al. J Nutr Metab. 2019 Apr.; Sturgeon C, et al. Ann N Y Acad Sci. 2017). We realized this assay in a small number of patients at baseline since low numbers of samples were remaining from our cohort since several analyses have been done. Zonulin was clearly correlated with stools' concentrations of propionate/butyrate suggesting that higher concentrations of butyrate/propionate remaining in stools of patients were related to an increased intestinal permeability. This result suggests that high concentrations of SCFA in stools of patients may reflect a poor consumption by colonocytes and/or bacteria increasing gut permeability. Since increased intestinal permeability is also related to inflammation, we assessed correlation between zonulin and serum inflammatory proteins; we found that inflammatory proteins such as TNF α /MCP-1 and zonulin were correlated (see next figure).

In conclusion, we believe that the link between measurable SCFA and peripheral inflammation is not a direct cause-and-effect relationship, but rather reflects intestinal permeability. These new data have been added revised figure 5. Moreover we added explanations in the discussion section of the revised manuscript to address this point.

Zonulin is correlated to stools' butyrate and propionate concentrations and inflammatory proteins before ipilimumab introduction. a. Data from SCFA quantification in the stool and zonulin in the serum from patients were tested for correlation ($n=9$, where both data could be assessed) before ipilimumab treatment (V1). b. Data from TNF α and MCP-1 quantifications in the serum were tested for correlation against serum's zonulin from patients ($n=12$, where both data were available) before ipilimumab treatment (V1). p and r are both indicated on each graph. Zonulin concentrations in serum were determined with an ELISA kit (Immundiagnostik AG, Bensheim, Germany) and measured with an ELISA plate reader at 450 nm against 620 nm as reference (Wegh CAM et al. *J Nutr Metab.* 2019 Apr 1; 2019 : 2472754)

To better characterize if butyrate could induce or not inflammation at μM concentrations, we have realized *in vitro* experiments with human PBMC. We used escalating concentration of butyrate that are consistent with the concentrations quantified in our cohorts of patients (0, 10 μM , 50 μM and 100 μM). PBMCs were harvested in complete medium alone or anti-CD3 alone to mimic TCR engagement. In these experiments we monitored after 4 days of culture Treg cells and we harvest the supernatants from all conditions at the end of the culture to be able to quantify the inflammatory proteins.

As shown in the revised Figure 5

- Using physiological range of butyrate concentrations, no inhibition of the production of inflammatory proteins (revised figure 5) was observed contrary to previous published data using higher concentrations confirming that at these physiological concentrations no inhibition of NF-KappaB might be induced as already discussed earlier in this point by point reply.
- Significant increase of regulatory T cells was observed in culture after engagement of CD3 at 50 and 100 μM but not at lower concentration i.e 10 μM . This data is consistent with the observation made in patients where an augmentation of Treg cells is observed only in patients with the highest butyrate levels.

Fig. 5. Baseline immunological markers according to serum butyrate or propionate levels. **a** and **b**, Percentage of Treg (FoxP3⁺ among CD4⁺ T cells (**a**) and sCD25 (**b**) according to median concentration of serum butyrate (left panel) and propionate (right panel) concentrations. **c** and **d**, Pearson correlations between butyrate and propionate quantification in serum and percentages of peripheral Treg (**c**; n=38) and sCD25 (**d**; n=30) before ipilimumab treatment (V1). **e**, Percentage of Treg in human PBMC (n=8) after 4 days of culture with or without anti-CD3 and escalating concentrations of butyrate. **f**, Mean (SEM) systemic TNF α according to median concentration of serum butyrate. **g**, TNF α secretion by human PBMC stimulated during 4 days with or without anti-CD3 and escalating concentrations of butyrate. **h**, Spearman correlation between SCFA quantification in the stool and zonulin in the serum from patients (n=9, where both data were available) before ipilimumab treatment (V1). **i**, Spearman correlation between zonulin and serum concentrations of TNF α (n=12, where both data were available). Each dot represents one patient. P-values are indicated on each graph; Mann-Whitney (**a**, **b**, **f**) and Two-way ANOVA followed by Tukey's tests for multiple comparisons was used (**e** and **g**). * p < 0.05; ** p < 0.01; *** p < 0.001; ns, not significant.

Thus it does not seem incoherent to observe inflammatory proteins along with Treg increase in patients with high butyrate concentrations in the periphery. These new data and discussion are now added in the revised Figure 5 the revised manuscript.

• **A (*F.prausnitzii*) and B (inflammatory proteins) have been associated in the previous study; B and C (Butyrate) and A and C have been associated in the current study. This doesn't necessarily means that *F.prausnitzii* is responsible of the lower inflammatory proteins in the current study**

We have shown in this PBPR that inflammation might be more related to the permeability of the intestinal barrier. Thus as for butyrate we believe that the link between measurable *F. prau* and peripheral inflammation is not a direct cause-and-effect relationship and we agree with reviewer claim "This doesn't necessarily means that *F.prausnitzii* is responsible of the lower inflammatory proteins in the current study". We have now clarified these points in the revised manuscript.

• Two out of the three murine model used are colon cancer lines. The phenotype of the third model, the MCA-OVA sarcoma, used for mechanistic demonstration, is less convincing, and the mechanistic data are sometimes opposite to the human data. Due to the well described role of butyrate on normal and malignant colonocytes, the use of melanoma cell lines, other than B16, such as BRAF/PTEN, RET could represent a better option.

Figure 3. The choices of the murine cell lines where two out of the three murine models used are colon cancer lines generate concern. Maybe some of the new melanoma cell lines (BRAF/PTEN, RET) could represent a better option.

First we used all tumor cell lines subcutaneously thus in the same microenvironment as we would have done for melanoma cell lines. Moreover and as described before the role of butyrate on all cell lines was tested *in vitro* and showed that these effects were not restricted to colon cancer cell lines but could also be observed with B16 (melanoma) as well as 4T1 (breast carcinoma) cell lines.

That said and as suggested by the reviewer we have search for a collaboration to have a melanoma cell lines, other than B16, such as BRAF/PTEN, RET. However after several messages, we never have any answers from the people that designed these cell lines and these cell lines are not commercially available. Thus we have used the B16K1 model (kindly given by Bruno Ségui, INSERM UMR 1037, CRCT, 31037 Toulouse, France). B16K1 is a genetically modified cell line generated from B16F10 cells, and that stably expresses the MHC-I molecule H-2Kb, known to stimulate CD8+ T cell-dependent immune responses. This cell line has been recently shown to respond to anti-PD-1 (and combo anti-PD-1/anti-CTLA-4) and is a melanoma tumor model (*Bertrand F. et al. Nature Communications volume 8, Article number: 2256 (2017)*). However this model was never tested using anti-CTLA-4 treatment as a standalone treatment.

We have realized an experiment using this model to see if butyrate could limit anti-CTLA-4 activity also in this model. We have time to realize this experiment only once (n=5 mice per group). Unfortunately, anti-CTLA-4 treatment showed weak efficacy in this model compared to others such as the CT26 model. Moreover and contrary to CT26 or other cell lines used in our study, butyrate could delay the growth of the tumor, thus to be rigorous we had to normalized tumor growth data using ratio: S_{TD_i} corresponding to the surface of the tumor the day of treatment initiation and S_{TD_x} corresponding to the surface of the tumor at day x. S_{TD_x} to S_{TD_i} ratio (S_{TD_x}/S_{TD_i}) at each day was then calculated. As shown in this PBPR we still observed that butyrate could limit anti-CTLA-4 antitumor effect.

Butyrate combined to anti-CTLA-4 in mice bearing the B16FK1 tumor model. Twenty C57BL/6 mice were treated 2 weeks before tumor inoculation and all along the experiment with sodium butyrate in the drinking water (n=10) at the concentrations of 100mM or pH-matched water (n=10) and changed every week. Three hundred thousand B16FK1 tumor cells were inoculated subcutaneously (s.c) in the right flank of mice fed with sodium butyrate in the drinking water or PH-matched water at day 0. Mice received intraperitoneally (i.p) injections of 100 μ g of anti-CTLA-4 (aCTLA-4, n=5) or its isotype control (Isotype, n=5) at D9, D13 and D16. Mice were euthanized when the tumor size was ≥ 300 mm² or boundary points were reached. Tumor volume was measured every 2-3 days; S_{TDi} corresponds to the surface of the tumor the day of treatment initiation and S_{TDx} correspond to the surface of the tumor at day x. S_{TDx} to S_{TDi} ratio (S_{TDx}/S_{TDi}) is depicted. Two-way ANOVA was used, p values are indicated on the graphs, * p < 0.05, ns means not significant.

Since this experiment was not confirmed, we preferred not to add this experiment in the revised version. If the editors ask to do so we will do it.

Altogether using 4 subcutaneously inoculated tumor cell lines (CT26, MC38, MCA101ova and B16FK1) and two independent cohorts in patients, we systematically observed that butyrate limited anti-CTLA-4 efficacy.

Due to the well described role of butyrate on normal and malignant colonocytes, colon cancer cell line might induce more questions, such as the actual concentration of butyrate reaching the tumor site, the growth rate in presence of butyrate, etc... Butyrate has profound effects on differentiation, proliferation and apoptosis of colonic epithelial cells by regulating expression of various genes associated with these processes. In colorectal cancer cells, butyrate inhibits histone deacetylases to increase the expression of genes that slow the cell cycle and induce apoptosis. Old data from the 1990's showed a decrease of invitro growth of several human colon carcinoma cell lines in presence of butyrate. Butyrate has also been suggested for its protective role in colon cancer.

In our hands tumor cell proliferation was inhibited by sodium butyrate *in vitro* as described by others in several tumor cell models, colon carcinoma cell lines as CT26 and MC38 but also melanoma and mammary carcinoma as B16F10 and 4T1 models (see the next figure). Thus these effects are not restricted to colon carcinoma cell lines.

Sodium butyrate modified accumulation of the B16F10, MC38, CT26 and 4T1 cell line in vitro. *In vitro* proliferation of B16F10, MC38, CT26 and 4T1 cell lines after 48, 72 and 96h of culture in complete medium with escalating concentration of butyrate.

• **Comments on memory T cells and antigen specific T cells in mice**

- Several data do not fit the flow of the demonstration and do not match human data (role of butyrate in the antigen-specific production of IFN γ and in the accumulation of memory T cells in tDLN, figure 4f and supp 10)
- Abstract: Butyrate does not limit the accumulation of memory T cells in mice with cancer (cf comments on Figure Supp 10)
- Line 400 “As in mice, longitudinal immune monitoring in patients with MM, provided evidence that butyrate was associated with less accumulation of memory T cells and resistance to ipilimumab treatment.” This is actually not true in mice. In figure supp 10, the accumulation of memory T cells is similar in the two anti-CTLA4 treated groups. However, butyrate induces by itself an accumulation in the IgG2b treated group, rendering not significant the comparison between treated and not treated with anti-CTLA4 in the butyrate-treated groups
- Several data do not fit the flow of the demonstration and do not match human data (role of butyrate in the antigen-specific production of IFN γ and in the accumulation of memory T cells in tDLN, figure 4f and supp 10)
- Figure supp 10: The authors analyzed the memory T cell in the tumor draining lymph node in presence or absence of butyrate. Human data have shown that increase of peripheral memory CD4 cells following CTLA4 blockade is associated to a better clinical response. The authors claim that they are validating this data by observing an increase of memory CD4 cells in the tumor draining lymph node (please add following anti-CTLA4 therapy in the text, line 289). But more importantly, the data actually suggest that butyrate increases again by itself the memory CD4 pool (Isotype treated group + butyrate versus isotype + water).
- The two groups receiving anti-CTLA4 have a similar % of memory CD4, even slightly higher in the group receiving butyrate. This requires more justification.
- Line 288-290: Please complete the sentence to indicate the group in which the memory T cells are actually increased.

We understand the reviewers comment however we did not stated that Butyrate did not limit the accumulation of memory T cells in mice with cancer (or in patients). We only showed that butyrate limits anti-CTLA-4-induced accumulation of memory T cells in mice and in patients which seems different in our opinion. Since our

conclusions were confusing, and we apologize for the lack of clarity, we have rephrased all parts of the paper dealing with these results.

One's must keep in mind that settings were different in mice and in patients.

- First, in mice models, we assessed memory T cells in the tumor draining lymph node whereas in patients, lymph nodes were not collected, data were assessed in the blood.
- Secondly, in our study, patients were monitored longitudinally allowing kinetics and matched analyses. In mice, the results concerning memory T cells were not assessed longitudinally and compared 4 different groups at the same time without any matched analyses. In mice, in the butyrate + isotype group, there was a higher proportion of memory CD4⁺ T cells in tDLN, even though not statistically significant, compared to mice with water + isotype. In mice with anti-CTLA-4 + Butyrate no accumulation of memory T cells is observed compared to isotype + Butyrate ($p > 0.05$). In line with this result, the proportion of memory CD4⁺ T cells is also higher in the high butyrate group vs low butyrate group in patients even though not significant. But Ipilimumab increased significantly memory CD4⁺ T cells in patients that have low concentration of serum butyrate in comparison with patients with high concentrations of butyrate. These data are represented by ratio in the Fig. 6d ($p < 0.001$) and demonstrate that butyrate limits ipilimumab-induced accumulation of memory T cells. It doesn't mean that butyrate alone limited the accumulation of memory T cells. As show in this PBPR, representing data from mice and patients in the same way, the two set of data seemed similar (see the next figure).

Memory CD4⁺ T cells in mice and in cancer patients. **a.** Percentage of central memory CD4⁺ T cells (CD3⁺CD4⁺CD44^{hi}CD62L⁻) in tDLN of mice (n= 6 per group) drinking water (black and white circles) compared to mice supplemented in sodium butyrate (black and white squares). One-way ANOVA was used; *p*-values are indicated on each graph. **b.** Percentages (%) of memory CD4⁺ T cells (CD3⁺CD4⁺CD45RA⁺) were monitored in fresh whole blood with T cell 1 panel before ipilimumab treatment (V₁) and after one or two injections of ipilimumab (V₃) according to butyrate concentrations in the serum of patients at baseline (left panel). Each dot represents one patient. *P*-values are indicated on each graph; Wilcoxon match-test was performed. * $p < 0.05$; ** $p < 0.01$; *** $p < 0.001$; ns, not significant.

Finally and to have a fair comparison between mice and human data, we also realized new experiments to have a longitudinal study of Memory T cells in the blood of tumor bearing mice thus same compartment as shown in the revised Supp Fig. 14. We confirmed again that butyrate could limit the antitumor efficacy of anti-CTLA-4 treatment. Moreover, These new set of experiments confirmed that in mice (as in patients) accumulation of

memory T cells after anti-CTLA-4 treatment is reduced in butyrate supplemented mice as well as ICOS induction on T cells (we have added ICOS data in human in the revised manuscript).

Altogether, our data in mice as well as in patients seemed to us similar.

Supplementary Fig. 14. Longitudinal blood immune monitoring in mice. Mice were fed with butyrate (n=10) or PH-matched water (n=10) from day -14 (D-14), CT26 tumor cells were inoculated at day 0 (D0) in both groups and treatment started at day 7 (D7) with anti-CTLA-4 blocking mAbs (αCTLA-4, n=5) or its isotype control (Isotype, n=5) and was also performed at day 10 (D10) and day 13 (D13). Blood samples were performed at day 6 (D6) after tumor inoculation but before starting the treatments (pre-αCTLA-4) and at day 15, two days after the last injection of treatments (post-αCTLA-4). **a**, Left panel summarized the experimental settings of the longitudinal experiment and right panel represents the median tumor sizes (mm2 +/- SEM) at Day 20 in each group. **b**, Percentages (%) of memory CD4⁺ T cells were monitored in fresh whole blood before (pre-αCTLA-4) and after αCTLA-4 treatment (post-αCTLA-4) (left panel); Percentages (%) of memory CD8⁺ T cells were monitored in fresh whole blood before (pre-αCTLA-4) and after αCTLA-4 treatment (post-αCTLA-4) (middle panel); Percentages (%) of ICOS⁺ CD4⁺ T cells were monitored in fresh whole blood before (pre-αCTLA-4) and after αCTLA-4 treatment (post-αCTLA-4) (right panel). Each dot represents one mouse. P-values are indicated on each graph; Two-way ANOVA was used (**a**, right panel) and paired t tests were used (**b**). * p < 0.05; ** p < 0.01; *** p < 0.001; ns, not significant.

Since the text of our manuscript was not always clear and in line with reviewers comments we have systematically add “following anti-CTLA-4 therapy” or “anti-CTLA-4-induced accumulation of memory T cells” in the revised manuscript.

For the induction of antigen-specific T cells in patients, this analysis would have supposed 1/ to have access to the draining lymph nodes and/or the tumor samples of all patients 2/ to know HLA genotype of all patients and 3/ to know antigens expressed by each tumor for all patients. This would be mandatory to be able to realize *in vitro* restimulation of antigen-specific T cells for each patient. We must admit that it’s very difficult to have all these requirements in clinical research settings. This is the main reason why we decided to address this question in mice. We read carefully the revised manuscript to check that sentences depicting production of IFN γ by antigen-specific T cells are clearly assigned to mice data without any possible confusion.

We could also discuss that it may be due the HDAC activity of butyrate. Indeed we have shown that butyrate is able to reduce the proliferation of tumor cell lines as well as to increase the level of expression of MHC class I molecules on the surface of tumor cell lines (revised supp Fig 13). This could increase tumor immunogenicity in butyrate supplemented groups leading to a slight augmentation of memory T cells. Despite this, the efficacy of anti-CTLA-4 was diminished in mice drinking sodium butyrate. In our manuscript our main point was that ipilimumab increased significantly memory CD4+ T cells in patients that have low concentration of serum butyrate in comparison with patients with high concentrations of butyrate (as demonstrated by ratio). This means that butyrate only restrained ipilimumab-induced accumulation of memory T cells.

Supplementary Fig. 13: Sodium butyrate modified proliferation and H-2Kb expression of the MCA101-OVA cell line. **a,** Tumor growth of MCA101-OVA in mice treated with three injections of isotype control + PH-matched water (IgG2b) or isotype control + sodium butyrate (IgG2b + but). The graph depicts the mean \pm SD of tumor sizes from one representative experiment out of 2 independent experiments (n = 10 mice per group). **b,** *In vitro* proliferation of MCA101-OVA cell line after 96h of culture in complete medium with escalating concentrations of sodium butyrate in vivo. **c,** Percentages of H-2Kb surface-expression after 72h of culture in complete medium with escalating concentrations of sodium butyrate in vivo. Mann–Whitney tests were used. ns, not-significant

These results have been added in supplementary files and we discussed more deeply in the discussion section as requested by reviewers.

• **Too many indirect evidences are made between clinical benefit, *F. prausnitzii*, Butyrate and anti-cancer immunity.**

Our first objective was to show that the metabolic footprint associated with the gut microbiota composition was associated with immune checkpoints blockers efficacy rather than single bacteria explaining everything. Thus this paper is not primarily intended to promote *F. Prau* against other bacteria. We apologize if the reviewer felt like it was our objective and we will be careful in the revised manuscript not to suggest this. This is probably why reviewer thinks that many indirect evidences were made between clinical benefit, *F. prausnitzii* and anti-cancer since this was not the primary endpoint of this work.

However we feel that the link between butyrate and anti-cancer immunity is quiet direct since in preclinical models the only parameter that differ between groups is the oral feeding with butyrate demonstrating that an artificial augmentation of butyrate is related to a diminished anti-CTLA-4-induced immunity in tumor bearing mice. Moreover and as already discussed, we have added *in vitro* experiments using Human PBMC to demonstrate that butyrate is able increase Tregs population *in vitro*. We also harvested human PBMC in medium alone or anti-CD3 or anti-CD3 + anti-CD28 to be able to monitored the effects of butyrate when CD28 is stimulated. These experiments recapitulate the blockade of CTLA-4 that favors CD28 stimulation. In these experiments we monitored after 4 days of culture ICOS induction on T cells and we harvest the supernatant from all conditions at the end of the culture to be able to quantify sCD25.

As shown in the revised 6:

- ICOS induction on T cells after ipilimumab treatment has been suggested as a surrogate marker of ipilimumab efficacy in patients (*Fu T et al. Cancer Res 2011; 71(16): 5445–5454*). In these new experiments ICOS expression was clearly induced after anti-CD3 and anti-CD3 + anti-CD28 engagement compared to medium alone. Very interestingly butyrate limited ICOS induction in a dose dependent manner only in the condition where both CD3 and CD28 were stimulated. This data demonstrates that butyrate counters CD28 engagement/signaling. This is consistent with patients (revised figure 6) and mice data (new supplemental figure 12) where butyrate limited anti-CTLA-4-induced immunity. In patients we confirmed that the increase of ICOS on T cells after ipilimumab treatment was higher in patients with low levels of butyrate compared to patients with high levels (revised figure 6).
- Quantification of sCD25 (an IL-2 surrogate marker (*Lotze MT. et al. Cancer Res 1987; 47(8): 2188–2195*)) confirmed the previous observation; butyrate limited sCD25 in a dose dependent manner only in the condition where both CD3 and CD28 were stimulated. Again consistent with patients data where the increased of sCD25 after ipilimumab treatment was higher in patients with low levels of butyrate compared to patients with high levels.

Data clearly suggest that butyrate directly acts on PBMC and that butyrate counters CD28 signaling since inhibition is observed even in the presence of a strong agonistic antibody (anti-CD28 agonist). This inhibition led to less IL-2 production (as demonstrated by diminution of sCD25) and less ICOS induction on T cells, probably as the reflection of the diminution of IL-2 production since IL-2 is well known as a key ICOS inducer (*Riley JL et al. J Immunol 2001; 166(8):4943–4948*).

These new data and discussion are now added in the revised 6 of the revised manuscript. Moreover, these new experiments bring evidence of a direct modulation of human primary immune cells by butyrate.

Finally we have realized the impact of Butyrate and propionate in to independent cohorts of patients showing that these metabolites were in both cohorts associated with poor clinical benefit. We have done so since a multivariate analysis in the two cohorts where the number of patients is quiet low was not reliable due to a lack of statistic power.

Revised Fig. 6. Immune markers during the course of ipilimumab. **a**, t-SNE map of markers included in the T cell 1 panel in all patients (n=8). **b**, t-SNE map with the gating of T cell (CD3⁺) populations : CD4⁺ T cells (red outline), CD8⁺ T cells (blue outline), black outlines (regions 1 : Naïve CD4⁺ or CD8⁺ T cells; regions 2 : Central memory CD4⁺ or CD8⁺ T cells; region 3 : Effector memory CD4⁺ or CD8⁺ T cells; regions 4: Terminally effector CD4⁺ or CD8⁺ T cells). **c**, t-SNE map at baseline (V₁) and post-ipilimumab (V_{2,3}) in patients with long term clinical benefit (upper panel) and poor clinical benefit (lower panel). **d**, Post-ipilimumab (V_{2,3}) to baseline (V₁) ratio (Ratio V_{2,3}/V₁) of memory CD4⁺ T cells (% among CD4⁺ T cells) according median concentration of serum butyrate at baseline. **e**, Serum concentration of sCD25 was monitored before ipilimumab treatment (V₁) and after one or two injections of ipilimumab (V_{2,3}) according to butyrate concentration. **f**, As in **d** but for ICOS⁺CD4⁺ T cells (% ICOS⁺ among CD4⁺ T cells). **g**, sCD25 quantification in the supernatants of PBMC after 4 days of culture in medium +/- plastic-bound anti-CD3ε +/- anti-CD28 agonist with escalating dose of butyrate. **h**, as in **g** but ICOS mean fluorescence intensity (MFI) among CD4⁺T cells. Each dot represents one patient. n are indicated on graphs; p-values are indicated on each graph; Wilcoxon (**e**) and Mann–Whitney (**d,f**) and two-way ANOVA (**g,h**) tests were used. * p < 0.05; ** p < 0.01; *** p < 0.001; ns, not significant.

Specific comments are as follows: Major comments:

Title: The author should specify the type of “gut microbial metabolites” they refer to in their study (buturate and propionate or SCFAs)

We have changed the title as “Systemic Short Chain Fatty Acids Limit The Anti-Tumor Effect Of CTLA-4 Blockade in Hosts With Cancer” to fit with reviewer comment.

Results text:

Line 108: Are the 39 French patients included in this study also part of previous study from the same group published in 2017, in Annals of Oncology or is this an independent cohort? The cohort (GOLD) and period of collection (March 2013-December 2014) are the same in the two studies. The authors mentioned they studied 39 melanoma patients, but in the clinical table there are 38 melanoma patients and 2 prostate cancer patients.

We apologize for this error. **Fifty French patients** (GOLD cohort) with metastatic melanoma and prostate cancer treated with ipilimumab were prospectively enrolled at Gustave Roussy Cancer Campus (Villejuif) between March 2013 and June 2015. All the stool collected were re-analyzed with the same technical analysis using Ion S5 Torrent Sequencing Analysis for 38/50 patients. We corrected in the material and methods section and Supplementary Table 1 and 2.

Significant changes in human fecal *F. prausnitzii* populations have been described with host age, Could the author launch a multivariate analysis to validate that *F. prausnitzii* does correlate only with clinical response. The age range of the cohort varies from 36 to 85 y/o and there is a significant drop of *F. prausnitzii* in elderly patient.

We thank the reviewer for his/her comment. We evaluated the level of *Facecalibacterium* (NGS) and *F. prausnitzii* (Q-PCR) in stools according to the age of patients in this French cohort. We did not observe any association. Note that the age range was 42 to 85 y/o in our cohort of patients treated with ipilimumab.

No association between age of patients and Faecalibacterium. 16S BACT/ ng DNA (left panel) *F.prau*/ng DNA (middle panel) and *Facecalibacterium* (% total reads; right panel). Data from Q-PCR analyses (left and middle panels) and NGS analysis (right panel) from patients stools (n=38) were tested for correlation against age (yrs). p and r are indicated on each graphs. ns means not significant.

Roseburia genera, if significant between PB and LTB should be represented in the figure 1, especially due to the butyrate-producing capability of *Roseburia hominis* specie.

We have now added *Roseburia* in the figure 1 of the revised manuscript as asked by reviewer 2. We have now checked for association between clinical responses and *roseburia* as asked by the reviewer (see also previously in this PBPR).

Line 120-125: The effort provided to invalidate the published data of the association of *Akkermansia* and clinical benefit in anti-PD1 treated patient, in the context of ipilimumab, should be also provided toward *Bifidobacterium* as there is not significant higher reads % in LTB than PB. Moreover, the published data demonstrating association of *Akkermansia* with clinical benefit was done in epithelial cancers (lung and renal) and not in MM patient. Due to the difference of mechanisms of action of CTLA4 and PD1 blockades, and the different diseases, this difference might not be surprising.

The sentence in our first manuscript seemed to us factual and we did not intend to invalidate *akkermansia*. "*Akkermansia* has been suggested as associated with clinical benefit after immune checkpoints treatment in lung and renal cancer patients¹⁵, however no correlation or link with clinical features was found for *Akkermansia* in patients with MM treated with ipilimumab (Supplementary Fig. 1). Note that *Akkermansia* represented a very weak proportion of the total microbiota in these MM patients (median +/- SD of % of total reads 0.03% +/- 0.27)." Our first idea was just to check for bacteria that have been found in other studies.

Faecalibacterium has been identified in MM patients treated with anti-PD-1 in as in our studies with ipilimumab in an American cohort of patients (Gopalakrishnan, V. et al. Science (2017)). Thus clinical activity of anti-PD-1 and anti-CTLA-4 could be associated to the same microbiota composition at least in MM patients. Thus we tried to see in MM patients if *Akkermansia* (found in anti-PD-1 treated patients) could be related to clinical responses in our cohort of patients but we failed to do so. We agree that the two different diseases may explain these

differences. In accordance with reviewer commentary, we discarded the entire sentence “*Akkermansia* has been suggested as associated with clinical benefit after immune checkpoints treatment in lung and renal cancer patients¹⁵, however no correlation or link with clinical features was found for *Akkermansia* in patients with MM treated with ipilimumab (Supplementary Fig. 1). Note that *Akkermansia* represented a very weak proportion of the total microbiota in these MM patients (median +/- SD of % of total reads 0.03% +/- 0.27)” from our revised manuscript.

Line 150, SCFAs are not the only metabolites produced by bacteria from dietary components which have an effect on immune system (indoles derivatives, polyamines).

We agree that SCFA are not the only metabolites produced by bacteria that can have immune modulatory properties. Indoles derivatives are well-known to promote IL-22 by innate lymphoid cells (Wang X, et al. (2014) *Nature* 514:237–241; Goto Y, et al. (2014) *Science* 345:1254009). Others metabolites as polyamines, retinoic acid, tryptophan metabolites play roles on immune cells in the intestinal tract (Wang Get al. *Cell Mol Life Sci.* 2019 Jun 27).

In the current study, our hypothesis was driven by our previous results concerning the association between the response to ipilimumab and the gut microbiota enriched in *Faecalibacterium* and other Firmicutes (Chaput and Lepage et al *Ann Oncol* 2017). In this first study, percentages of three bacteria highlighted as biomarkers at baseline for longer OS (>18 months) were *F. prausnitzii*, butyrate producing bacterium and *Gemmiger formicilis*; moreover we have showed a link between microbiota composition and peripheral immune cells such as Tregs (Chaput and Lepage et al *Ann Oncol* 2017). Another study have shown that *Faecalibacterium* could also be associated to anti-PD-1/L1 clinical responses in patients with MM and that gut microbiota composition was also associated to a specific immune profile at the periphery with less Treg and myeloid-derived suppressive cells in patients with higher proportion of *Faecalibacterium* (Gopalakrishnan V et al. *Science* 2018). These bacteria are described as butyrate producer or as being enriched in the microbiota of patients where butyrate producing bacteria are in high abundance which is the case for *Gemmiger* even though *Gemmiger* is not by itself a butyrate producer (Anand S. *Front Immunol.* 2016). All of these data led us to assume that SCFA and, in particular, butyrate or propionate could play a mandatory role. Indeed, butyrate or propionate are known to have many effects on the adaptive immunity especially Treg. This explains why we focused on SCFA.

According to the reviewer’s comment, we will explain this choice deeper the revised manuscript.

Line 246-249: The authors described MHCII staining differences that are not shown in the paper.

We are sorry for this mistake it is an error. We added accordingly in the revised manuscript (revised Fig. 4).

Revised Fig. 4. Sodium butyrate inhibited anti-CTLA-4-induced DC maturation and T cell priming. **a**, Percentages of dendritic cells (DC defined as CD3⁺CD19⁺NKp46⁺CD11c⁺MHC-II⁺ among CD45⁺ cells) in tDLN from pooled independent experiments (n=14-15 mice per group). **b**, Percentages (%) of CD80 among DC in the tDLN from pooled independent experiments (n=14-15 mice per group). **c**, Mean fluorescence intensity of CD86 among DC in the tDLN from pooled independent experiments (n=14-15 mice per group). **d**, Mean fluorescence intensity of MHC class-II among DC in the tDLN from pooled independent experiments (n=14-15 mice per group). **e and f**, Correlation between the percentage of CD86 (**e**) and CD80 (**f**) and serum butyrate concentration in mice treated with anti-CTLA-4 **g**, IFN γ concentration after *ex vivo* restimulation of tDLN with MHC-I OVA (SIINFEKL) peptides from one representative experiment out of two independent experiments (n=6 mice per group). *p*-values are indicated on each graph; One-way ANOVA followed by Tukey's tests for multiple comparisons was used (**a**, **b**, **c** and **d**). Spearman test were used (**e** and **f**). Two-way ANOVA followed by Tukey's tests for multiple comparisons was used (**g**) * *p* < 0.05; ** *p* < 0.01; *** *p* < 0.001; ns, not-significant.

Line 288-290: Please complete the sentence to indicate the group in which the memory T cells are actually increased.

We rephrased that sentence according to your comment.

Line 293: In vitro data on PBMC from healthy donor show that butyrate promotes the induction of Tregs. (Asarat et al, 2016 Immunol Invest.)

Yes, we agree but no data was provided. In the article by Asarat et al. concentrations used in vitro were high (2 mM) compared to concentrations observed in our cohorts. We added *in vitro* data with PBMCs using a range of concentrations consistent with observed concentrations in patients' blood for butyrate: 0 μM to 100 μM (revised Fig. 5 and 6). We will rephrase the sentence accordingly to the reviewers comment.

Figure 1 a, The figure legend and text indicate that the analysis was performed on n=39 Metastatic Melanoma patients, however there are 42 bars. Please clarify what these 42 bars correspond to? This seems to correspond to the 41 stools samples available at V1 according to the supp table 2, but do not match the 40 cancer patients in supp table 1 (among the 40, 2 are prostate cancer). The patients should be clustered by clinical response to help the reader to see the increase of the genera mentioned in LTB versus PB.

There were 38 patients analyzed and depicted contrary to reviewer's remark. The confusion might come from the non-representation of samples F08, F13 and F27 on the figure. Indeed these samples F08, F13 and F27 were used as technical replicates from same patients. This was done to insure that the data could be reproducible from one experiment to another. The revised Figure 1 showed patients clustered by clinical response to help the reader as ask by reviewer 2.

The authors mentioned they studied 39 melanoma patients, but in the clinical table there are 38 melanoma patients and 2 prostate cancer patients.

Yes, we agree. Since 2/50 included patients (4%) were treated with a prostate cancer and all the others for metastatic melanoma, so we decided to highlight MM in our manuscript.

Figure 1b, The authors should also mention that with a larger cohort, they do not validate their previous data published in Annals of oncology, 2017 where they have shown an association with higher % of Bacteroides read at baseline and poor benefit. Same comment for Clostridium.

We already answered to this comment previously in this PBPR.

Again the main conclusion published in the abstract of our Annals of Oncology was "*baseline gut microbiota enriched with Faecalibacterium and other Firmicutes is associated with beneficial clinical response to ipilimumab*". For this reason we think that we do validate our previous data.

Figure 1d, By using the median of relative abundance of F. Prausnitzii, the p value is not significant, this should be added to the main text. And the figure should be maybe replaced by a Kaplan-Meier graph with OS.

The reviewer is right for Figure 1d which is totally misleading since it not the right data and in discordance with fig 2b in reality. We really apologize for this mistake this graph do not correspond to PFS of patients. Looking at the curve it is not PFS but OS data. We have corrected the data from OS to PFS to be in accordance with figure 2b showing PFS (revised Figure 1). For OS data we agree with reviewer, but we must keep in mind that OS is not the best echo of ipilimumab efficacy contrary to PFS since progressive patients will rapidly be switched to another treatment. In our cohort, 60% of patients did receive subsequent treatment after progression (supplemental table 1). That said we have added OS data in the revised manuscript to fit with reviewer 2 commentary.

We have corrected this figure and checked all other figures to be sure no errors remain in the revised manuscript (the new revised figure 1 is shown previously in this reply). Moreover we now showed OS data in the revised manuscript (Figure 2 already shown in this PBPR)

Figure 1e The authors should validate the high correlation between their 16S Metagenome data and qPCR data for the 2 other bacteria tested in figure 2 (B.fragilis and E.coli)

Yes. As recommended by the reviewer we added these correlations in supplementary Figure 3. Note that the number of samples were lower than 38 because the NGS did not allow us to analyze as deeply (OTU) than in Q-PCR.

Revised Supplementary Fig. 3. Spearman correlation between the relative abundance of *B. fragilis* (n=17; were both data were available) assessed by 16S metagenomic (% of total reads NGS) and Q-PCR (% Q-PCR) analyses on feces at baseline. r and p values are indicated on the graph.

Figure 2a The spearman correlation between 16S *F. Prausnitzii*/ng of DNA and PFS and between 16S *B.fragilis*/ng of DNA and PFS appear to be very similar. The correlation seems to be even better for *B.fragilis*/BACT and PFS than it is for *Prausnitzii*/ BACT. This is contradictory with the data shown in 2b and Supp 2b, and also opposite to the data already published by the same group in 2017, in *Annals of Oncology*.

Yes we agree that these representations with red and blue colors are sometimes confusing. Thus we decided to discard that kind of too schematic and confusing representations of our manuscript and replace by real data to increase the clarity of the paper. Moreover the Kaplan-meyer analysis with *B. Fragilis* and *E. Coli* is showed in supplemental figure 3 demonstrating no association between enrichment with these bacteria and clinical outcome. All these data are now represented in the revised Figure 2 and Supplementary Figure 3.

Mean +/- SEM PFS (left panel) and OS (right panel) according to the median of *B. Fragilis* (16S *F.prau*/ng DNA) assessed by Q-PCR analyses on feces at baseline (V1). Mann-Whitney tests were used. p-values are indicated on each graph * p < 0.05; ** p < 0.01; ns, not significant.

E.coli and B. fragilis are not associated with improved PFS in MM patients. Kaplan–Meier survival curves of PFS of patients from the French cohort classified into two groups according to median of *E. coli*/ng DNA (Low vs high *E. coli*; a) and *B. fragilis*/ng DNA (low vs high *B. fragilis*; b) at baseline.

We do not feel in contradiction with our previous data. Our current data show that in patients with longer survival rates, microbiota is enriched in *F. Prausnitzii*. This data is in line with our previous results published showing association between clinical outcomes and cluster enriched in *Faecalibacterium* and other *Firmicutes* (Chaput et al, Annals of Oncology 2017). Moreover, we showed that at an OTU level, *F. Prausnitzii* L2-6 was highly represented in patients with OS longer than 18 months. Cluster B was enriched in *Bacteroides* (at a genera level), but we never showed a significant relation between *B. fragilis* and poor clinical benefit as mentioned in supplementary Fig 5 (Chaput et al, Annals of Oncology 2017) we just showed that *B. Fragilis* was not significantly associated with better outcome which is quite different in our opinion.

Figure 2b Again, the data in left panel 2a is approaching significant but not significant as the pvalue is not <0.05 As the correlation seems to be stronger in OS (figure 2a) than PFS it's maybe what should be depicted.

As discussed these representations are not suitable because it is hard to see color shades so we have discard these representations. Moreover and to fit with reviewer comments we added OS data. We decided to realized a figure as in our previous study (Annals Oncol 2017) dividing OS to be more convincing. We are now showing the association of OS with *F. Prausnitzii* and *B. fragilis*. As shown in the revised Fig 2a, all patients with an OS longer than 18 months have high enrichment with *F. Prausnitzii* while *B. fragilis* was not enriched in patients with longer OS at V1.

Quantity Mean 16S / ng ADN of F. Prausnitzii and B. fragilis at baseline. Each patient's microbiota is presented in the graph (n=38). 0-6M: overall survival ranging from 0 to 6 months (n=4), >6-18M: overall survival ranging from 6 to 18 months (n=23), >18M: overall survival >18 months (n=11).

Figure 2c The author should mention that the study is only considering 33 patients out of the 39 and should explain why 6 were excluded in the method section.

We never excluded patients from analysis and we are sorry if reviewer 2 had this feeling.

We have now indicated the number of patients for each figure (or figure legend) and added a comprehensive table explaining the variation in the number of patient. Note that we never excluded any data from our study. The variation in numbers is due to the fact that we did not have all the samples (whole blood, dry tubes for sera, stools...) at each time point (V1, V2, V3...) for every patients. This is explained in the revised Supplementary Table 2. French cohort samples collection.

Due to the serum and blood SCFA correlation, the authors should maybe also represent and discussed the correlation between fecal SCFA (measured in figure supp3) and F. Prausnitzii abundance in the same compartment.

Yes we agree with this comment however we found no clear correlations between F. Prausnitzii (assessed by Q-PCR) abundance and SCFA (C2, C3 or C4) concentrations or proportions in the fecal compartment even though a tendency were observed. Thus we prefer to focus on serum concentrations rather than fecal. Moreover, technically it is much harder to have the dosage of SCFA in feces compared to serum.

The data would be easier to read and more convincing if shown as correlation and not hectographs with the median of concentration.

We agree and removed all hectographs and replaced by more convincing graphs.

Figure Supp 3 and 6: Why the values (of butyrate in particular) in the serum at baseline are different between the figure supp 3 and 6? In Supp 3, one patient has a butyrate concentration close to 0umol/L of serum and no patient have value above 30 umol/L while in the figure supp 6, no patient has value close to 0umol/L and 3 are above 30umol/L.

We apologize for the error concerning the butyrate concentration close to 0 μM which doesn't exist. We removed it from the figure. For the other concentrations, in the Figure Supp 3 serum SCFA concentrations at baseline and during treatment are depicted and not only at baseline. No butyrate concentration is above 30 μM at baseline in our figure Supp 3 as well as in Figure Supp 6 in the French cohort (black circles). Three patients were above 30 μM in the Italian cohort (white circles).

Figure Supp 8: The increase of butyrate in the serum of mice receiving butyrate in drinking water is not striking. The decrease of acetate and propionate following oral administration of butyrate are more convincing.

We agree with the reviewer's comment, butyrate concentrations in serum is very variable. In material and methods section we detailed our procedure « For flow cytometry and serum analysis, mice were sacrificed 14 days post tumor inoculation, blood and draining lymph nodes were removed. Single cells were obtained after mechanical disruption. Serum samples were stored at -80°C until further analysis of SCFA dosages». Since butyrate was given 14 days before tumor inoculation, blood samples were realized after 28 days from the beginning of butyrate intake.

The variability of water intake between mice is well known of researchers when adding sodium butyrate in the drinking water and is mainly due to a strong unpleasant smell. This repels some mice. This is why we have realized correlations between butyrate concentrations and B7 costimulatory molecules in mice treated with CTLA-4; this allows us to see the link with the concentrations despite this variability.

In order to better appreciate the similarity between murine and human data, the author should clarify when the serum has been collected for the dosage of SCFA in mice.

We agree with the reviewer's comment, butyrate concentrations in serum is very variable. In material and methods section we detailed our procedure « For flow cytometry and serum analysis, mice were sacrificed 14 days post tumor inoculation, blood and draining lymph nodes were removed. Single cells were obtained after mechanical disruption. Serum samples were stored at -80°C until further analysis of SCFA dosages». Since butyrate was given 14 days before tumor inoculation, blood samples were realized after 28 days from the beginning of butyrate intake.

Figure 4f : Although butyrate restrains the OVA-specific IFN γ production following CTLA4 blockade during in vitro restimulation, the data also suggest that butyrate as standalone is able to induce OVA-specific IFN γ production (comparison between the two IgG2b treated groups). This could also explain the slightly slower tumor growth in isotype + butyrate group compared to isotype + water.

We agree with reviewer comment. We now discuss that it may be due the HDAC activity of butyrate. Indeed we have shown that butyrate is able to reduce the proliferation of tumor cell lines as well as to increase the level of expression of MHC class I molecules on the surface of tumor cell lines (revised supp Fig 13). This could increase tumor immunogenicity in butyrate supplemented groups leading to a slight augmentation of memory T cells and OVA-specific IFN γ production in DLN. Despite this, the efficacy of anti-CTLA-4 was diminished in mice drinking sodium butyrate.

Supplementary Fig. 13: Sodium butyrate modified proliferation and H-2Kb expression of the MCA101-OVA cell line. **a**, Tumor growth of MCA101-OVA in mice treated with three injections of isotype control + PH-matched water (IgG2b) or isotype control + sodium butyrate (IgG2b + but). The graph depicts the mean \pm SD of tumor sizes from one representative experiment out of 2 independent experiments ($n = 10$ mice per group). **b**, *In vitro* proliferation of MCA101-OVA cell line after 96h of culture in complete medium with escalating concentrations of sodium butyrate in vivo. **c**, Percentages of H-2Kb surface-expression after 72h of culture in complete medium with escalating concentrations of sodium butyrate in vivo. Mann–Whitney tests were used. ns, not-significant

These results have been added in supplementary files and we discussed in the discussion section as requested by reviewers.

Figure supp 11 Did the authors also find an increase of T regs in the blood of mice following Butyrate oral administration?

Yes we have checked in the longitudinal study in mice and found an increase of Treg cells also in mice following butyrate oral administration. This data has been added as supplemental figure 17 in the revised manuscript.

Revised Supplementary Fig. 17. Comparison of the proportion of Treg cells in the blood. Percentage of Treg cells (CD3⁺CD4⁺FoxP3⁺) in the blood of naïve mice at baseline ($n = 20$), after drinking water (H₂O, $n = 10$; white circles) or drinking sodium butyrate in drinking water (Butyrate, $n = 10$; white squares). Wilcoxon (baseline versus butyrate (white squares) and baseline versus H₂O (white circles) or Mann–Whitney (H₂O versus butyrate groups) tests were used. * $p < 0.05$; ** $p < 0.01$; *** $p < 0.001$; ns, not significant.

Figure 5 The authors discussed the association between F. Prausnitzii and lower inflammatory proteins previously described in their 2017 study (IL-8, IL-6 and sCD25) in line with their actual data (MCP-1, TNFa, sCD25).

We have written explanations earlier in this PBPR

A (F-prausnitzii) and B (inflammatory proteins) have been associated in the previous study; B and C (Butyrate) and A and C have been associated in the current study. This doesn't necessarily means that F.prausnitzii is responsible of the lower inflammatory proteins in the current study.

We have written explanations earlier in this PBPR

How the authors explain the increase of TNF α in the High butyrate group? Butyrate and propionate repress LPS-induced TNF α expression by upregulation of PGE2 and COX-2 via inhibition of HDAC in PBMCs and RAW 264.7 (Cox et al, World J Gastroenterol 2009; Usami et al, Nutr Res 2008, etc)

We have written explanations earlier in this PBPR

Figure 6g and discussion text (line 404). The data are overinterpreted and only show that IL-2 is increased significantly in patient with a LTB compared to PB after ipilimumab therapy. No link is been demonstrated with butyrate in the current graph.

As discussed and showed before we now bring evidence that butyrate is able to limit sCD25 *in vitro* only when PBMC are stimulated with anti-CD3 + anti-CD28. We have added this new data in the revised manuscript. This data is thus consistent with a significant increase of IL-2 only in patients with LTB compared to PB. However and to fit with reviewer comment we discarded this data in the revised manuscript.

Discussion:

Line 400 “As in mice, longitudinal immune monitoring in patients with MM, provided evidence that butyrate was associated with less accumulation of memory T cells and resistance to ipilimumab treatment.” This is actually not true in mice. In figure supp 10, the accumulation of memory T cells is similar in the two anti-CTLA4 treated groups. However, butyrate induces by itself an accumulation in the IgG2b treated group, rendering not significant the comparison between treated and not treated with anti-CTLA4 in the butyrate-treated groups.

We have written explanations earlier in this PBPR

Minor comments:

Introduction, Reference 10, should be added along with reference 7, lines 71 and 77. The authors should be more precise when discussing microbiome and avoid the term “groups” of bacteria line 89 Line 127: 16S rRNA sequencing should be referenced along to metagenomic to avoid the confusion with shotgun metagenomic. The graphs 2a-b should be added to figure 1 to keep consistent.

We modified accordingly.

The graph 4a should be added to figure 3 to validate the negative impact of butyrate in anti-tumor efficacy of CTLA4 blockade

We modified accordingly with a revised figure 2 and 4 in the new manuscript.

The design of the graphs could be ameliorated and sometimes the legends is confusing. Ex: Figure 6 e,f, g. The legend is considering the color of the bars or the color of the dots.

We changed the figure as recommended by the reviewer.

In general, the manuscript needs careful editing to address grammatical and spelling inaccuracies.

We carefully check the paper for grammatical and spelling inaccuracies

Reviewer #3 (Remarks to the Author):

It is my honor to review the paper " Systemic gut microbial metabolites limit the anti-tumor effect of CTLA-4 blockade in hosts with cancer" by Coutzac et al.

The paper addresses a very current, important and interesting topic of interaction between microbiome and immune system, and the influence of the microbiome on the responses to immune therapies, here to a CTLA-4 blocking antibody. The paper is interesting to the community and thought provoking. The paper is well written and the references are appropriate.

We thank the reviewer for this enthusiastic comment.

I have a few comments and questions.

1. This paper is loaded with experiments and data. A lot of data is included in the supplementary submission. Because of publisher limitations on the length of submitted papers it appears that most of results are described in a very abridged way and the discussion could have more in-depth analysis. Some journals would allow authors to submit longer papers.

We agree with the reviewer but we followed during the first submission the Nature Communications guidelines. In this revised manuscript, we have added new data and explanations in the discussion section to have more in-depth analysis.

2. The therapy with ipilimumab results in response rate of around 15%, and about 22% of patients have a long term benefit. It is important to clarify:

- how were the discussed 39 subjects selected for the baseline stool analysis? If they had been selected sequentially

Among the 50 patients enrolled prospectively, we only have access to stools samples from only 38 (not 39 this was an error and we apologize for this) patients before the first infusion of ipilimumab.

- we would have expected to see many more patients in the poor-benefit group. If they had not been selected prospectively the selection could have influenced the results of the study

We thank the reviewer for this comment. We confirm that all patients in the French and the Italian cohort were enrolled prospectively and successively. In our current study we do not evaluate objective response rate but PFS. Long-term clinical benefit was defined by response decrease of tumor burden 50% relative to baseline, according to immune related response criteria or stable disease (decrease of tumor burden of less than 50% but with less than 25% increase relative to nadir) for more than 6 months. Patients with poor benefit were defined as patients with a lack of long-term benefit, i.e. with progression-free survival (PFS) of <6 months (with immune related progressive disease defined as a confirmed increase in tumor burden 25% relative to nadir) (Snyder A, et al. *N Engl J Med* 2014; 371(23): 2189–2199.)

- were all V1 samples collected before subjects started ipilimumab?

Yes all samples described at baseline were collected before any ipilimumab infusion. We added the collected samples of the French cohort in a revised version in the Supplementary Table 2.

Supplementary Table 2. French cohort samples collection. The shaded cells correspond to the samples collected for each patient while ND corresponds to samples not collected for serum, fresh whole blood and stools

at V₁ (baseline i.e. before ipilimumab introduction), V₂ (before the second injection of ipilimumab) and V₃ (before the third injection of ipilimumab)

Inclusion#number	StoolsR_PCR/NGS[V1]	SCFAStools[V1]	SCFASerum[V1]	SerumBiomarkers[V1]	SerumBiomarkers[V2]	PBMCT[V1]	PBMCT[V2]	PBMCT[V3]
1		ND						ND
3								
4								
5	ND	ND	ND					ND
6								
8								
9								
11	ND	ND	ND					
12								
14					ND		ND	ND
15					ND			ND
16								ND
17								
19	ND	ND	ND			ND		ND
20				ND	ND	ND		
21				ND	ND			
22								
23,5	ND			ND	ND		ND	
24								
27	ND	ND	ND					
28								
30	ND							
34								
35			ND	ND		ND		
36			ND	ND	ND	ND	ND	
37								
38								
39								
41	ND	ND		ND	ND	ND		
42				ND	ND	ND		
46								
47	ND	ND	ND	ND	ND	ND	ND	ND
48								
52	ND	ND						
55								
56	ND	ND						
57								
57,5	ND						ND	ND
64								
66								
67								
71			ND	ND	ND			
73				ND	ND			ND
74				ND	ND			
75				ND	ND		ND	
76				ND	ND			
78				ND	ND			
80				ND	ND			
82			ND	ND	ND			ND
83	ND			ND	ND			
	38	41	40	34	33	44	44	38

- was ipilimumab the first line treatment?

As described in Supplementary Table 1, 38% of patients received prior treatment in the French cohort and 33% in the Italian.

Characteristics	French (n=50)	Italian (n=45)
Median age – year [range]	64 [36-85]	66 [22-86]
Male sex (%)	27 (54)	31 (62)
Immune checkpoint-blockade indication, n (%)		
Melanoma	48 (96)	45 (100)
Prostate carcinoma	2 (4)	0 (0)
Melanoma stage *, n (%)		
M1a	12 (24)	7 (16)
M1b	2 (4)	6 (13)
M1c	36 (72)	32 (71)
Prior therapies, n (%)		
None	31 (62)	30 (67)
Chemotherapy	5 (10)	3 (7)
BRAF-inhibitor	2 (4)	9 (20)
Hormonotherapy	2 (4)	0
Immune check point inhibitor	7 (14)	1 (2)
Others	3 (8)	2 (4)
Treatment after progression, n (%)		
None	20 (40)	9 (20)

Anti-PD-1	21 (42)	28 (62)
Others	9 (18)	8 (18)
Ipilimumab induced colitis (%)	10 (20)	9 (20)
Median survival - month [range]	18 [1-35]	12 [2-36]
Median progression - month [range]	4 [1-35]	3 [1-32]

- what were the results of stool analysis from V2-V4 samples? Did the bacteria profile change over the course of therapy?

In this current study we did not analyze stools from V2-V4. In our previous study *Chaput and Lepage, Annals of Oncology 2017*, we already published that gut microbiota composition was not modified by ipilimumab treatment (Supplementary Figure 1). In line with these results, Wargo's team published the same results in patients treated with PD-1 blockade (*Gopalakrishnan and al, Science 2018*). However, concentrations of SCFA in stool as well as in serum were analyzed longitudinally over the course of ipilimumab in the current study since it was not done in previous study. As for microbiota composition, we could not find significant differences in paired test analyses for SCFA (*Supp. Fig. 6*).

Supplementary Fig. 6. SCFA quantification in stool and sera in the French cohort. a, Percentage (left panel) and concentration (right panel) of fecal SCFA. **b,** Percentage (left panel) and concentration (right panel) of serum SCFA. Each dot represents one patient.

3. It appears that the statistical difference between 2 groups was mainly achieved by the presences of outliers while the majority of results were comparable, for example Fig 1b, 2d.

In Fig 1b and 2d, each point represents raw data of one patient. Indeed, there is an inter-individual variability in the relative abundance of the eight predominant bacterial genera (Fig 1b) as well as the percentage of serum acetate, propionate and butyrate among total SCFA (Fig. 2d).

To be more convincing for the reviewer, we assessed the percentage of patients with high versus low relative abundance of *F. prau* at baseline (V1) according to clinical benefit and Butyrate levels at V1 as shown in the next figure. This data confirmed that High *F. Prau* is associated to clinical outcome with 56% of patient benefiting from treatment while only 9% of patients with low *F.Prau* will benefit from treatment. Considering butyrate levels, 75% of patients with high *F.Prau* have low levels of butyrate while only 38% of patients with low *F. Prau* had low levels of butyrate.

(a) Percentage of patients with long-term benefit (LTB, blackbar) and poor benefit (PB, white bar) stratified according High and low faecalibacterium prausnitzii (*F. prau*). (b) Percentage of patients with high butyrate (white bar) and low butyrate levels (black bar) stratified according High and low *F. prau*. All patients with clinical response and stool at baseline (n=38) were included (a); All patients with serum butyrate quantification and stool at baseline (n=33) were included (b). Fischer's exact test was used; p values are indicated in the graphs. .

4. The main concern is about differences between French sample and Italian sample analysis. How would you explain that butyrate level is important for French patients the propionate level for Italian ones?

We would like to moderate reviewer's comments. Indeed 1/ butyrate and propionate are two SCFA that do have similar biological effects (contrary to acetate). Butyrate and propionate are often associated with the induction of Treg contrary to acetate which has been mainly incriminated in the modulation of B cells immunity. 2/ serum butyrate and propionate before introduction of ipilimumab, were both associated with Treg while acetate failed to do so (revised fig 5). 3/ we have now added new analysis showing that both butyrate and propionate are linked to PFS and/or OS (revised figure 2 and revised Supp Fig. 10) and 4/ one's must keep in mind that serum butyrate and propionate are closely correlated contrary to acetate in both cohorts (revised Supp Fig. 10).

We have analyzed butyrate and propionate in our cohorts not only using Kaplan–Meier but also correlations between serum concentrations of butyrate, propionate and PFS or OS (revised Fig. 2). We found that both butyrate and propionate were indeed related to patients' outcomes while acetate failed to do so. Since we have the data for two cohorts we pooled the data from the Italian and French cohort (n=85) and still both butyrate and propionate from pooled cohorts are associated to PFS while acetate failed to do so (revised Supp Fig. 10).

Revised Fig. 2. Association between *F. prausnitzii*, systemic concentrations of SCFA and clinical outcomes. **a**, Quantity Mean 16S / ng DNA of *F. Prausnitzii* and *B. fragilis* at baseline (Q-PCR). Each patient's microbiota is presented in the graph (n=38). 0-6M: overall survival ranging from 0 to 6 months (n=4), >6-18M: overall survival ranging from 6 to 18 months (n=23), >18M: overall survival >18 months (n=11). Two-Way ANOVA analysis was performed. *P*-values are indicated on each graph ** $p < 0.01$; *** $p < 0.001$; ns, not significant. **b**, Kaplan–Meier survival curves of PFS of patients classified into two groups according to median of *F. Prausnitzii*/ng DNA (Low vs high *F. prau*; left panel) and 16S BACTERIA/ ng DNA (low vs high BACT; right panel). Date were performed using Q-PCR analyses on feces at baseline (n=38) **c**, Spearman correlation between the relative abundance of *Firmicutes* assessed by 16S DNA analysis (% of total reads NGS) and serum butyrate at baseline (n=33). **d**, Q-PCR analyses on feces at baseline of *F. prausnitzii* (16S *F.prau*/ng DNA) according to the median concentrations of serum acetate, propionate and butyrate. **e**, As in **d** but proportion. Each dot represents a mean quantity (assessed in duplicate for each sample) for one patient. Mann–Whitney tests were used. *P*-values are indicated on each graph * $p < 0.05$; ** $p < 0.01$; ns, not significant. Kaplan–Meier survival curves of PFS according to the median value of serum propionate (left panels) and butyrate (right panels) concentrations in French (**f**) and Italian (**g**) cohort. **h**, Pearson correlations between serum concentrations of acetate, butyrate and propionate and PFS where both data were available (n=40); **i**, As in (**h**) but for OS (n=40). Each dot represents one patient. *P* and *r* are indicated on each graph.

Revised Supplementary Fig. 10. Correlations between serum SCFA in two independent cohorts of MM patients. **a**, Pearson correlations between serum of propionate (C3) and butyrate (C4, left panel), acetate (C2) and butyrate (middle panel) and acetate and propionate (right panel) in the French (n=40). **b**, As in (a) but for the Italian cohort (n=45). **c**, Pearson correlation between serum concentrations of acetate, butyrate and propionate and PFS pooling French and Italian cohorts (n=85); Each dot represents one patient. **d**. Kaplan–Meier survival curves of PFS according to the median value of serum acetate concentrations in French (left panel) and Italian (right panel) cohort. p and r are indicated on each graph; ns means non-significant.

Finally, associations between propionate/butyrate and immune biomarkers other than Treg cells such as sCD25 at baseline is in line numerous studies showing that butyrate and propionate have similar biological effects contrary to acetate (see next figure). Altogether these data support the fact that both butyrate and/or propionate are associated with poor outcomes as well as immune regulatory profile at baseline in patients.

Propionate and butyrate are correlated to Treg and sCD25 at baseline. Pearson correlations between butyrate, propionate and acetate quantification in serum and percentages of peripheral Treg (right panel; n=38) and sCD25 (left panel; n=30) before ipilimumab treatment (V1).

5. Figure 4c - it is not clear from this figure if there is a difference between aCTLA4+ water and anti-CTLA4+butyrate. This is what is suggested in the text of the paper, but I am not sure if this figure actually supports it.

We agree with the reviewer, these data are much more convincing for CD86 than CD80. However we do have a significant negative correlation between SCFA levels in the blood and % of CD80 on DCs. Thus we concluded that butyrate could limit the induction of CD80 and CD86 during anti-CTLA-4 treatment.

6. Did you attempt to stimulate dendritic cells (for example with TLR9 agonists) to overcome the influence of butyrate?

We thank the reviewer for this suggestion. We did an experiment with TLR9 agonists as suggested to overcome the influence of butyrate as suggested.

In a very disappointing way we could not overcome the influence of butyrate in these new experimentations in mice. Note that the median survival for the butyrate + CTLR ODN + aCTLA-4 was 18 days while it was 42 days for water + ODN CTL + aCTLA-4 (p<0.05) arguing again in this experimentation of an inhibition of antitumor effect of butyrate in mice treated with aCTLA-4. Finally median survival of mice treated with butyrate + CpG + aCTLA-4 was 18 days as for butyrate + CTLR ODN + aCTLA-4 demonstrating that CpG was not able to overcome the influence of butyrate in our hands.

Supplementary Fig. 18. CpG agonists did not overcome butyrate inhibition in combination with anti-CTLA-4. Twenty mice were treated 2 weeks before tumor inoculation and all along the experiment with sodium butyrate in the drinking water (n=10) at the concentrations of 100mM or pH-matched water (n=10) and changed every week. Two hundred thousand of CT26 tumor cells were inoculated subcutaneously (s.c) in the right flank of Balb/c mice fed with sodium butyrate in the drinking water (right panel) or PH-matched water (left panel) at day 0 . Mice received intraperitoneal (i.p) injections of 100 µg of anti-CTLA-4 (aCTLA-4, n=5) or its isotype control (Isotype, n=5) at D7, D10 and D13. Mice received subcutaneously (s.c) injections of 30 µg of CpG (CpG, n=5) or ODN control (CTRL ODN, n=5) at D7, D10 and D13. Tumor growth was measured three times a week. Mice were euthanized when the tumor size was $\geq 300 \text{ mm}^2$ or boundary points were reached. Percent survival of mice with PH-matched water (left panel); Percent survival of mice with butyrate (right panel); Log-rank (Mantel-Cox) test was used. P values are indicated on the graphs, * $p < 0.05$; ** $p < 0.01$; *** $p < 0.001$ otherwise not-significant.

One trivial explanation could be the CpG regimen i.e. unique dose of 30µg at day 7. Higher dose and/or repeated treatment could have been a better option.

Another explanation for this unsuccessful treatment could rely on observation made using PBMC in *in vitro* experiments (revised figure 5 and 6) as shown and described previously in this PBPR. These new *in vitro* with Human PBMC data and *in vivo* data observed in mice thus suggest that butyrate is able to inhibit ipilimumab-induced DC maturation (less induction of CD80/86) but also inhibit CD28 signaling pathway. Thus restoration of CD80/CD86 through the use of TLR agonists that favor DC maturation such as TLR9 agonists (CpG) might not be sufficient to counter the deleterious effect of these SCFA.

In consequence and in light of this new data the specific modulation of butyrate/propionate signaling pathway might be mandatory to restore anti-CTLA-4 efficacy in patients with high systemic levels of SCFA. SCFA are known to act also through specific receptors (GPR41, GPR43 and GPR109A) expressed on immune cells the use of GPR specific inhibitors might be a better option. However this has to be demonstrated and will be part of our future research. These new data and discussion are now added in the revised Figure 5 and 6 and new supplemental figure for CpG/butyrate data (new Supp Fig. 18).

7. Is it possible that the clinical correlation is incidental and related to other prognostic factors of these patients rather than microbiome?

Yes we have discussed this important point. The limited number of patients in our French cohort did not allow us to make a robust multivariate analysis. This why, we confirmed our data with an independent cohort of Italian patients.

Due to the discovery nature of our work, one's cannot exclude that it won't be validated in larger cohorts of patients. We discussed this point and we will have to do such work in larger cohorts. Nowadays, ipilimumab is not

used as a standalone treatment in many patients, thus first we will have to validate if SCFA are also associated to clinical outcomes in patients treated with anti-PD-1/PD-L1 since *Faecalibacterium* was also associated to clinical outcomes in MM patients (Gopalakrishnan et al. Science 2017). This is part of ongoing research in our laboratory.

8. Fig 4d shows that patients with a high butyrate level appear to have a high level of memory cells before the treatment has been started. How can it be explained?

We understand the reviewers comment however we did not stated that Butyrate did not limit the accumulation of memory T cells in mice with cancer (or in patients). We only showed that butyrate limits anti-CTLA-4-induced accumulation of memory T cells in mice and in patients which seems different in our opinion. Since our conclusions were confusing, and we apologize for the lack of clarity, we have rephrased all parts of the paper dealing with these results.

In figure 4d we show mice data not patients. We supposed that reviewer 3 was discussing figure 6d (human data for memory CD4+ T cells). We totally agree with his comment even though no statistical significance could be reach. We did not monitor the type of memory CD4+ T cells or the function of memory T cells that accumulated in in high butyrate groups. Since butyrate induces accumulation of regulatory T cells in vivo in mice and humans it could also explains this observation. We have discussed this point in the revised manuscript. Indeed butyrate was shown to regulate the size and the function of Treg pool as well as accelerating extrathymic peripheral differentiation of Treg cells (*N. Arpaia et al., Nature. 2013*).

We could also discuss that it may be due the HDAC activity of butyrate. Indeed we have shown that butyrate is able to reduce the proliferation of tumor cell lines as well as to increase the level of expression of MHC class I molecules on the surface of tumor cell lines (revised supp Fig 13). This could increase tumor immunogenicity in butyrate supplemented groups leading to a slight augmentation of memory T cells. Despite this, the efficacy of anti-CTLA-4 was diminished in mice drinking sodium butyrate. In our manuscript our main point was that ipilimumab increased significantly memory CD4+ T cells in patients that have low concentration of serum butyrate in comparison with patients with high concentrations of butyrate (as demonstrated by ratio). This means that butyrate only restrained ipilimumab-induced accumulation of memory T cells.

Supplementary Fig. 13: Sodium butyrate modified proliferation and H-2Kb expression of the MCA101-OVA cell line. **a.** Tumor growth of MCA101-OVA in mice treated with three injections of isotype control + PH-matched water (IgG2b) or isotype control + sodium butyrate (IgG2b + but). The graph depicts the mean \pm SD of tumor sizes from one representative experiment out of 2 independent experiments ($n = 10$ mice per group). **b.** *In vitro* proliferation of MCA101-OVA cell line after 96h of culture in complete medium with escalating concentrations of sodium butyrate in vivo. **c.** Percentages of H-2Kb surface-expression after 72h of culture in complete medium with escalating concentrations of sodium butyrate in vivo. Mann–Whitney tests were used. ns, not-significant

These results have been added in supplementary files and we discussed more deeply in the discussion section as requested by reviewers.

Reviewers' comments:

Reviewer #1 (Remarks to the Author):

The manuscript is much improved. Thank you for addressing my concerns.

Reviewer #2 (Remarks to the Author):

The authors provided efforts trying to address reviewer's comments, however, concerns are still remaining regarding the interpretations of the results, their analysis.

Specific comments are provided in red in the attached file.

Reviewer #3 (Remarks to the Author):

I want to thank the authors for a detailed response to my comments. I believe all of my concerns or comments have been addressed. I do not have any additional comments. The paper can be accepted.

+In blue: original reviewers' questions.

In black: author's response.

In red: new reviewers' comments,

The authors provided efforts trying to address reviewer's comments, however, concerns are still remaining regarding the interpretations of the results, their analysis.

Thank you for the effort providing the toxicity status of this cohort.

Line 108: Are the 39 French patients included in this study also part of previous study from the same group published in 2017, in Annals of Oncology or is this an independent cohort? The cohort (GOLD) and period of collection (March 2013-December 2014) are the same in the two studies. The authors mentioned they studied 39 melanoma patients, but in the clinical table there are 38 melanoma patients and 2 prostate cancer patients.

We apologize for this error. **Fifty French patients** (GOLD cohort) with metastatic melanoma and prostate cancer treated with ipilimumab were prospectively enrolled at Gustave Roussy Cancer Campus (Villejuif) between March 2013 and June 2015. All the stool collected were re-analyzed with the same technical analysis using Ion S5 Torrent Sequencing Analysis for 38/50 patients. We corrected in the material and methods section and Supplementary Table 1 and 2.

The main issue regarding this cohort remains on the composition of this cohort, how different it is from the precedent study in Annals of oncology in 2017 and there is also concern regarding the accuracy of the metadata considering the recurrent mistakes/changes in patient numbers/ graphs and legends identified and revised in this version.

The French cohort was originally n=40 and now 10 more patients have been added to the new version of the supplemental table 1 with modified values. 24 males before, 27 now. Can the authors comments on these changes?

The French cohort (GOLD) was n=55 in 2017, from Annals of Oncology.

38 patients were analyzed at baseline for stool microbiota in the current study using NGS or Q-PCR while 26 patients were part of the publication in the Annals of oncology analyzed by 16S. Meaning that some patients must have been common between the two publications, this should be stated in the manuscript.

Especially when the authors are claiming "as previously described" or "this study confirmed". The authors should clarify that they are using X new patients and X patients previously analyzed and replicated their previous results with another sequencing technic.

2. Lines 299-301 and figure 5b. It is stated that positive association between serum concentrations of inflammatory proteins (MCP-1, TNF, sCD25) and serum SCFAs concentrations were noted. Considering that SCFAs increase the number of Tregs and exert anti-inflammatory activities, this is opposite than expected. Please explain.

The authors detailed how they believe that SCFA and peripheral inflammation reflects permeability. However, the correlation between SCFA and zonulin is weak and seems to rely on one outlier patient who have a higher zonulin sera. Maybe the X axis scale should be adjusted.

???

How the authors explained that they changed the data in the original 5d (Patient with high butyrate do have high TNF) to a revised figure 5f where the results are opposite (High butyrate patient have a lower TNF concentration) ???

The authors seem to have switch the legend/data one more time. However, even by considering a switch, the values seem to be different.

Why the authors did not show (at least in supplemental material) the MCP1 data?

As a general note, it would help the reader if the authors keep consistent with the color legend, at least inside the same figure (Fig 6, for instance 6d and 6f. black low butyrate in 6d and black for high butyrate in 6f.

Figure Supp 8: The increase of butyrate in the serum of mice receiving butyrate in drinking water is not striking. The decrease of acetate and propionate following oral administration of butyrate are more convincing.

We agree with the reviewer's comment, butyrate concentrations in serum is very variable. In material and methods section, we detailed our procedure « For flow cytometry and serum analysis, mice were sacrificed 14 days post tumor inoculation, blood and draining lymph nodes were removed. Single cells were obtained after mechanical disruption. Serum samples were stored at -80°C until further analysis of SCFA dosages”. Since butyrate was given 14 days before tumor inoculation, blood samples were realized after 28 days from the beginning of butyrate intake.

The variability of water intake between mice is well known of researchers when adding sodium butyrate in the drinking water and is mainly due to a strong unpleasant smell. This repels some mice. This is why we have realized correlations between butyrate concentrations and B7 costimulatory molecules in mice treated with CTLA-4; this allows us to see the link with the concentrations despite this variability.

The smell is strong but that do not seem to repel all the mice (in our animal facility, our mice drink even more in the butyrate group than in the water group).

If drink consumption is monitored, which is an easy control to perform to make sure about the treatment dosage by oral drink, this point will be addressed.

Also, the author did change the butyrate solution only once a week while changing every 2/3 days is commonly recommended.

A lot of inconsistencies have been found all along the manuscript in the patient numbers and description and need to be addressed (some prostate cancer patients, the number of patients depicted in figure 1 do not match the text and the table of patients' description and the data from the dosage of SCFA in the same compartment at the same time point are different across figures).

This is still true and concerning as some statistic seem to rely on outliers.

Even in the Point by point reply, the authors show data with 39 patients for microbiome while they are supposed to have 38 samples.

While they had 41 stool samples at V1 available in their first submission and only 38 now?

Association of F. Prau with clinical outcomes. a. PFS and b. OS according to the median of F. prausnitzii (16S F.prau/ng DNA) assessed by Q-PCR analyses on feces at baseline. Thirty-nine patients were included in these analyses; Unpaired T tests with Welch's correction were used. p-values are indicated on each graph.

The microbiome data seem to have been re-analyzed? The statistics in figure 1 have changed. Can the authors comment on this change?

Why the claim that *Roseburia* was linked with clinical benefit (PFS) in the first submission (text) is no longer true?

The confusion with 50 Melanoma is still present in the manuscript while it's 48MM and 2 prostate cancer patients. Even if it's a majority of MM, the authors should not put MM only in the Figure legends or make sure the statistic are still true when considering only the MM patients.

• **The role of propionate is suggested but not demonstrated as well as butyrate, other than the increase of Tregs in PBMC at baseline.**

Since we have the data for two cohorts we pooled the data from the Italian and French cohort (n=85) and still both butyrate and propionate from pooled cohorts are associated to PFS while acetate failed to do so (revised Supp Fig. 10).

For the pooled cohort a Kaplan-Meier survival curves of PFS would be better.

Figure 3. The choices of the murine cell lines where two out of the three murine models used are colon cancer lines generate concern. Maybe some of the new melanoma cell lines (BRAF/PTEN, RET) could represent a better option.

First we used all tumor cell lines subcutaneously thus in the same microenvironment as we would have done for melanoma cell lines. Moreover and as described before the role of butyrate on all cell lines was tested *in vitro* and showed that these effects were not restricted to colon cancer cell lines but could also be observed with B16 (melanoma) as well as 4T1 (breast carcinoma) cell lines.

That said and as suggested by the reviewer we have search for a collaboration to have a melanoma cell lines, other than B16, such as BRAF/PTEN, RET. However after several messages, we never have any answers from the people that designed these cell lines and these cell lines are not commercially available. Thus we have used the B16K1 model (kindly given by Bruno Ségui, INSERM UMR 1037, CRCT, 31037 Toulouse, France). B16K1 is a genetically modified cell line generated from B16F10 cells, and that stably expresses the MHC-I molecule H-2Kb, known to stimulate CD8+ T cell-dependent immune responses. This cell line has been recently shown to respond to anti-PD-1 (and combo anti-PD-1/anti-CTLA-4) and is a melanoma tumor model (*Bertrand F. et al. Nature Communications volume 8, Article number: 2256 (2017)*). However this model was never tested using anti-CTLA-4 treatment as a standalone treatment.

We have realized an experiment using this model to see if butyrate could limit anti-CTLA-4 activity also in this model. We have time to realize this experiment only once (n=5 mice per group). Unfortunately, anti-CTLA-4 treatment showed weak efficacy in this model compared to others such as the CT26 model. Moreover and contrary to CT26 or other cell lines used in our study, butyrate could delay the growth of the tumor, thus to be rigorous we had to normalized tumor growth data using ratio: STD_i corresponding to the surface of the tumor the day of treatment initiation and STD_x corresponding to the surface of the tumor at day x. STD_x to STD_i ratio (STD_x/STD_i) at each day was then calculated. As shown in this PBPR we still observed that butyrate could limit anti-CTLA-4 antitumor effect.

How the authors explain the effect of Butyrate as a stand-alone could delay the tumor growth?

Also, the normalized results should be associated with the raw tumor growth data. Treating different groups with a different starting tumor size is always questionable.

• **Comments on memory T cells and antigen specific T cells in mice**

There is still an accumulation/ increase of memory T cell following anti-CTLA4 blockade in the butyrate group and the two anti-CTla4 treated groups have similar frequency of memory Tcells.

The data, even with supplemental figure 14 are still not strongly suggesting an ipilimumab-limited accumulation of memory T cell following butyrate

Figure 1 a, The figure legend and text indicate that the analysis was performed on n=39 Metastatic Melanoma patients, however there are 42 bars. Please clarify what these 42 bars correspond to? This seems to correspond to the 41 stools samples available at V1 according to the supp table 2, but do not match the 40 cancer patients in supp table 1 (among the 40, 2 are prostate cancer). The patients should be clustered by clinical response to help the reader to see the increase of the genera mentioned in LTB versus PB.

There were 38 patients analyzed and depicted contrary to reviewer’s remark. The confusion might come from the non-representation of samples F08, F13 and F27 on the figure. Indeed these samples F08, F13 and F27 were used as technical replicates from same patients. This was done to insure that the data

could be reproducible from one experiment to another. The revised Figure 1 showed patients clustered by clinical response to help the reader as asked by reviewer 2.

There is a lot of switch in legends, figures, and patient numbers. This is a major concern, They were 42 bars on the original graph figure 1a, with a legend indicating n=39. Even if they were 3 duplicated patients, $42-3=39$ not 38.

No matter if the authors did replicates of same patients, it should be clarified in the legend. It's misleading as it's assumed that 1 bar=1 patient.

The revised figure is much better and seem to have been reanalyzed to correspond to the figure legend but this is a general comment concerning for the general procedure of analysis/ data representation of the all manuscript that is creating concern about data accuracy.

6. Did you attempt to stimulate dendritic cells (for example with TLR9 agonists) to overcome the influence of butyrate?

We thank the reviewer for this suggestion. We did an experiment with TLR9 agonists as suggested to overcome the influence of butyrate as suggested.

In a very disappointing way we could not overcome the influence of butyrate in these new experimentations in mice. Note that the median survival for the butyrate + CTRL ODN + aCTLA-4 was 18 days while it was 42 days for water + ODN CTL + aCTLA-4 ($p < 0.05$) arguing again in this experimentation of an inhibition of antitumor effect of butyrate in mice treated with aCTLA-4. Finally median survival of mice treated with butyrate + CpG + aCTLA-4 was 18 days as for butyrate + CTRL ODN + aCTLA-4 demonstrating that CpG was not able to overcome the influence of butyrate in our hands.

Supplementary Fig. 18. CpG agonists did not overcome butyrate inhibition in combination with anti-CTLA-4. Twenty mice were treated 2 weeks before tumor inoculation and all along the experiment with sodium butyrate in the drinking water ($n=10$) at the concentrations of 100mM or pH-matched water ($n=10$) and changed every week. Two hundred thousand of CT26 tumor cells were inoculated subcutaneously (s.c) in the right flank of Balb/c mice fed with sodium butyrate in the drinking water (right panel) or PH-matched water (left panel) at day 0. Mice received intraperitoneal (i.p) injections of

100 µg of anti-CTLA-4 (aCTLA-4, n=5) or its isotype control (Isotype, n=5) at D7, D10 and D13. Mice received subcutaneously (s.c) injections of 30 µg of CpG (CpG, n=5) or ODN control (CTRL ODN, n=5) at D7, D10 and D13. Tumor growth was measured three times a week. Mice were euthanized when the tumor size was $\geq 300 \text{ mm}^2$ or boundary points were reached. Percent survival of mice with PH-matched water (left panel); Percent survival of mice with butyrate (right panel); Log-rank (Mantel-Cox) test was used. P values are indicated on the graphs, * $p < 0.05$; ** $p < 0.01$; *** $p < 0.001$ otherwise not-significant.

One trivial explanation could be the CpG regimen i.e. unique dose of 30µg at day 7. Higher dose and/or repeated treatment could have been a better option.

Another explanation for this unsuccessful treatment could rely on observation made using PBMC in *in vitro* experiments (revised figure 5 and 6) as shown and described previously in this PBPR. These new *in vitro* with Human PBMC data and *in vivo* data observed in mice thus suggest that butyrate is able to inhibit ipilimumab-induced DC maturation (less induction of CD80/86) but also inhibit CD28 signaling pathway. Thus restoration of CD80/CD86 through the use of TLR agonists that favor DC maturation such as TLR9 agonists (CpG) might not be sufficient to counter the deleterious effect of these SCFA.

In consequence and in light of this new data the specific modulation of butyrate/propionate signaling pathway might be mandatory to restore anti-CTLA-4 efficacy in patients with high systemic levels of SCFA. SCFA are known to act also through specific receptors (GPR41, GPR43 and GPR109A) expressed on immune cells the use of GPR specific inhibitors might be a better option. However this has to be demonstrated and will be part of our future research. These new data and discussion are now added in the revised Figure 5 and 6 and new supplemental figure for CpG/butyrate data (new Supp Fig. 18).

The interpretation of this new data should be discussed.

In order to correctly identify a difference between Butyrate and water in presence of TLR9 agonist and CTLA4 blockade, the groups should be presented on the same graph and the statistics calculated accordingly.

The expression of the result as a median of survival when they are 5 animals per group do not seem to be appropriate.

If one compares the actual survival of the groups receiving butyrate + anti-CTLA4+ CpG and the group receiving Water + anti-CTLA4+ CpG, there is still one mice alive at the end of the experiment in the butyrate + anti-CTLA4+ CpG group while none remaining in the water group.

One could argue the survival is 20% in the butyrate versus 0% in the water group, CpG could indeed overcome the influence of butyrate.

Also, accordingly to the statistics, butyrate does not seem to limit anti-CTLA4 efficacy in this experiment if you compare the groups treated with butyrate + anti-CTLA4+ ODN ctrl and the group treated with butyrate + Isotype + ODN ctrl (**), same statistic than in the water group.

In order to support the conclusions, the experiments should be repeated.

In blue: original reviewers' questions

In black: author's response after first reviewing

In red: new reviewers' comments

In green: author's response after second reviewing

First we are deeply grateful for the comments of reviewers 1 and 3 that stressed they do not have any additional comments since all their concerns were addressed in the revised version.

New reviewer 2 comments

The authors provided efforts trying to address reviewer's comments, however, concerns are still Remaining regarding the interpretations of the results, their analysis. Thank you for the effort providing the toxicity status of this cohort.

We thank the reviewer 2 for this comment.

Line 108: Are the 39 French patients included in this study also part of previous study from the same group published in 2017, in Annals of Oncology or is this an independent cohort? The cohort (GOLD) and period of collection (March 2013- december 2014) are the same in the two studies. The authors mentioned they studied 39 melanoma patients, but in the clinical table there are 38 melanoma patients and 2 prostate cancer patients.

We apologize for this error. Fifty French patients (GOLD cohort) with metastatic melanoma and prostate cancer treated with ipilimumab were prospectively enrolled at Gustave Roussy Cancer Campus (Villejuif) between March 2013 and June 2015. All the stool collected were re-analyzed with the same technical analysis using Ion S5 Torrent Sequencing Analysis for 38/50 patients. We corrected in the material and methods section and Supplementary Table 1 and 2.

The main issue regarding this cohort remains on the composition of this cohort, how different it is from the precedent study in Annals of oncology in 2017 and there is also concern regarding the accuracy of the metadata considering the recurrent mistakes/changes in patient numbers/ graphs and legends identified and revised in this version. The French cohort was originally n=40 and now 10 more patients have been added to the new version of the supplemental table 1 with modified values. 24 males before, 27 now. Can the author's comments on these changes?

We are surprised about this comment that suggested that we added 10 more patients in this new version. We did not add any patients in the analyses in the revised version compared to the first submission. As responded before, it was a mistake in the clinical table where only 40 patients were described for the clinical characteristics. These 40 patients corresponded to patients with SCFA quantification in the serum which was not correct. We depicted in the *Supplementary Table 2 (French cohort samples collection)* but also in the first version of the paper the samples from 50 patients from the beginning → thus there is no change in the number of patients from the first version. For this reason, it seemed correct to us to revise the clinical table for the French cohort including all patients and not only those with SCFA dosage in their serum. For the Italian cohort it was different since only serum were accessible for this cohort (no stools, no whole blood immune monitoring) thus number of patients with serum assessment of SCFA = number of patients of the entire cohort. Moreover in the first revised version and to add clarity as asked by reviewer 2, we modified the table to precise what samples were used for all experiments realized but still in 50 patients as always.

Another possible misunderstanding from reviewer 2 might come from the numbering of patients. Indeed in the first submission it was an arbitrary numbering of patients from 1 to 50 that did not

corresponded to the real inclusion numbers and chronological order of inclusion to prevent any possibility of making a link between a patient and health data, as imposed by regulations. In the second version we did use the real inclusion numbers and the chronological order of inclusion. Finally since it is still confusing, we proposed a revised third version with more detailed keeping an arbitrary numbering of patients from 1 to 50 less questionable in regards to patients' "anonymization" and European laws.

Finally the editor mentioned that Reviewer #2 is asking the reasons for the increased number of patients in our current manuscript compared to the original cohort published in your previous paper (Ann Oncol; DOI: 10.1093/annonc/mdx108). Regarding this point, we already clarified this point since the patients were not part of any clinical trial, but standard of care, thus additional patients were enrolled because this research could include a maximum of 100 patients treated by immune checkpoints (anti-CTLA-4 or anti-PD-1). In our previous paper (Ann Oncol; DOI: 10.1093/annonc/mdx108) we generated data from the first patients included and treated with anti-CTLA-4 (n=26) but the GOLD research was still ongoing at that time. In the present study we included all patients from the GOLD study treated with anti-CTLA-4. No more patients are now part of this study that is not recruiting anymore. I Hope that these additional information are clear.

Revised Supplementary Table 2. French cohort samples collection. Samples collected for each patient for serum, fresh whole blood (WB) and stools at V_1 (baseline i.e. before ipilimumab introduction), V_2 (before the second injection of ipilimumab) and V_3 (before the third injection of ipilimumab) are indicated. YES = Analysis performed on this sample; NS = No Sample; ND** = Not enough aliquots to perform all analyses. YES£= for patient #18 CCR7 flow cytometry antibody was missing in the tube thus memory T cells was not available at V3, for patient #26 ICOS flow cytometry antibody was missing in the tube thus ICOS positive T cells were not available at V1 for this patient.

Patients #	Stools Q-PCR/NGS	SCFA stools (V1)	SCFA serum (V1)	Serum markers (V1)	Serum markers (V2)	Serum markers (V3)	WB (V1)	WB (V2)	WB (V3)
1	YES	ND**	YES	YES	YES	YES	YES	YES	NS
2	YES	YES	YES	YES	YES	YES	YES	YES	YES
3	YES	YES	YES	YES	YES	YES	YES	YES	YES
4	NS	NS	NS	YES	YES	NS	YES	YES	NS
5	YES	YES	YES	YES	YES	YES	YES	YES	YES
6	YES	YES	YES	YES	YES	YES	YES	YES	YES
7	YES	YES	YES	YES	YES	YES	YES	YES	YES
8	NS	NS	NS	YES	YES	YES	YES	YES	YES
9	YES	YES	YES	YES	YES	YES	YES	YES	YES
10	YES	YES	YES	YES	NS	NS	YES	NS	NS
11	YES	YES	YES	YES	NS	YES	YES	YES	NS
12	YES	YES	YES	YES	YES	NS	YES	YES	NS
13	YES	YES	YES	YES	YES	YES	YES	YES	YES
14	NS	NS	NS	YES	YES	NS	YES	YES	NS
15	YES	YES	YES	YES	YES	YES	YES	YES	YES
16	YES	YES	NS	NS	NS	NS	NS	YES	YES
17	YES	YES	YES	YES	YES	YES	YES	YES	YES
18	ND**	YES	YES	ND**	NS	NS	YES	NS	YES
19	YES	YES	YES	YES	YES	NS	YES	YES	NS
20	NS	NS	NS	YES	YES	YES	YES	YES	YES
21	YES	YES	YES	YES	YES	YES	YES	YES	YES
22	ND**	YES	YES	YES	YES	NS	YES	YES	NS
23	YES	YES	YES	YES	YES	YES	YES	YES	YES
24	YES	YES	NS	NS	YES	NS	NS	YES	NS
25	YES	YES	NS	NS	NS	NS	NS	NS	YES
26	YES	YES	YES	YES	YES	YES	YES	YES	YES
27	YES	YES	YES	YES	YES	YES	YES	YES	YES
28	YES	YES	YES	YES	YES	YES	YES	YES	YES
29	NS	NS	YES	ND**	NS	NS	NS	YES	YES
30	YES	YES	YES	ND**	NS	NS	NS	YES	YES
31	YES	YES	YES	YES	YES	YES	YES	YES	YES
32	NS	NS	NS	NS	NS	NS	NS	NS	NS
33	YES	YES	YES	YES	YES	YES	YES	YES	YES
34	NS	NS	YES	YES	YES	YES	YES	YES	YES
35	YES	YES	YES	YES	YES	YES	YES	YES	YES
36	NS	NS	YES	YES	YES	YES	YES	YES	YES
37	YES	YES	YES	YES	YES	YES	YES	YES	YES
38	ND**	YES	YES	YES	YES	NS	YES	NS	NS
39	YES	YES	YES	YES	YES	YES	YES	YES	YES
40	YES	YES	YES	YES	YES	YES	YES	YES	YES
41	YES	YES	YES	YES	YES	YES	YES	YES	YES
42	YES	YES	NS	NS	NS	NS	YES	YES	YES
43	YES	YES	YES	ND**	NS	NS	YES	YES	NS
44	YES	YES	YES	ND**	NS	NS	YES	YES	YES
45	YES	YES	YES	ND**	NS	NS	YES	NS	YES
46	YES	YES	YES	ND**	NS	NS	YES	YES	YES
47	YES	YES	YES	ND**	NS	NS	YES	YES	YES
48	YES	YES	YES	ND**	NS	NS	YES	YES	YES
49	YES	YES	NS	NS	NS	NS	YES	YES	NS
50	ND**	YES	YES	ND**	NS	NS	YES	YES	YES
Total "YES" samples, n (%)	38 (76%)	41 (82%)	40 (80%)	34 (68%)	33 (66%)	27 (54%)	44 (88%)	44 (88%)	37 (74%)

The French cohort (GOLD) was n=55 in 2017, from Annals of Oncology. 38 patients were analyzed at baseline for stool microbiota in the current study using NGS or Q-PCR while 26 patients were part of the publication in the Annals of oncology analyzed by 16S. Meaning that some patients must have been common between the two publications, this should be stated in the manuscript. Especially when the authors are claiming “as previously described” or “this study confirmed”. The authors should clarify that they are using X new patients and X patients previously analyzed and replicated their previous results with another sequencing technic.

Yes some patients were analyzed in the Annals of Oncology 2017 but all patients were reanalyzed again in this study using a unique technology for bacterial DNA extraction as well as for Ion S5 Torrent Sequencing Analysis and Q-PCR analysis. In our previous work published in Annals of Oncology, both 454 pyrosequencing (Life Sciences, a Roche company, Branford, CT) and MiSeq (Illumina, Inc., San Diego, CA) technologies were applied. That was a limitation of our previous work at that time as discussed during the review process of this paper accepted by Annals of Oncology. Note that in our previous work no Q-PCR was not realized.

The GOLD study was a medical biological research that was still ongoing after the first 55 patients enrolled and described in Annals of Oncology 2017. The present study is done with all patients included in the GOLD study (from March 2013 to June 2015). To be clear and avoid any confusion we have rewritten this part (as in the Annals of Oncology 2017) in the second revised version of the Supplemental Material and Methods : “Eighty-five patients were prospectively enrolled in the French study (GOLD cohort) at Gustave Roussy Cancer Campus (Villejuif) between March 2013 and June 2015. Among them, 28 patients did never receive ipilimumab, 4 patients received only a single infusion of ipilimumab and died soon thereafter due to their disease and 3 patients were lost for the follow-up. They were therefore excluded from further analyses. The remaining fifty patients with metastatic melanoma (48/50) or metastatic prostate carcinoma (2/50) were followed-up for at least 6 months. Fifty Italian patients with metastatic melanoma treated with ipilimumab were prospectively enrolled at Istituto Nazionale Tumori Fondazione G. Pascale (Napoli) between July 2014 and March 2016 respectively. Among them, 5 patients received only a single infusion of ipilimumab and died soon thereafter due to their disease and were therefore excluded from further analyses. The remaining forty-five patients with metastatic melanoma were followed-up for at least 6 months.”

Finally and as asked by reviewer 2 we specified in the second revised version that some patients must have been common between the two publications. We tried to be more precise in our sentences by removing all “as previously described” or “this study confirmed” that should have been now replaced by “as previously analyzed and replicated with sequencing technics...”

2. Lines 299-301 and figure 5b. It is stated that positive association between serum concentrations of inflammatory proteins (MCP-1, TNF, sCD25) and serum SCFAs concentrations were noted. Considering that SCFAs increase the number of Tregs and exert anti-inflammatory activities, this is opposite than expected. Please explain.

The authors detailed how they believe that SCFA and peripheral inflammation reflects permeability. However, the correlation between SCFA and zonulin is weak and seems to rely on one outlier patient who have a higher zonulin sera. Maybe the X axis scale should be adjusted. ???

Yes we agree and we present now the data separately with a linear scale. We thank reviewer two for this comment. See revised Figure 5 in the second revised version.

How the authors explained that they changed the data in the original 5d (Patient with high butyrate do have high TNF) to a revised figure 5f where the results are opposite (High butyrate patient have a lower TNF concentration) ??? The authors seem to have switch the legend/data one more time.

Yes thank you for this remark; it is clear that we had switched the legend/data. We have corrected in the revised figure 5.

However, even by considering a switch, the values seem to be different.

In the first version in was median (95% CI) value, in the second it was Mean (SEM). We add in this PBPR the graph with median (95% CI) with the same scale as in first version to show reviewer 2 that it is the same data.

Why the authors did not show (at least in supplemental material) the MCP1 data?

We now show all MCP-1 data in the revised figure 5 as asked by the reviewer.

As a general note, it would help the reader if the authors keep consistent with the color legend, at least inside the same figure (Fig 6, for instance 6d and 6f. black low butyrate in 6d and black for high butyrate in 6f).

OK we have modified according to reviewer suggestion. Now low butyrate is black and high butyrate is white in the revised figure 5 and 6.

Fig. 5. Baseline immunological markers according to serum butyrate or propionate levels. **a** and **b**, Percentage of Treg (FoxP3⁺ among CD4⁺ T cells (**a**) and sCD25 (**b**) according to median concentration of serum butyrate (left panel) and propionate (right panel) concentrations. **c** and **d**, Pearson correlations between butyrate and propionate quantification in serum and percentages of peripheral Treg (**c**; n=38) and sCD25 (**d**; n=30) before ipilimumab treatment (V1). **e**, Percentage of Treg in human PBMC (n=8) after 4 days of culture with or without anti-CD3 and escalating concentrations of butyrate. **f**, Mean (SEM) concentrations of systemic TNF α (left panel) and MCP-1 (right panel) according to median concentration of serum butyrate. **g**, Mean (SEM) of systemic TNF α (left panel) and MCP-1 (right panel) according to median concentration of serum propionate. **h**, TNF α (left panel) and MCP-1 (right panel) secretions by human PBMC stimulated during 4 days with or without anti-CD3 and escalating concentrations of butyrate. **i**, Spearman correlation between butyrate (left panel) and propionate (right panel) quantifications in the stool and zonulin in the serum from patients (n=9, where both data were available) before ipilimumab treatment (V1). **j**, Spearman correlation between zonulin and serum concentrations of TNF α (light grey squares) and MCP-1 (black squares) (n=12, where both data were available). Each dot represents one patient. P-values are indicated on each graph; Unpaired t test (**a**, **b**, **f**, **g**) and Two-way ANOVA followed by Tukey's tests for multiple comparisons was used (**e** and **g**). * p < 0.05; ** p < 0.01; *** p < 0.001; ns, not significant.

Figure Supp 8: The increase of butyrate in the serum of mice receiving butyrate in drinking water is not striking. The decrease of acetate and propionate following oral administration of butyrate are more convincing.

We agree with the reviewer's comment, butyrate concentrations in serum is very variable. In material and methods section, we detailed our procedure « For flow cytometry and serum analysis, mice were sacrificed 14 days post tumor inoculation, blood and draining lymph nodes were removed. Single cells were obtained after mechanical disruption. Serum samples were stored at -80°C until

further analysis of SCFA dosages". Since butyrate was given 14 days before tumor inoculation, blood samples were realized after 28 days from the beginning of butyrate intake.

The variability of water intake between mice is well known of researchers when adding sodium butyrate in the drinking water and is mainly due to a strong unpleasant smell. This repels some mice. This is why we have realized correlations between butyrate concentrations and B7 costimulatory molecules in mice treated with CTLA-4; this allows us to see the link with the concentrations despite this variability.

The smell is strong but that do not seem to repel all the mice (in our animal facility, our mice drink even more in the butyrate group than in the water group). If drink consumption is monitored, which is an easy control to perform to make sure about the treatment dosage by oral drink, this point will be addressed.

We have also monitored the volume of water in the cage of mice supplemented with C4 compared to the cage of mice not supplemented with C4. We confirmed that the volume of water diminished quiet similarly in both groups. However it seems difficult to estimate the amount of water drunk by each mouse in the same cage (n=5 mice per cage in our animal facility). Moreover, it is well known that there are always dominant mice in a cage, with the dominant mice having greater access to food and water. This also can contribute to variability in mice experiments. In our case, we did not said that mice supplemented with butyrate drink less than others (mice without butyrate); we tried to say that within mice supplemented with butyrate there is a possible variability due to variable drink intake between mice in the same cage (some mice being also annoyed by the smell). The serum butyrate assay in mice supplemented with butyrate clearly shows this variability regardless of the cause.

Also, the author did change the butyrate solution only once a week while changing every 2/3 days is commonly recommended.

In the different papers we have read, changing every 2/3 days butyrate was not really described (Nicholas Arpaia, Nature 2013 not clearly described "Butyrate was administered to mice after prior treatment with antibiotics for at least 1 wk."); Kim et al. J Med Food 2018 not specified; Hui et al international immunopharmacology 2019 : not specified "On the same day as the primary immunization, mice were provided with 100 mM sodium butyrate (Sigma Aldrich, St. Louis, MO) in drinking water for 5 weeks."...and a lot of more). In our knowledge butyrate is described as stable at least for 2 years in solution. From a chemical point of view, the conjugated bases R-COO⁻ of carboxylic acids are generally rather weak bases, the negative charge on the molecule is delocalized on the two oxygen atoms of the carboxyl group by mesomerism, which explains the relative stability of this type of molecules. For all these reasons, we supposed that changing at least once a week seemed acceptable to us.

A lot of inconsistencies have been found all along the manuscript in the patient numbers and description and need to be addressed (some prostate cancer patients, the number of patients depicted in figure 1 do not match the text and the table of patients' description and the data from the dosage of SCFA in the same compartment at the same time point are different across figures).

This is still true and concerning as some statistic seems to rely on outliers.

We don't feel confident with this commentary and we don't feel that our data are relying only on outliers. For Faecalibacterium the main results from figure 1 we have analyzed the data in three different ways: T-test comparing LTB and PB; T-test comparing OS>18m and OS<18m and Log-rank (Mantel-Cox) test usually used for clinical data. We think that if it were only due to outliers then it

would not always come out as significant. We have checked all data as said in the first revised version and we did thank the reviewer for that since it was not clear.

Even in the Point by point reply, the authors show data with 39 patients for microbiome while they are supposed to have 38 samples.

Error in the PBPR figure legend, this figure was not in the manuscript. We apologize for this PBPR's error.

Association of F. Prau with clinical outcomes. a. PFS and b. OS according to the median of *F. prausnitzii* (16S *F.prau*/ng DNA) assessed by Q-PCR analyses on feces at baseline. Thirty-eight patients were included in these analyses; Unpaired T tests with Welch's correction were used. p-values are indicated on each graph.

While they had 41 stool samples at V1 available in their first submission and only 38 now?

There were 41 stools in the first, second and third version. We have analyzed SCFA in the stools as first experiments since our goal was to show a link with SCFA. Then, after discussion with different colleagues that told us that a link with *F. Prau* should be done, we set-up a Q-PCR assay for *F. Prau*, *B. Fragilis* and *E. Coli*. Finally, other commentaries from colleagues convinced us to explore microbiota composition using NGS even though it was not our first goal. For these reasons and due to the chronology of experiments some stool aliquots were completely exhausted, explaining that we have 41 stool samples (SCFA assays) and only 38 left for the microbiota analyses. As explained, we have redo the Supplementary Table 2 and hope this is more clear.

The microbiome data seem to have been re-analyzed? The statistics in figure 1 have changed. Can the authors comment on this change?

Yes not significant statistics have been incorrectly copied, in order to avoid this we have replaced all not significant statistics throughout the manuscript with "ns". For *Faecalibacterium* this is an error we wrote 0.003 instead of 0.013 (first version), for Gemminger it is as in the first version 0.04 and not 0.02; we thank the reviewer for this.

Why the claim that *Roseburia* was linked with clinical benefit (PFS) in the first submission (text) is no longer true?

Roseburia was indeed linked to the OS and not PFS as showed in the first point by point reply which is a clinical benefit. However this sentence was somehow confusing since the clinical benefit mainly referred to LTB in our manuscript. This reason why we removed this sentence. To fit with reviewer comment and our data we will rephrase the sentence by clearly indicating that *roseburia* was found only associated with longer OS (cf figure next page).

Percentages of four genus at baseline and overall survival. Each patient is presented in the graph. 0-6M : overall survival ranging from 0 to 6 months (n=4), 6-12M : overall survival ranging from 6 to 12 months (n=12), 12-18M overall survival ranging from 12 to 18 months (n=11), >18M : overall survival >18 months (n=11). Kruskal-Wallis test followed by multiple comparisons were realized. * : $p < 0.05$.

The confusion with 50 Melanoma is still present in the manuscript while it's 48MM and 2 prostate cancer patients. Even if it's a majority of MM, the authors should not put MM only in the Figure legends or make sure the statistic are still true when considering only the MM patients.

Yes we agree with reviewer statement and we paid attention to modify this in our manuscript. Note that no prostate cancer patients were involved in NGS or Q-PCR data since stool aliquots were completely exhausted for the two patients (No prostate cancer in Figure 1; in figure 2 prostate cancer only in fig2 fghi; Fig S10). However we have generated other data from biological samples with these patients as SCFA dosage, blood... Thus we have re-run analyses without these two patients to be sure that the data remains the same. As shown in the following figures (only for this PBPR), clinical data for butyrate, propionate and acetate as well as correlations are in the same way without these two patients.

SCFA and clinical outcome without the two prostate cancer patients. **a**, Kaplan–Meier survival curves of PFS according to the median value of serum butyrate (left panel), propionate (middle panel) and acetate (right panel) concentrations in the pooled cohort (French + Italian; n=83 patients). **b**, Pearson correlations between serum of propionate, butyrate, acetate and PFS in the pooled cohort (French + Italian; n=83 patients). **c**, Pearson correlations between serum of propionate (C3) and butyrate (C4, left panel), acetate (C2) and propionate (middle panel) and acetate and butyrate (right panel) in the French cohort without prostate cancer patients (n=38).

- The role of propionate is suggested but not demonstrated as well as butyrate, other than the increase of Tregs in PBMC at baseline.

Since we have the data for two cohorts we pooled the data from the Italian and French cohort (n=85) and still both butyrate and propionate from pooled cohorts are associated to PFS while acetate failed to do so (revised Supp Fig. 10).

For the pooled cohort a Kaplan-Meier survival curves of PFS would be better.

We did realize a pooled analysis. It is now shown in the supplementary figure 10 (Kaplan-Meier survival curves).

Figure 3. The choices of the murine cell lines where two out of the three murine models used are colon cancer lines generate concern. Maybe some of the new melanoma cell lines (BRAF/PTEN, RET) could represent a better option.

First we used all tumor cell lines subcutaneously thus in the same microenvironment as we would have done for melanoma cell lines. Moreover and as described before the role of butyrate on all cell lines was tested *in vitro* and showed that these effects were not restricted to colon cancer cell lines but could also be observed with B16 (melanoma) as well as 4T1 (breast carcinoma) cell lines.

That said and as suggested by the reviewer we have search for a collaboration to have a melanoma cell lines, other than B16, such as BRAF/PTEN, RET. However after several messages, we never have any answers from the people that designed these cell lines and these cell lines are not commercially available. Thus we have used the B16K1 model (kindly given by Bruno Ségui, INSERM UMR 1037, CRCT, 31037 Toulouse, France). B16K1 is a genetically modified cell line generated from B16F10 cells, and that stably expresses the MHC-I molecule H-2Kb, known to stimulate CD8+ T cell-dependent immune responses. This cell line has been recently shown to respond to anti-PD-1 (and combo anti-PD-1/anti-CTLA-4) and is a melanoma tumor model (*Bertrand F. et al. Nature Communications volume 8, Article number: 2256 (2017)*). However this model was never tested using anti-CTLA-4 treatment as a standalone treatment. We have realized an experiment using this model to see if butyrate could limit anti-CTLA-4 activity also in this model. We have time to realize this experiment only once (n=5 mice per group). Unfortunately, anti-CTLA-4 treatment showed weak efficacy in this model compared to others such as the CT26 model. Moreover and contrary to CT26 or other cell lines used in our study, butyrate could delay the growth of the tumor, thus to be rigorous we had to normalized tumor growth data using ratio: STD_x corresponding to the surface of the tumor the day of treatment initiation and STD_x corresponding to the surface of the tumor at day x. STD_x to STD_i ratio (STD_x/STD_i) at each day was then calculated. As shown in this PBPR we still observed that butyrate could limit anti-CTLA-4 antitumor effect.

How the authors explain the effect of Butyrate as a stand-alone could delay the tumor growth?

We have showed in the first PBPR that butyrate could influence proliferation of several cell lines. In our hands tumor cell proliferation was inhibited by sodium butyrate *in vitro* as described by others B16 being particularly sensitive since at the lowest dose (0.1 μ M) we observed around 70%, 27% and 25% of cell reduction at 48h, 72h and 96h respectively. For MC38, CT26 and 4T1 at 0.1 μ M cell reduction was weak or absent. See the next graph. This is our main explanation for this cell line.

Sodium butyrate modified accumulation of the B16F10, MC38, CT26 and 4T1 cell line in vitro. In vitro proliferation of B16F10, MC38, CT26 and 4T1 cell lines after 48, 72 and 96h of culture in complete medium with escalating concentration of butyrate.

Also, the normalized results should be associated with the raw tumor growth data. Treating different groups with a different starting tumor size is always questionable.

Yes this is why normalization is sometimes needed since raw data are difficult to interpret. We agree with the reviewer that this is rarely done when looking at mice data while in the human setting it is realized systematically since the progression, stabilization or response is determined as the % of augmentation (or diminution) according to the initial volume of measurable target lesions (=since the experimental treatment started) and only an augmentation 20% is considered as a progression, a diminution of 30% is considered as Partial response (PR) otherwise we conclude to a stable disease (SD). We think that this type of normalization (when needed) should be more systematic in mice setting and could increase the reproducibility between mice and human data.

In our case we have demonstrated first in humans that SCFA seemed to limit ipilimumab clinical efficacy and it seemed to be true also in mice in 4 different tumor models.

- Comments on memory T cells and antigen specific T cells in mice There is still an accumulation/ increase of memory T cell following anti-CTLA4 blockade in the butyrate group and the two anti-CTLA4 treated groups have similar frequency of memory T cells. The data, even with supplemental figure 14 are still not strongly suggesting an ipilimumab-limited accumulation of memory T cell following butyrate Figure 1 a, The figure legend and text indicate that the analysis was performed on n=39 Metastatic Melanoma patients, however there are 42 bars. Please clarify what these 42 bars correspond to? This seems to correspond to the 41 stools samples available at V1 according to the supp table 2, but do not match the 40 cancer patients in supp table 1 (among the 40, 2 are prostate cancer). The patients should be clustered by clinical response to help the reader to see the increase of the genera mentioned in LTB versus PB.

There were 38 patients analyzed and depicted contrary to reviewer's remark. The confusion might come from the non-representation of samples F08, F13 and F27 on the figure. Indeed these samples F08, F13 and F27 were used as technical replicates from same patients. This was done to insure that the data could be reproducible from one experiment to another. The revised Figure 1 showed patients clustered by clinical response to help the reader as ask by reviewer 2.

There is a lot of switch in legends, figures, and patient numbers. This is a major concern, They were 42 bars on the original graph figure 1a, with a legend indicating n=39. Even if they were 3 duplicated patients, 42-3=39 not 38. No matter if the authors did replicates of same patients, it should be clarified in the legend. It's misleading as it's assumed that 1 bar=1 patient. The revised figure is much better and seems to have been reanalyzed to correspond to the figure legend but this is a general comment concerning for the general procedure of analysis/ data representation of the all manuscript that is creating concern about data accuracy.

There were 38 patients analyzed and depicted. The confusion might come from the non-representation technical replicates from same patients. This was done to insure that the data could be reproducible from one experiment to another. It seems to us that this was quite clear in material and methods: "Extractions series were realized during five consecutive days (5 series), for each series, stool sample from the same patient was extracted, this sample confirmed the reproducibility of analyzes". This means that the sample of the same patient was extracted 5 times: at Day1, Day 2, Day3, Day 4 and Day 5 but only 1 analysis of this patient was kept in final analyses (42-4=38). Sorry for the confusion in the first PBPR.

Moreover confusion might come from the numeration for NGS and PCR (F01, F02....) which are arbitrary numbers that have been attributed for each extracted samples for these specific experiments to protect confidentiality of patients' health data.

6. Did you attempt to stimulate dendritic cells (for example with TLR9 agonists) to overcome the influence of butyrate?

We thank the reviewer for this suggestion. We did an experiment with TLR9 agonists as suggested to overcome the influence of butyrate as suggested. In a very disappointing way we could not overcome the influence of butyrate in these new experimentations in mice. Note that the median survival for the butyrate + CTLR ODN + aCTLA-4 was 18 days while it was 42 days for water + ODN CTL + aCTLA-4 ($p < 0.05$) arguing again in this experimentation of an inhibition of antitumor effect of butyrate in mice treated with aCTLA-4. Finally median survival of mice treated with butyrate + CpG + aCTLA-4 was 18 days as for butyrate + CTLR ODN + aCTLA-4 demonstrating that CpG was not able to overcome the influence of butyrate in our hands. Supplementary Fig. 18. CpG agonists did not overcome butyrate inhibition in combination with anti-CTLA-4. Twenty mice were treated 2 weeks before tumor inoculation and all along the experiment with sodium butyrate in the drinking water (n=10) at the concentrations of 100mM or pH-matched water (n=10) and changed every week. Two hundred thousand of CT26 tumor cells were inoculated subcutaneously (s.c) in the right flank of Balb/c mice fed with sodium butyrate in the drinking water (right panel) or PH-matched water (left panel) at day 0. Mice received intraperitoneal (i.p) injections of 100 μ g of anti-CTLA-4 (aCTLA-4, n=5) or its isotype control (Isotype, n=5) at D7, D10 and D13. Mice received subcutaneously (s.c) injections of 30 μ g of CpG (CpG, n=5) or ODN control (CTRL ODN, n=5) at D7, D10 and D13. Tumor growth was measured three times a week. Mice were euthanized when the tumor size was ≥ 300 mm² or boundary points were reached. Percent survival of mice with PH-matched water (left panel); Percent survival of mice with butyrate (right panel); Log-rank (Mantel-Cox) test was used. P values are indicated on the graphs, * $p < 0.05$; ** $p < 0.01$; *** $p < 0.001$ otherwise not significant. One trivial explanation could be the CpG regimen i.e. unique dose of 30 μ g at day 7. Higher dose and/or repeated treatment could have been a better option. Another explanation for this unsuccessful treatment could rely on observation made using PBMC in *in vitro* experiments (revised figure 5 and 6) as shown and described previously in this PBPR. These new *in vitro* with Human PBMC data and *in vivo* data observed in mice thus suggest that butyrate is able to inhibit ipilimumab-induced DC maturation (less induction of CD80/86) but also inhibit CD28 signaling pathway. Thus restoration of CD80/CD86

through the use of TLR agonists that favor DC maturation such as TLR9 agonists (CpG) might not be sufficient to counter the deleterious effect of these SCFA. In consequence and in light of this new data the specific modulation of butyrate/propionate signaling pathway might be mandatory to restore anti-CTLA-4 efficacy in patients with high systemic levels of SCFA. SCFA are known to act also through specific receptors (GPR41, GPR43 and GPR109A) expressed on immune cells the use of GPR specific inhibitors might be a better option. However this has to be demonstrated and will be part of our future research. These new data and discussion are now added in the revised Figure 5 and 6 and new supplemental figure for CpG/butyrate data (new Supp Fig. 18).

The interpretation of this new data should be discussed. In order to correctly identify a difference between Butyrate and water in presence of TLR9 agonist and CTLA4 blockade, the groups should be presented on the same graph and the statistics calculated accordingly. The expression of the result as a median of survival when they are 5 animals per group do not seem to be appropriate. If one compares the actual survival of the groups receiving butyrate + anti-CTLA4+ CpG and the group receiving Water + anti-CTLA4+ CpG, there is still one mice alive at the end of the experiment in the butyrate + anti-CTLA4+ CpG group while none remaining in the water group. One could argue the survival is 20% in the butyrate versus 0% in the water group, CpG could indeed overcome the influence of butyrate. In order to support the conclusions, the experiments should be repeated.

We are surprised since this commentary was asked by the reviewer 3 and not the reviewer 2 as suggested here. Reviewer 3 seemed ok with data and interpretation. Since reviewer 2 add a comment on this experiment and suggest that our interpretation was misleading due to the survival of one mouse, we added in this PBPR the row data per mice. In our opinion the data clearly shows that in our setting CpG was not able to leverage [Butyrate + anti-CTLA-4 activity]. We really feel that the two groups (figure bellow) are comparable even though one mouse experienced a CR in one group and only a long stabilization in the other group. As explained before, it is not the primary objective of this paper to counter butyrate activity but rather ongoing research in our lab as said in our previous PBPR.

CpG agonists did not overcome butyrate inhibition in combination with anti-CTLA-4. Twenty mice were treated 2 weeks before tumor inoculation and all along the experiment with sodium butyrate in the drinking water (n=10) at the concentrations of 100mM or pH-matched water (n=10) and changed every week. Two hundred thousand of CT26 tumor cells were inoculated subcutaneously (s.c) in the right flank of Balb/c mice fed with sodium butyrate in the drinking water (right panel) or PH-matched water (left panel) at day 0. Mice received intraperitoneal (i.p) injections of 100 µg of anti-CTLA-4 (aCTLA-4, n=5) or its isotype control (Isotype, n=5) at D7, D10 and D13. Mice received subcutaneously (s.c) injections of 30 µg of CpG (CpG, n=5) or ODN control (CTRL ODN, n=5) at D7, D10 and D13. Tumor growth was measured three times a week. Mice were euthanized when boundary points were reached. CT26 tumor size are shown after treatment start (D0=D7) in each group. Tumor size over time after tumor inoculation in each group. Each line represents one mouse. Progressive disease (PD), stable disease (SD) and complete responses (CR) are indicated on each graph.

Also, accordingly to the statistics, butyrate does not seem to limit anti-CTLA4 efficacy in this experiment if you compare the groups treated with butyrate + anti-CTLA4+ ODN ctrl and the group treated with butyrate + Isotype + ODN ctrl (**), same statistic than in the water group.

We never said that butyrate + anti-CTLA4 did not have any anti-tumor effect we said that butyrate limits anti-CTLA-4 activity which is very different thus the right groups that should be compared are butyrate + anti-CTLA4 and water + anti-CTLA4. In this experiment water + anti-CTLA4 + CTRL ODN showed a better anti-tumor efficacy compared to butyrate + anti-CTLA4 + CTRL ODN (see next graphs). This is consistent with previous data.

REVIEWERS' COMMENTS:

Reviewer #2 (Remarks to the Author):

The authors addressed the reviewer's concern.